# Fossil History of Curculionoidea (Coleoptera) from the Paleogene

**Andrei A. Legalov** [1,2] 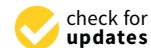

[1]   Institute of Systematics and Ecology of Animals, Siberian Branch, Russian Academy of Sciences, Ulitsa Frunze, 11, 630091 Novosibirsk, Novosibirsk Oblast, Russia; fossilweevils@gmail.com; Tel.: +7-9139471413

[2]   Biological Institute, Tomsk State University, Lenin Ave, 36, 634050 Tomsk, Tomsk Oblast, Russia

**Abstract:** Currently, some 564 species of Curculionoidea from nine families (Nemonychidae—4, Anthribidae—33, Ithyceridae—3, Belidae—9, Rhynchitidae—41, Attelabidae—3, Brentidae—47, Curculionidae—384, Platypodidae—2, Scolytidae—37) are known from the Paleogene. Twenty-seven species are found in the Paleocene, 442 in the Eocene and 94 in the Oligocene. The greatest diversity of Curculionoidea is described from the Eocene of Europe and North America. The richest faunas are known from Eocene localities, Florissant (177 species), Baltic amber (124 species) and Green River formation (75 species). The family Curculionidae dominates in all Paleogene localities. Weevil species associated with herbaceous vegetation are present in most localities since the middle Paleocene. A list of Curculionoidea species and their distribution by location is presented.

**Keywords:** Coleoptera; Curculionoidea; fossil weevil; faunal structure; Paleocene; Eocene; Oligocene

## 1. Introduction

Research into the biodiversity of the past is very important for understanding the development of life on our planet. Insects are one of the Main components of both extinct and recent ecosystems. Coleoptera occupied a special place in the terrestrial animal biotas of the Mesozoic and Cenozoics, as they are characterized by not only great diversity but also by their ecological specialization. The largest superfamily in the Coleoptera is the Curculionoidea, which, among beetles, is one of the Main plant consumers. They develop in living or dead tissues of various plant organs as well as in soil, feeding on roots. As an exception, some weevil larvae can develop completely exposed on plants, as well as on the surface of the soil as detritivores or predators.

The purpose of this study is to evaluate the diversity of Curculionoidea in the Paleogene and to show the specificity of faunas in various epochs, ages and localities. This is the first time a review of the Paleogene Curculionoid beetle has been undertaken.

## 2. Materials and Methods

The Paleogene is the first period of the Cenozoic, consisting of three epochs, the Paleocene, Eocene and Oligocene (Figure 1). The Paleogene started at about 66 Ma and ended around 23 Ma. The Late Cretaceous preceded it. The Miocene (from the Neogene) began after the Paleogene. The Paleocene consists of three ages (Danian, Selandian and Thanetian). The Eocene is subdivided into three subepochs (Early, Middle and Late) with four ages (Ypresian, Lutetian, Bartonian and Priabonian). The Bridgerian stands out for the North American Provincial Ages at the end of the Ypresian and the beginning of the Lutetian. This age corresponds with the Green River Formation. The Oligocene consists of two subepochs (Early and Late) and two ages (Rupelian and Chattian).

| Period | Epoch | Subepoch | Age | |
|---|---|---|---|---|
| Paleogene | Paleocene 65.5±0.3–55.8±0.2 | | Danian 65.5±0.3–61.7±0.2 | |
| | | | Selandian 61.7±0.2–58.7±0.2 | |
| | | | Thanetian 58.7±0.2–55.8±0.2 | |
| | Eocene 55.8±0.2–33.9±0.1 | Early 55.8±0.2–48.6±0.2 | Ypresian 55.8±0.2–48.6±0.2 | |
| | | | | Bridgerian 50.3–46.2 |
| | | Middle 48.6±0.2–37.2±0.1 | Lutetian 48.6±0.2–40.4±0.2 | |
| | | | Bartonian 40.4±0.2–37.2±0.1 | |
| | | Late 37.2±0.1–33.9±01 | Priabonian 37.2±0.1–33.9±0.1 | |
| | Oligocene 33.9±0.1–23.03±0.1 | Early 33.9±0.1–28.4±0.1 | Rupelian 33.9±0.1–28.4±0.1 | |
| | | Late 28.4±0.1–23.03±0.1 | Chattian 28.4±0.1–23.03±0.1 | |

**Figure 1.** Geochronology of the Paleogene (Ma).

Curculionoidea have been found from 53 localities in 18 countries (Figure 2, Figure 3, Figure 4, Figure 5, Figure 6, Figure 9, Figure 12, Figure 14, Figure 20, Figure 21), spanning all ages of the Paleogene (Table 1).

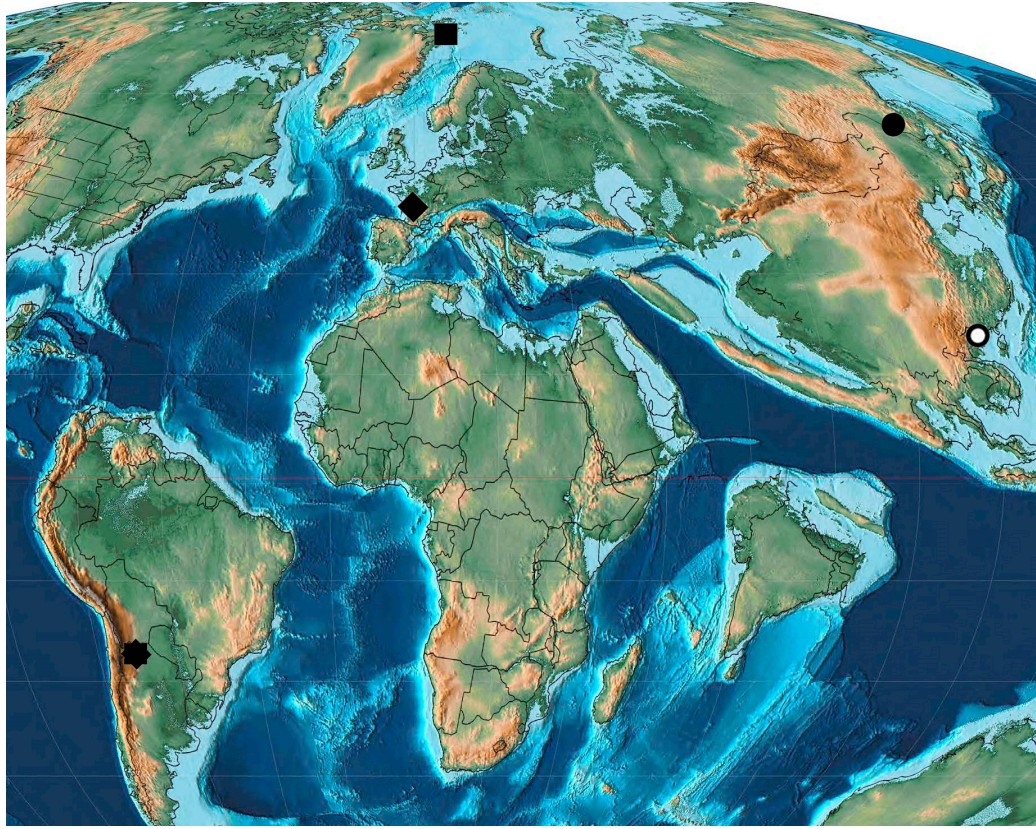

**Figure 2.** Paleocene Curculionoidea deposits: octagon—Sunchal; square—Starostin; rhombus—Menat; circle—Arkhara; ring—Mirs Bay.

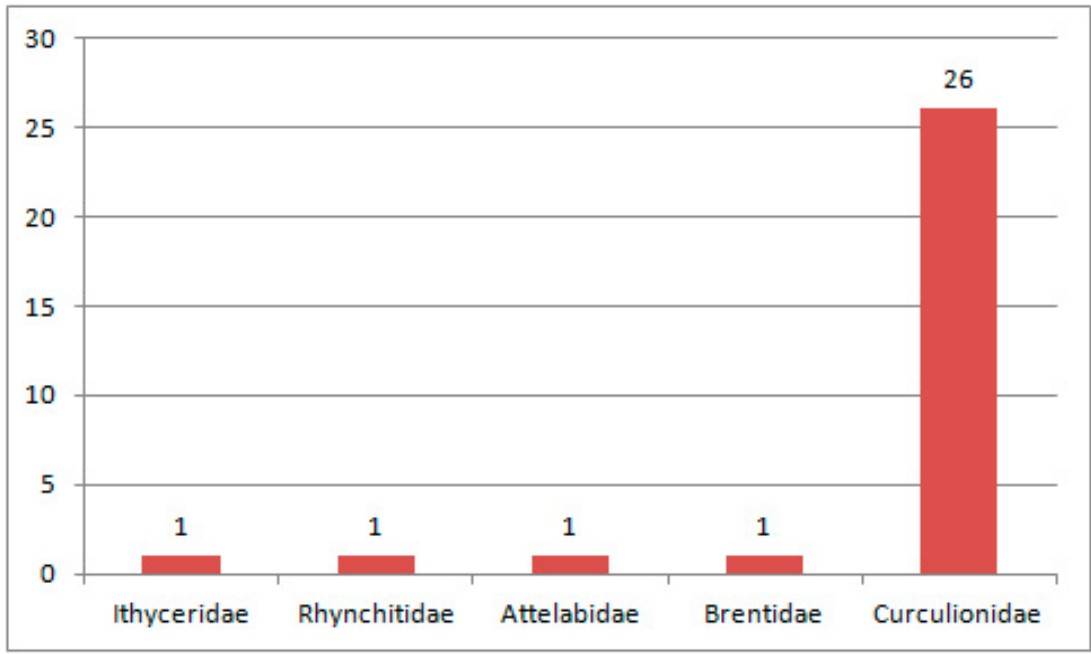

**Figure 3.** Composition of species of Curculionoidea in the Paleocene fauna.

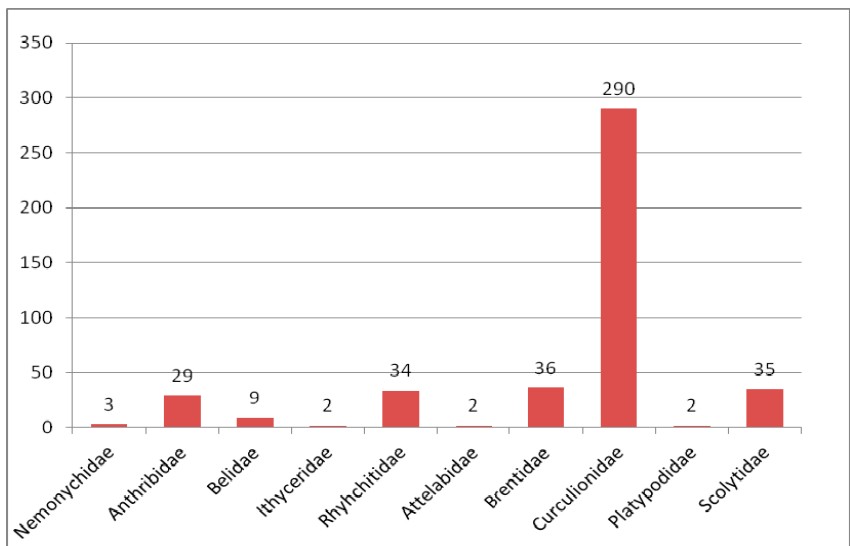

**Figure 4.** Composition of species of Curculionoidea in the Eocene fauna.

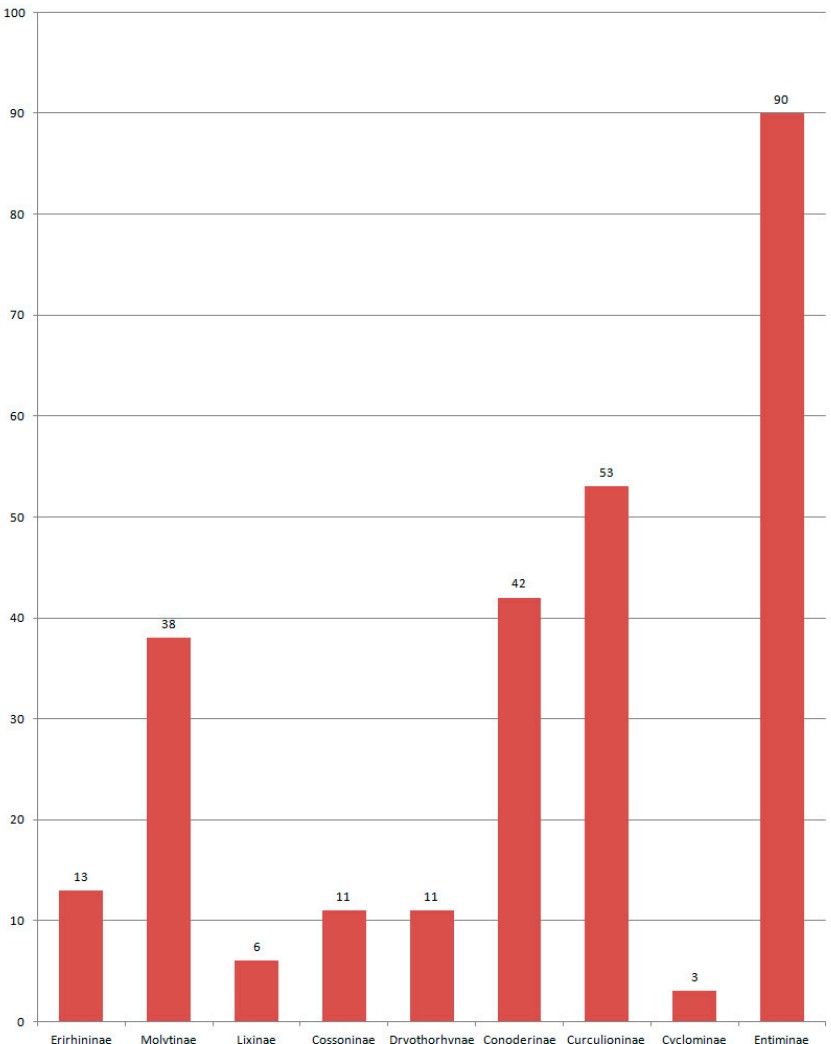

**Figure 5.** Composition of species of Curculionidae in the Eocene fauna.

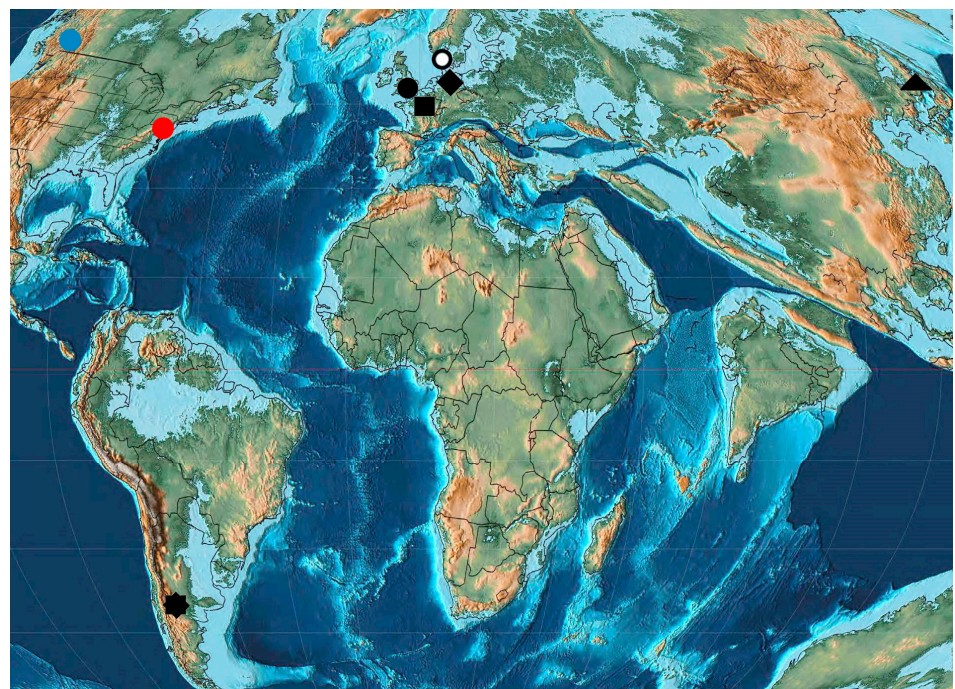

**Figure 6.** Early Eocene Curculionoidea deposits: square—Oise amber; circle—Peckham and London Clay; rhombus—Havighorst; ring—Mors; red circle—Republic; blue circle—Quilchena; triangle—Tadushi; octagon—Huitrera Formation.

**Table 1.** Localities with Curculionoidea from the Paleogene.

| Name | Abbreviation | Location | Age |
|---|---|---|---|
| Arkhara | Arkh (P1d) | Russia: Amurskaya Region, Arkharinskii District, quarry at Arkhara Railway Station | Lower Paleocene, Danian, Cagayan |
| Starostin | Star (P1d) | Norway: Spitsbergen, Svalbard and Jan Mayen, Firkanten Formation | Lower Paleocene, Danian, 63 ± 2 Ma |
| Sunchal | Sunc (P1d) | Argentina: northern Argentina, Jujuy Province, La Mendieta, Maiz Gordo Formation | Lower Paleocene, Danian, 66.0−55.8 Ma |
| Menat | Mena (P1sl-t) | France: Puy-de-Dome | Middle-Upper Paleocene, Selandian-Thanetian, 61.0−59.0 Ma |
| Mirs Bay | Mirs (P1) | China: Hong Kong, Peng Chau Island, gray sandy shales, Ping Chau Formation | Paleocene |
| Mors | Mors (P2i) | Denmark, Fur Formation | Lower Eocene, Ypresian, 54.0 Ma |
| London Clay | LonC (P2i) | United Kingdom: England: Sussex, Bognor Regis | Lower Eocene, Ypresian, 54.0–50.0 Ma |
| Roan Mountain | RoaM (P2i-l) | United States: Colorado, Green River Formation | Lower-Middle Eocene, Bridgerian, 53.5–48.5 Ma |
| Green River | GreR (P2i-l) | United States: Colorado, Wyoming, Utah States, 3–4 km west of railway crossing of Green River, Green-River Formation | Lower-Middle Eocene, Bridgerian, 53.5–48.5 Ma |
| Oise amber | OisJ (P2i) | France: Paris basin, Creil, Oise | Lowermost Eocene, Ypresian, 53.0 Ma |

**Table 1.** *Cont.*

| Name | Abbreviation | Location | Age |
|---|---|---|---|
| Huitrera Formation | | Argentina: Arroyo Chacay, Río Negro, near Estancia Don Hipólito, W, about 60 km east from San Carlos de Bariloche, Huitrera Formation | Lower Eocene, Ypresian, 54.24 ± 0.45 Ma |
| Peckham | **Peck (P2i)** | England: South London, Reding Beds | Lower Eocene, Ypresian |
| Havighorst | **Havi (P2i)** | Germany: east of Hamburg, Schleswig Holstein | Lower Eocene, Ypresian |
| Quilchena | | Canada: British Columbia, 3 km south of Nicola Lake, lacustrine shale | Lower Eocene, Ypresian, 51.5 ± 0.4 Ma |
| Republic | **Repu (P2i)** | United States: Northeast Washington State, Klondike Mountain Formation | Lower Eocene, Ypresian, 49.0–48.0 Ma |
| Tadushi | **Tadu (P2i)** | Russia: Russian Far East, Primorsky Krai, Kavalerovsky District, Pestrushka River near mouth and Ugol'nyi Creek, tributary of Zerkal'nyi (Tadushi) River, near village of Suvorovo, Tadushi Formation | Lower Eocene, Ypresian |
| Messel | **Mess (P2l)** | Germany: near Frankfurt, oil shales | Middle Eocene, Upper Ypresian Lower Lutetian, 48.27 ± 0.22–47.0 Ma |
| Corfe | **Corf (P2l)** | United Kingdom: England: Dorset, Corfe Clay, Lower Bagshot Beds | Middle Eocene, Lutetian, |
| Bournemouth | **Bour (P2l)** | United Kingdom: England, Dorset, Bagshot Series | Middle Eocene, Lutetian, 50.0–42.0 Ma |
| Geiseltal | **Geis (P2l)** | Germany: near Halle | Middle Eocene, Lutetian, 47.5–42.5 Ma |
| Eckfelder Maar | **Maar (P2l)** | Germany: Rheinland-Pfalz, Rhine Palatine | Middle Eocene, Lutetian, 44.3 Ma |
| Baltic amber (Kaliningrad region) | **BalJ (P2b)** | Russia: Kaliningrad Region, Baltic Sea coast and Amber Jantarnyi quarry near Kaliningrad, Prussian Formation | Middle Eocene, Bartonian, 48.0–33.0 Ma |
| Polish amber | **PolJ (P2b)** | Poland: Gdansk city area, at the Wisla River Estuary, Baltic amber, Prussian Formation; | Middle Eocene, Bartonian, 48.0–33.0 Ma |
| Scandinavian amber | **ScanJ (P2b)** | Denmark, amber deposits on the Danish coast, Baltic amber, Prussian Formation | Middle Eocene, Bartonian, 48.0–33.0 Ma |
| Rovno amber | **RovJ (P2b)** | Ukraine: Rovno Region, Klesov and Dubrovice quarries | Middle Eocene, Bartonian, 48.0–33.0 Ma |
| Bitterfeld amber | | Germany: Saxony-Anhalt, Goitzsche near Bitterfeld | Middle Eocene, Bartonian, 48.0–33.0 Ma |
| Romanian amber | | Romania: Alunisului Hill, Valea Sibiciului, Colti | Middle Eocene, Bartonian, 48.0–33.0 Ma |
| Kutschlin | **Kuts (P2b)** | Czech Republic, northwest Bohemia, near Bilina | Middle Eocene, Bartonian |
| Celas | **Cela (P2p)** | France: Gard Department, railway Uze's–Saint-Julien-de-Casig-nac, Fumades, Corents, Bassein Ales | Upper Eocene, Priabonian |

**Table 1.** *Cont.*

| Name | Abbreviation | Location | Age |
|---|---|---|---|
| Ales-Monteils | **Ales (P2p)** | France: Gard | Upper Eocene, Priabonian |
| Florissant | **Flor (P2p)** | United States: Colorado, Rocky Mountains near Pike's Peak, Florissant Formation | Latest Eocene, Priabonian, 34.07 ± 0.10 Ma |
| Biamo | **Biam (P2p)** | Russia: Russian Far East, Primorsky Region, Pozharskii District, source of Barachek Creek, right tributary of Bol'shaya Svetlovodnaya (formerly Biamo) River | Upper Eocene, Priabonian |
| Fonseca | **Fons (P2p)** | Brazil: Minas Gerais State, near Fonseca municipality, Whitish shales, under bituminous shales, Fonseca Basin, Fonseca Formation | Oligocene, Priabonian |
| White River Badlands | **WhiR (P3r)** | United States: South Dakota | Lower Oligocene, Rupelian, 34.0–31.2 Ma |
| Brunnstatt | **Brun (P3r)** | France: Alsace, Haut-Rhine Dept., 5 km southwest of Mulhouse | Lower Oligocene, Rupelian, 33.9 Ma |
| Corent | **Core (P3r)** | France: South France, Puy-de-Dom Department, Gergovia Plateau, south of Clermon-Ferran | Lower Oligocene, Rupelian |
| Cereste | | France: Alpes-de-Haute Province, Basses Alp Department | Lower Oligocene, Rupelian |
| Seifhennersdorf | | Germany: Saxony, Oberlausitz | Lower Oligocene, Rupelian, 30.5–30.2 Ma |
| Ahuehuetes | | Mexico: Puebla, 4.5 km NNE from Tepexi de Rodrigues, Los Ahuehuetes, Coatzingo Formation | Lower Oligocene, Rupelian |
| Kundratice | | Czech Republic: near Litomioice and Seifhennersdorf | Lower Oligocene, Rupelian |
| Sieblos | **Sieb (P3)** | Germany: Hessen, Rhon | Middle Oligocene |
| Kleinkembs | **Klei (P3)** | Germany: Baden-Wurtemberg, 10 km northwest of Lorrach, Pays de Bade, Salt Formation | Middle Oligocene |
| Gaube | **Gaub (P3)** | France: Puy-de-Dome, ravin de la Gaube | Lower Oligocene, Rupelian |
| Puy-Saint-Jean | **PuStJ (P3)** | France: Puy-de-St. Jean | Middle Oligocene |
| Luzice | **Luzi (P3h)** | Czech Republic: northwest Bohemia | Upper Oligocene, Chattian |
| Enspel | | Germany: Rheinland-Pfalz, Westerwald, Bad Marienberg | Upper Oligocene, Chattian, 24.79 – 24.56 Ma |
| Ashutas | | Kazakhstan: East Kazakhstan Region, Kurshim District, right bank of River Cherny Irtysh | Upper Oligocene, Chattian |
| Perekishkyul' | | Azerbaijan: middle course of Sumgait River, village of Perekeshkul, Maikop Formation | Upper Oligocene, Chattian |
| Rott | **Rott (P3h)** | Germany: North Rhine-Westfalia, Siebengebirge, near Bonn, Rott Formation | Latest Oligocene, Upper Chattian, 24.0–23.0 Ma |
| Aix-en-Provance | **Aix (P3h)** | France: Bouches-du-Rhane, Aix-en-Provence Formation | Latest Oligocene, Upper Chattian |
| Kap Dalton | **KapD (P3)** | Denmark: Greenland | Oligocene |
| Kenderlyk II | | Kazakhstan: East Kazakhstan Region, Zaisan district, left bank of Kenderlyk River, 6 km from village of Kenderlyk | Oligocene |

Maps for Paleogene localities were from Scotese [1].

The ages of fossil deposits used are from the website Fossilworks—http://fossilworks.org and some publications [2–21]. Baltic amber from the Prussian Formation ranges from 33–48 million years (Lutetian-Bartonian) [22] and in this work the Bartonian age is accepted [23].

The systematics of the superfamily Curculionoidea is currently not stable [24–30]. Previously, it included the Mesozoic family Obrieniidae, the systematic position of which is assumed to be related to the Curculionoidea [31] or considered as a family within the Archostemata [32] or a separate superfamily within the Curculioniformia [30,33] and is not considered in this paper. The number of families, groups related to these families and their taxonomic status are not universally accepted [24–30,34–36]. Also the families Nemonychidae, Anthribidae, Belidae, Ithyceridae, Brentidae, Curculionidae, Platypodidae and Scolytidae not universally accepted. In this work the higher classification proposed by the author [29,30,37–42] is adopted. The subfamily Cimberidinae is considered part of the family Nemonychidae [26,29,37,38,43], in contrast to the opinion of Seunggwan et al. [44]. The families Anthribidae and Belidae are accepted in the traditional composition [25,27,29,30,32,38,45]. Ithyceridae and Caridae are often regarded as unrelated groups [25–28,32,36,44]; however, the author considers the family Ithyceridae (incl. Caridae and Ulyanidae) as a diverse, predominantly extinct group, including five fossil (Mongolocarinae, Baissorhynchinae, Montsecanomalinae, Ulyaninae, Slonikinae) and three recent subfamilies (Carinae, Chilecarinae, Ithycerinae) [29,30,38,46–51]. Rhynchitidae and Attelabidae are considered as independent families [30,38,52–61]. The family Brentidae consists of six subfamilies [38]. Nanophyinae and Apioninae, sometimes considered as separate families [26,62–64], are included in Brentidae [25,27,30,32,36,38]. The family Curculionidae includes Erirhininae according to Zherikhin and Egorov [65] and Legalov [30,41]. Scolytidae and Platypodidae are considered as separate families [29,30,34,66].

Many Paleogene species were described in the 19th - first half of the 20th century in modern genera. The descriptions and illustrations of Many have been re-studied. If the generic affiliation was in doubt, then the genus has been placed in quotes. The results of these studies were presented by Legalov [30,47,50,67–71].

The studied fossil forms are deposited in—A. Bukejs's collection, Daugavpils, Latvia; A. Górski's collection, Bielsko-Biala, Poland; Borissiak Paleontological Institute of the Russian Academy of Sciences, Moscow, Russia; C. Gröhn's collection (Glinde, Germany) deposited in the Center of Natural History (formerly Geological–Paleontological Institute and Museum), Hamburg, Germany; Center of Natural History (formerly Geological–Paleontological Institute and Museum), Hamburg, Germany; Centre de Conservation du musée des confluences, Lyon, France; Earth Institute, Warsaw, Poland; F. Kernegger's collection, Hamburg, Germany, deposited in the Forschungsinstitut Senckenberg, Frankfurt am Main, Germany; Friedhelm Eichmann, Hannover, Germany; Górnośląskie Muzeum Przyrodnicze w Bytomiu, Poland; Kaliningrad Regional Amber Museum, Kaliningrad, Russia; Legalov's fossil insects collection Maintained at Institute of Systematics and Ecology of Animals of the Siberian Branch of the Russian Academy of Science, Novosibirsk, Russia; Muséum national d'histoire naturelle, Paris, France; Museum of Amber Inclusions, University of Gdańsk, Poland; Museum of the World Ocean, Kaliningrad, Russia; Naturhistorisches Museum Mainz, Landessammlung für Naturkunde Rheinland-Pfalz; Poinar amber collection Maintained at Oregon State University, Corvallis, OR, USA; Schmalhausen Institute of Zoology of the National Academy of Sciences of Ukraine, Kiev, Ukraine; V. Alekseev's collection, Kaliningrad, Russia; V. Gusakov's collection, Russia, Moscow; Zoological Museum, University of Copenhagen, Denmark.

The specimens were studied using a stereomicroscope Zeiss Stemi 2000-C in the Institute of Systematic and Ecology of Animals (Novosibirsk, a Leica M165C binocular microscope, with Leica DFG 425, MBS 10, MBS 12 in the Paleontological Institute (Moscow), a Leica MZ 16.0 stereomicroscope with a DFC290 camera in the Zoological Institute (St. Petersburg) and an Olympus SCX9 stereomicroscope with an Olympus camera and a Nikon SMZ1500 with Microscope Eyepiece Camera 9.0MP Aptina Color CMOS MU900 in the Muséum National d'Histoire Naturelle (Paris).

The present work is registered in ZooBank (www.zoobank.org) under LSID urn—lsid:zoobank.org:pub:125CCA9E-7288-4C12-88C9-BDEBDC300A9C.

## 3. Results

Four species of the family Nemonychidae, 33 species of the family Anthribidae, three species of the family Ithyceridae, nine species of the family Belidae, 41 species of the family Rhynchitidae, three species of the family Attelabidae, 47 species of the family Brentidae, 384 species of the family Curculionidae, two species of the family Platypodidae and 37 species of the family Scolytidae were described from the Paleogene (Table 2).

**Table 2.** List of species of Curculionoidea from Paleogene [8,29,30,47,50,67,69–175].

| No. | Taxon | Locality | Age | Sources |
|---|---|---|---|---|
| | Nemonychidae Bedel, 1882 Cretonemonychinae Gratshev et Legalov, 2009 Eocaenonemonychini Legalov, 2013 | | | |
| 1 | *Eocaenonemonyx kuscheli* Legalov, 2013 | GreR | P2i-l | 72 |
| 2 | *"Sitones" grandaevus* Scudder, 1876 | GreR | P2i-l | 73 |
| 3 | *"Eugnamptus" decemsatus* Scudder, 1878 | KapD | P3 | 74 |
| | Cimberidinae Gozis, 1882 Kuschelomacerini Riedel, 2010 | | | |
| 4 | *Kuschelomacer kerneggeri* Riedel, 2010 | BalJ | P2b | 75 |
| | Anthribidae Billberg, 1820 Anthribinae Billberg, 1820 Cratoparini LeConte, 1876 | | | |
| 5 | *"Euparius" elusus* (Scudder, 1878) | GreR | P2i-l | 74 |
| 6 | *"E." repertus* (Scudder, 1878) | Flor | P2p | 74 |
| 7 | *"E." adumbratus* (Wickham, 1911) | Flor | P2p | 76 |
| 8 | *"E." arcessitus* (Scudder, 1893) | Flor | P2p | 77 |
| | Anthribini Billberg, 1820 | | | |
| 9 | *"Ormiscus" partitus* (Scudder, 1890) | GreR | P2i-l | 78 |
| 10 | *"Trigonorhinus" pristinus* (Scudder, 1876) | GreR | P2i-l | 73 |
| 11 | *"T." sordidus* (Scudder, 1893) | Flor | P2p | 77 |
| | Ecelonerini Lacordaire, 1865 | | | |
| 12 | *Pseudochirotenon eocaenicus* Legalov, 2018 | GreR | P2i-l | 69 |
| 13 | *Saperdirhynchus prisctitillator* Scudder, 1893 | Flor | P2p | 77 |
| | Mecocerini Lacordaire, 1865 | | | |
| 14 | *Pseudoecocerus aleksevi* Legalov, 2020 | BalJ | P2b | 71 |
| | Tropiderini Lacordaire, 1865 | | | |
| 15 | *"Tropideres" remotus* Scudder, 1893 | GreR | P2i-l | 77 |
| 16 | *"T." vastats* Scudder, 1893 | Flor | P2p | 77 |
| | Allandrini Pierce, 1930 | | | |
| 17 | *Pseudomecorhis orlovi* Zherikhin, 1971 | BalJ | P2b | 79 |
| 18 | *P. simulator* Voss, 1953 | BalJ | P2b | 80 |
| 19 | *Allandroides vossi* Legalov, 2015 | BalJ | P2b | 30 |
| | Oiserhinini Legalov, Kirejtshuk et Nel, 2019 | | | |
| 20 | *Oiserhinus insolitus* Legalov, Kirejtshuk et Nel, 2019 | OisJ | P2i | 81 |
| | Zygaenodini Lacordaire, 1865 | | | |
| 21 | *Glaesotropis (Pseudoglaesotropis) balticus* Legalov, 2020 | BalJ | P2b | 71 |

**Table 2.** *Cont.*

| No. | Taxon | Locality | Age | Sources |
|---|---|---|---|---|
| 22 | *G. (P.) martynovi* (Legalov, 2012) | BalJ | P2b | 82 |
| 23 | *G. (Glaesotropis) diadiasashai* Gratshev et Perkovsky, 2008 | RovJ | P2b | 83 |
| 24 | *G. (G.) gusakovi* Legalov, 2015 | BalJ | P2b | 30 |
| 25 | *G. (G.) minor* Gratshev et Zherikhin, 1995 | BalJ | P2b | 84 |
| 26 | *G. (G.) succiniferus* Legalov, 2015 | BalJ | P2b | 30 |
| 27 | *G. (G.) weitschati* Gratshev et Zherikhin, 1995 | BalJ | P2b | 84 |
| 28 | *G. (Electranthribus) alleni* Legalov, 2015 | BalJ | P2b | 30 |
| 29 | *G. (E.) gratshevi* Legalov, 2015 | BalJ | P2b | 30 |
| 30 | *G. (E.) zherikhini* (Legalov, 2013) | BalJ | P2b | 47 |
| | Tribe incertae sedis | | | |
| 31 | *Stiraderes conradi* Scudder, 1893 | Flor | P2p | 77 |
| | Choraginae W. Kirby, 1819<br>Choragini W. Kirby, 1819 | | | |
| 32 | *"Choragus" fictilis* Scudder, 1890 | GreR | P2i-l | 78 |
| 33 | *"Ch." tertiarius* Heyden et Heyden, 1866 | Rott | P3h | 85 |
| | Valenfriesiini Alonso-Zarazaga et Lyal, 1999 | | | |
| 34 | *Eduardoxenus unicus* Legalov, Nazarenko et Perkovsky, 2018 | RovJ | P2b | 86 |
| | Urodontinae C.G Thomson, 1859 | | | |
| 35 | *"Bruchela" cincta* (Foerster, 1891) | Brun | P3r | 87 |
| 36 | *"B." priscus* (C. Heyden, 1862) | Rott | P3h | 88 |
| 37 | *"B." multipunctata* (Schlechtendal, 1894) | Rott | P3h | 89 |
| | Ithyceridae Schoenherr, 1823<br>Chilecarinae Legalov, 2009<br>Chilecarini Legalov, 2009 | | | |
| 38 | *Petropsis rostratus* Legalov, Kirejtshuk et Nel, 2017 | Mena | P1sl-t | 50 |
| | Ithycerinae Schoenherr, 1823 | | | |
| 39 | *Eoceneithycerus carpenteri* Legalov, 2013 | Repu | P2i | 47 |
| 40 | *Ithyceroides klondikensis* Legalov, 2015 | Repu | P2i | 30 |
| | Belidae Schoenherr, 1826<br>Oxycoryninae Schoenherr, 1840<br>Oxycraspedini Marvaldi et Oberprieler, 2006 | | | |
| 41 | *Oxycraspedus poinari* Legalov, 2016 | BalJ | P2b | 90 |
| | Metrioxenini Voss, 1953<br>Metrioxenina Voss, 1953 | | | |
| 42 | *Archimetrioxena electrica* Voss, 1953 | BalJ | P2b | 80 |
| 43 | *A. zherikhini* (Legalov, 2012) | BalJ | P2b | 82 |
| | Zherichinixenina Legalov, 2009 | | | |
| 44 | *Paltorhynchus bisculcatus* Scudder, 1893 | RoaM | P2i-l | 77 |
| 45 | *P. narwhal* Scudder, 1893 | Flor | P2p | 77 |
| 46 | *Succinometrioxena attenuata* Legalov et Poinar, 2020 | BalJ | P2b | 91 |
| 47 | *S. bachofeni* Legalov, 2013 | BalJ | P2b | 47 |
| 48 | *S. poinari* Legalov, 2012 | BalJ | P2b | 92 |
| | Allocorynitae Sharp, 1890<br>Palaeorhopalotriini Legalov, 2013 | | | |
| 49 | *Palaeorhopalotria neli* Legalov, 2013 | Ales | P2p | 47 |
| | Rhynchitidae Gistel, 1856<br>Sayrevilleinae Legalov, 2003<br>Sanyrevilleini Legalov, 2003 | | | |

**Table 2.** *Cont.*

| No. | Taxon | Locality | Age | Sources |
|---|---|---|---|---|
| 50 | *Baltocar convexus* Legalov, 2015 | BalJ | P2b | 30 |
| 51 | *B. groehni* Riedel, 2012 | BalJ | P2b | 93 |
| 52 | *B. hoffeinsorum* Riedel, 2012 | BalJ | P2b | 93 |
| 53 | *B. subnudus* Riedel, 2012 | BalJ | P2b | 93 |
| 54 | *B. succinicus* (Voss, 1953) | BalJ | P2b | 80 |
| | Vossicartini Legalov, 2003 | | | |
| 55 | *Germanocartus heydeni* (Schlechtendal, 1894) | Rott | P3h | 89 |
| | Rhynchitinae Gistel, 1856 | | | |
| | Auletini Desbrochers des Loges, 1908 | | | |
| | Auletina Desbrochers des Loges, 1908 | | | |
| 56 | *Electrauletes unicus* Legalov, 2015 | BalJ | P2b | 30 |
| | Pseudauletina Voss, 1933 | | | |
| 57 | *Eoropseudauletes plucinskii* Kania et Legalov, 2019 | BalJ | P2b | 94 |
| | Pseudomesauletina Legalov, 2003 | | | |
| 58 | *Pseudomesauletes groehni* Bukejs et Legalov, 2019 | RovJ | P2b | 95 |
| 59 | *P. culex* (Scudder, 1893) | Flor | P2p | 77 |
| 60 | *P. ibis* (Wickham, 1912) | Flor | P2p | 96 |
| 61 | *P. obliquus* (Wickham, 1913) | Flor | P2p | 97 |
| 62 | *P. striaticeps* (Wickham, 1911) | Flor | P2p | 76 |
| | Subtribe incertae sedis | | | |
| 63 | *"Teretrum" quiescitum* Scudder, 1893 | GreR | P2i-l | 77 |
| 64 | *T. primulum* Scudder, 1893 | Flor | P2p | 77 |
| 65 | *Docirhynchus terebrans* Scudder, 1893 | Flor | P2p | 77 |
| 66 | *"Trypanorhynchus" depratus* Scudder, 1893 | Flor | P2p | 77 |
| 67 | *"Paltorhynchus" rectirostris* Scudder, 1893 | Flor | P2p | 77 |
| 68 | *"Trypanorhynchus" sedatus* Scudder, 1893 | Flor | P2p | 77 |
| | Rhynchitini Gistel, 1856 | | | |
| | Temnocerina Legalov, 2003 | | | |
| 69 | *Eocenorhynchites vossi* Legalov, 2012 | BalJ | P2b | 74 |
| | Perrhynchitina Legalov, 2003 | | | |
| 70 | *Succinorhynchites alberti* Legalov, 2013 | BalJ | P2b | 47 |
| 71 | *Tatianaerhynchites goergesi* (Zherikhin, 1992) | Rott | P3h | 98 |
| | Rhynchitina Gistel, 1856 | | | |
| 72 | *Cartorhynchites struvei* Zherikhin, 1992 | Rott | P3h | 98 |
| 73 | *Opacoinvolvulus rottensis* (Zherikhin, 1992) | Rott | P3h | 98 |
| 74 | *O. zherichini* Legalov, 2003 | Rott | P3h | 55 |
| | Subtribe incertae sedis | | | |
| 75 | *"Rhysosternum" punctatolineatum* Piton, 1940 | Mena | P1sl-t | 99 |
| 76 | *Isothea alleni* Scudder, 1893 | Flor | P2p | 77 |
| | =*Trypanorhynchus corruptivus* Scudder, 1893 | | | 77 |
| 77 | *Prodeporaus curiosum* (Scudder, 1893) | Flor | P2p | 77 |
| 78 | *P. exanimale* (Scudder, 1893) | Flor | P2p | 77 |
| 79 | *P. exilis* (Wickham, 1913) | Flor | P2p | 97 |
| 80 | *P. minutissimus* (Wickham, 1913) | Flor | P2p | 97 |
| 81 | *P. smithii* (Scudder, 1893) | Flor | P2p | 77 |
| 82 | *"Prodeporaides" laminarum* (Wickham, 1916) | Flor | P2p | 100 |
| 83 | *"P." subterraneus* (Scudder, 1893) | Flor | P2p | 77 |
| 84 | *"P." vulcan* (Wickham, 1916) | Flor | P2p | 100 |
| 85 | *P. wymani* (Scudder, 1893) | Flor | P2p | 77 |
| 86 | *"Masteutes" saxifer* Scudder, 1893 | Flor | P2p | 77 |
| 87 | *"Rhynchites" hageni* Heyden et Heyden, 1866 | Rott | P3h | 85 |

**Table 2.** *Cont.*

| No. | Taxon | Locality | Age | Sources |
|---|---|---|---|---|
| | Eugnamptini Voss, 1930 | | | |
| 88 | *"Eugnamptidea" florissantensis* Wickham, 1913 | Flor | P2p | 97 |
| 89 | *E. robusta* Wickham, 1916 | Flor | P2p | 100 |
| 90 | *E. tertiaria* Wickham, 1912 | Flor | P2p | 101 |
| | Attelabidae Billberg, 1820 | | | |
| | Attelabinae Billberg, 1820 | | | |
| | Attelabitae Billberg, 1820 | | | |
| | Euscelini Voss, 1925 | | | |
| | Clinolabina Legalov, 2003 | | | |
| 91 | *Paleoclinolabus dormitus* (Scudder, 1893) | Flor | P2p | 77 |
| | Attelabini Billberg, 1820 | | | |
| 92 | *"Phytonomus" punctatus* Piton, 1940 | Mena | P1sl-t | 99 |
| 93 | *Palaeoalatorostrum schaali* Rheinheimer, 2007 | Mess | P2l | 102 |
| | Brentidae Billberg, 1820 | | | |
| | Apioninae Schoenherr, 1823 | | | |
| | Tanaitae Schoenherr, 1839 | | | |
| | Tanaini Schoenherr, 1839 | | | |
| 94 | *Cretotanaos bontsaganensis* Legalov, 2014 | Tadu | P2i | 103 |
| | Rhadinocybitae Alonso-Zarazaga, 1992 | | | |
| | Rhadinocybini Alonso-Zarazaga, 1992 | | | |
| 95 | *Baltocyba electrinus* Legalov, 2018 | PolJ | P2b | 104 |
| | Notapionini Zimmerman, 1994 | | | |
| 96 | *Archinvolvulus liquidus* Voss, 1972 | ScanJ | P2b | 105 |
| | Palaeotanaitae Kirejtshuk, Legalov et Nel, 2015 | | | |
| | Palaeotanaini Kirejtshuk, Legalov et Nel, 2015 | | | |
| 97 | *Palaeotanaos oisensis* Kirejtshuk, Legalov et Nel, 2015 | OisJ | P2i | 8 |
| | Aspidapiitae Alonso-Zarazaga, 1990 | | | |
| | Aspidapiini Alonso-Zarazaga, 1990 | | | |
| 98 | *Pseudaspidapion khnzoriani* (Zherikhin, 1971) | BalJ | P2b | 79 |
| 99 | *Baltoapion gusakovi* (Legalov, 2015) | BalJ | P2b | 30 |
| 100 | *B. subdiscedens* (Voss, 1953) | BalJ | P2b | 80 |
| | Kalcapiini Alonso-Zarazaga, 1990 | | | |
| 101 | *Melanapion* (*Melanapionoides*) *poinari* Legalov, 2015 | BalJ | P2b | 30 |
| 102 | *M.* (*M.*) *wanati* Legalov, 2012 | BalJ | P2b | 67 |
| 103 | *Succinapion telnovi* Legalov et Bukejs, 2014 | BalJ | P2b | 106 |
| | Apionitae Schoenherr, 1823 | | | |
| | Piezotrachelini Voss, 1959 | | | |
| 104 | *Conapium alleni* Legalov, 2012 | BalJ | P2b | 67 |
| 105 | *Baltoconapium anderseni* (Voss, 1972) | ScanJ | P2b | 105 |
| 106 | *Electrapion kuntzeni* (Wagner, 1924) | BalJ | P2b | 107 |
| | Aplemonini Kissinger, 1968 | | | |
| 107 | *Perapion menatensis* Legalov, Kirejtshuk et Nel, 2017 | Mena | P1sl-t | 50 |
| 108 | *P. rasnitsyni* Legalov, 2018 | GreR | P2i-l | 69 |
| 109 | *"P." profundum* (Schlechtendal, 1894) | Rott | P3h | 89 |
| | Apionini Schoenherr, 1823 | | | |
| | Toxorhynchina Scudder, 1893 | | | |

**Table 2.** *Cont.*

| No. | Taxon | Locality | Age | Sources |
|---|---|---|---|---|
| 110 | *Apionion evestigatum* (Scudder, 1893) | RoaM | P2i-l | 77 |
| 111 | *Toxorhynchus europeoeocenicus* Bukejs et Legalov, 2020 | RovJ | P2b | 108 |
| 112 | *T. michalskii* Legalov, in print. | PolJ | P2b | 109 |
| 113 | "*T.*" *arctus* (Scudder, 1893) | Flor | P2p | 77 |
| 114 | "*T.*" *confectum* (Scudder, 1893) | Flor | P2p | 77 |
| 115 | "*T.*" *corruptus* (Scudder, 1893) | Flor | P2p | 77 |
| 116 | *T. florissantensis* (Wickham, 1916) | Flor | P2p | 100 |
| 117 | *T. minusculus* (Scudder, 1893 | Flor | P2p | 77 |
| 118 | *T. oculatus* (Scudder, 1893 | Flor | P2p | 77 |
| 119 | "*T.*" *pumilum* (Scudder, 1893) | Flor | P2p | 77 |
| 120 | "*T.*" *refrenatum* (Scudder, 1893) | Flor | P2p | 77 |
| 121 | "*T.*" *reventus* (Scudder, 1893) | Flor | P2p | 77 |
| 122 | *T. scudderianum* (Wickham, 1916) | Flor | P2p | 100 |
| | Tribe incertae sedis | | | |
| 123 | "*Apion*" *cockerelli* Wickham, 1911 | Flor | P2p | 76 |
| 124 | "*A.*" *obtusus* (Scudder, 1893) | Flor | P2p | 77 |
| 125 | "*A.*" *levirostre* Foerster, 1891 | Brun | P3r | 87 |
| 126 | "*A.*" *parvum* Foerster, 1891 | Brun | P3r | 87 |
| 127 | "*A.*" *sulcatum* Foerster, 1891 | Brun | P3r | 87 |
| 128 | "*Rhynchites*" *orcinus* Heyden et Heyden, 1866 | Rott | P3h | 85 |
| 129 | "*Apion*" *primordiale* Heyden et Heyden, 1866 | Rott | P3h | 85 |
| 130 | "*A.*" *profundum* Schlechtendal, 1894 | Rott | P3h | 89 |
| 131 | "*Coeliodes*" *primigenius* Oustalet, 1874 | Aix | P3h | 110 |
| | Nanophyinae Gistel, 1856 | | | |
| | Nanophyini Gistel, 1856 | | | |
| 132 | *Palaeonanophyes zherikhini* Legalov, 2015 | GreR | P2i-l | 111 |
| 133 | *Baltonanophyes crassirostre* Legalov, 2018 | PolJ | P2b | 104 |
| 134 | *Zherikhiniellus rarus* Legalov, in print. | BalJ | P2b | 109 |
| 135 | "*Nanophyes*" *japetus* Heyden et Heyden, 1866 | Rott | P3h | 85 |
| | Corimaliini Alonso-Zarazaga, 1989 | | | |
| 136 | "*Corimalia*" *cycloptera* Theobald, 1937 | Cela | P2p | 112 |
| | Brentinae Billberg, 1820 | | | |
| | Trachelizini Lacordaire, 1865 | | | |
| | Stereodermina Sharp, 1895 | | | |
| 137 | *Cerobates* (*Cerobates*) *eocenicus* Legalov et Wappler, in print. | Maar | P2l | 113 |
| | Trachelizina Lacordaire, 1865 | | | |
| 138 | *Eckfelderolispa petrefacta* Legalov et Wappler, in print. | Maar | P2l | 113 |
| 139 | *E. perita* Legalov et Wappler, in print. | Maar | P2l | 113 |
| 140 | *E. manderschieta* Legalov et Wappler, in print. | Maar | P2l | 113 |
| | Curculionidae Latreille, 1802 | | | |
| | Erirhininae Schoenherr, 1825 | | | |
| | Erirhinini Schoenherr, 1825 | | | |
| | Erirhinina Schoenherr, 1825 | | | |
| 141 | *Erirhinites bognorensis* Britton, 1960 | LonC | P2i | 114 |
| 142 | "*Procas*" *vinculatus* Scudder 1893 | RoaM, WhiR | P2i-l, P3r | 77 |
| 143 | "*P.*" *verberatus* Scudder, 1893 | Flor | P2p | 77 |
| 144 | "*Erycus*" *brevicollis* Scudder, 1893 | Flor | P2p | 77 |

**Table 2.** *Cont.*

| No. | Taxon | Locality | Age | Sources |
|---|---|---|---|---|
| | Dorytomini Bedel, 1886 | | | |
| 145 | *Dorytomus bukejsi* Legalov, 2020 | BalJ | P2b | 71 |
| 146 | *D. electrinus* Legalov, 2016 | BalJ | P2b | 90 |
| 147 | *D. groehni* Bukejs et Legalov, 2019 | BalJ | P2b | 116 |
| 148 | *D. korotyaevi* Legalov, 2020 | BalJ | P2b | 71 |
| 149 | *D. nudus* Legalov, 2016 | BalJ | P2b | 115 |
| 150 | *D. vlaskini* Legalov, Nazarenko et Perkovsky, 2019 | RovJ | P2b | 117 |
| 151 | "*D.*" *coercitus* Scudder, 1893 | Flor | P2p | 77 |
| 152 | "*D.*" *vulcanicus* Wickham, 1912 | Flor | P2p | 101 |
| 153 | "*D.*" *williamsi* Scudder, 1893 | Flor | P2p | 77 |
| | Bagoini C.G. Thomson, 1859 | | | |
| 154 | "*Bagous*" *palintonus* Foerster, 1891 | Brun | P3r | 87 |
| | Molytinae Schoenherr, 1823<br>Molytini Schoenherr, 1823<br>Hylobiina W. Kirby, 1837 | | | |
| 155 | *Archaralites zherichini* Legalov, 2010 | Arkh | P1d | 118 |
| 156 | *Furhylobius troesteri* Legalov, 2015 | Mors | P2i | 30 |
| 157 | *Archaeoheilus packardii* (Scudder, 1893) | GreR | P2i-l | 77 |
| 158 | *A. provectus* (Scudder, 1876) | GreR | P2i-l | 73 |
| 159 | *A. scudderi* Legalov, 2018 | GreR | P2i-l | 69 |
| 160 | *A. ovalis* Legalov, 2018 | GreR | P2i-l | 69 |
| 161 | *A. deleticius* (Scudder, 1893) | WhiR | P3r | 77 |
| 162 | *A. lacoei* (Scudder, 1893) | Flor | P2p | 77 |
| 163 | "*Anisorhynchus*" *offosus* Oustalet, 1870 | Core | P3r | 119 |
| 164 | "*Hylobius*" *antiquus* Heyden et Heyden, 1866 | Rott | P3h | 85 |
| | Plinthini Lacordaire, 1863<br>Leiosomatina Reitter, 1913 | | | |
| 165 | *Leiosoma klebsi* Legalov, 2016 | BalJ | P2b | 115 |
| | Pissodini Gistel, 1856 | | | |
| 166 | *Lithopissodes luschitzensis* Beier, 1952 | Luzi | P3h | 120 |
| | Acicnemidini Lacordaire, 1865 | | | |
| 167 | *Electrotribus henningseni* (Voss, 1972) | ScanJ, PolJ<br>BalJ, PolJ | P2b | 105 |
| 168 | *E. theryi* Hustache, 1942 | | P2b | 121 |
| | =*Paleopissodes weigangae* Ulke, 1947 | | | 122 |
| | =*Anchorthorrhinus incertus* Voss, 1953 | | | 80 |
| | =*Isalcidodes Macellus* Voss, 1953 | BalJ | | 80 |
| 169 | *E. wolfschwenningerae* (Rheinheimer, 2007) | BalJ | P2b | 102 |
| 170 | *E. rarus* Legalov, 2020 | | P2b | 71 |
| | Magdalini Pascoe, 1870 | | | |
| 171 | "*Magdalis*" *sedimentorum* Scudder, 1893 | Flor | P2p | 77 |
| 172 | "*M.*" *moesta* Schlechtendal, 1894 | Rott | P3h | 89 |
| 173 | "*M.*" *deucalionis* (Heyden et Heyden, 1866) | Rott | P3h | 85 |
| 174 | "*M.*" *protogenius* (Heyden et Heyden, 1866) | Rott | P3h | 85 |
| | Cleogonini Gistel, 1856 | | | |
| 175 | *Rhysosternum aeternabile* Scudder, 1893 | Flor | P2p | 77 |
| 176 | *Rh. longirostre* Scudder, 1893 | Flor | P2p | 77 |
| 177 | "*Conotrachelus*" *florissantensis* Wickham, 1912 | Flor | P2p | 102 |

**Table 2.** *Cont.*

| No. | Taxon | Locality | Age | Sources |
|---|---|---|---|---|
| | Sciabregmini Legalov, Kirejtshuk et Nel, 2019 | | | |
| 178 | *Sciabregma rugosa* Scudder, 1893 | RoaM | P2i-l | 77 |
| 179 | *S. squamosa* Legalov, Kirejtshuk et Nel, 2019 | OisJ | P2i | 81 |
| 180 | *S. tenuicornis* Cockerell, 1921 | GreR | P2i-l | 123 |
| | Camptorhinini Lacordaire, 1865 | | | |
| 181 | *Camptorrhinites orarius* Britton, 1960 | LonC | P2i | 114 |
| 182 | *Korystina gracilis* Britton, 1960 | LonC | P2i | 114 |
| | Aedemonini Faust, 1898 | | | |
| 183 | *Electrorhinus friedhelmi* Legalov, 2020 | BalJ | P2b | 71 |
| | Cryptorhynchini Schoenherr, 1825 Cryptorhynchina Schoenherr, 1825 | | | |
| 184 | *Taylorius litoralis* Britton, 1960 | LonC | P2i | 114 |
| 185 | "*Cryptorhynchus*" *annosus* Scudder, 1876 | GreR | P2i-l | 73 |
| 186 | "*C.*" *coloradensis* Wickham, 1912 | Flor | P2p | 102 |
| 187 | "*C.*" *evinctus* Scudder, 1893 | Flor | P2p | 77 |
| 188 | "*C.*" *falli* Wickham, 1912 | Flor | P2p | 102 |
| 189 | "*C.*" *kerri* Scudder, 1893 | Flor | P2p | 77 |
| 190 | "*C.*" *profusus* Scudder, 1893 | Flor | P2p | 77 |
| 191 | *Oisecalles latosquamosus* Legalov, Kirejtshuk et Nel, 2019 | OisJ | P2i | 81 |
| 192 | *Cryptorrhynchites sculpturatus* Haupt, 1950 | Geis | P2l | 124 |
| 193 | *Succinacalles uniqus* Zherikhin, 1971 | BalJ | P2b | 79 |
| | Tylodina Lacordaire, 1865 | | | |
| 194 | *Baltacalles triumurbium* Bukejs, Alekseev et Legalov, 2020 | BalJ | P2b | 125 |
| 195 | "*Acalles*" *icarus* Heyden et Heyden, 1866 | Rott | P3h | 85 |
| | Tribe incertae sedis | | | |
| 196 | *Lutago fetosus* Britton, 1960 | LonC | P2i | 114 |
| 197 | *L. nanus* Britton, 1960 | LonC | P2i | 114 |
| 198 | *Pissodites argillosus* Britton, 1960 | LonC | P2i | 114 |
| 199 | "*Chalcodermus*" *kirschi* Deichmueller, 1881 | Kuts | P2b | 126 |
| 200 | *Laccopygus nilesii* Scudder, 1893 | Flor | P2p | 77 |
| 201 | *L. rhenanus* Meunier, 1924 | Rott | P3h | 127 |
| 202 | "*Acalles*" *exhumatus* Wickham, 1913 | Flor | P2p | 97 |
| 203 | "*Pissodes*" *planatus* Foerster, 1891 | Brun | P3r | 87 |
| 204 | "*Hylobius*" *deletus* Oustalet, 1870 | Core | P3r | 119 |
| 205 | "*Plinthus*" *redivivus* Oustalet, 1870 | Core | P3r | 119 |
| 206 | "*Hylobius*" *morosus* Oustalet, 1874 | Core | P3r | 110 |
| 207 | "*Molytes*" *hassencampi* Heyden, 1858 | Sieb | P3 | 128 |
| 208 | "*Pissodes*" *effossus* Heyden, 1858 | Sieb | P3 | 128 |
| 209 | "*Hylobius*" *carbo* Oustalet, 1874 | Aix | P3h | 110 |
| 210 | "*Plinthus*" *heerii* Oustalet, 1874 | Aix | P3h | 110 |
| | Lixinae Schoenherr, 1823 Lixini Schoenherr, 1823 | | | |
| 212 | *Lixus ligniticus* Piton, 1940 | Mena | P1sl-t | 99 |
| 213 | "*Larinus*" *longirostris* Foerster, 1891 | Brun | P3r | 87 |
| 214 | "*L.*" *bronni* Heyden et Heyden, 1866 | Rott | P3h | 85 |
| | Cleonini Schoenherr, 1826 | | | |

**Table 2.** *Cont.*

| No. | Taxon | Locality | Age | Sources |
|---|---|---|---|---|
| 215 | *Eocleonus subjectus* Scudder, 1893 | Flor | P2p | 77 |
| 216 | *"Cleonus" degenerates* Scudder, 1893 | Flor | P2p | 77 |
| 217 | *"C." estriatus* Wickham, 1912 | Flor | P2p | 102 |
| 218 | *"C." exterraneus* Scudder, 1893 | Flor | P2p | 77 |
| 219 | *"C." foersteri* Scudder, 1893 | Flor | P2p | 77 |
| 220 | *"C." primoris* Scudder, 1893 | Flor | P2p | 77 |
| 221 | *"C." rohweri* Wickham, 1911 | Flor | P2p | 76 |
| 222 | *"C." arvernensis* Oustalet, 1870 | Core | P3r | 119 |
| 223 | *"C." fouilhouxii* Oustalet, 1870 | Core | P3r | 119 |
| | Dryophthorinae Schoenherr, 1825 Stromboscerini Lacordaire, 1865 | | | |
| 224 | *Palaeodexipeus kirejtshuki* Legalov, 2016 | BalJ | P2b | 90 |
| 225 | *Rovnoslonik damzeni* Legalov, Nazarenko et Perkovsky, 2019 | RovJ | P2b | 117 |
| 226 | *Stenommatomorphus hexarthrus* Nazarenko, 2009 | RovJ | P2b | 129 |
| | Dryophthorini Schoenherr, 1825 | | | |
| 227 | *Rhinoporkus gratiosus* Legalov, Kirejtshuk et Nel, 2019 | OisJ | P2i | 81 |
| 228 | *Lithophthorus rugosicollis* Scudder, 1893 | Flor | P2p | 77 |
| 229 | *Spodotribus terrulentus* Scudder, 1893 | Flor | P2p | 77 |
| 230 | *Dryophthorus incertus* (Theobald, 1935) | Aix | P3h | 130 |
| | Sphenophorini Lacordaire, 1865 Sphenophorina Lacordaire, 1865 | | | |
| 231 | *"Scyphophorus" fossionis* Scudder, 1893 | Flor | P2p | 77 |
| 232 | *S. laevis* Scudder, 1893 | Flor | P2p | 77 |
| 233 | *"S." tertiarius* Wickham, 1911 | Flor | P2p | 76 |
| 234 | *Oryctorhinus tenuirostris* Scudder, 1893 | Flor | P2p | 77 |
| 235 | *"Sphenophorus" proluviosus* Heyden et Heyden, 1866 | Rott | P3h | 85 |
| | Tribe incertae sedis | | | |
| 236 | *Hipporhinops sternbergi* Cockerell, 1926 | Flor | P2p | 131 |
| | Cossoninae Schoenherr, 1825 Dryotribini LeConte, 1876 | | | |
| 237 | *Ampharthropelma decipiens* Voss, 1972 | ScanJ | P2b | 105 |
| 238 | *Caulophilus Martynovae* Legalov, Nazarenko et Perkovsky, 2019 | RovJ | P2b | 117 |
| 239 | *C. rarus* Legalov, 2016 | BalJ | P2b | 115 |
| 240 | *C. squamosus* Legalov, 2016 | BalJ | P2b | 115 |
| 241 | *C. sucinopunctatus* (Kuska, 1992) | BalJ | P2b | 132 |
| 242 | *C. zherikhini* Nazarenko, Legalov et Perkovsky, 2011 | RovJ | P2b | 133 |
| 243 | *Necrodryophthorus inquilinus* Voss, 1953 | BalJ | P2b | 80 |
| 244 | *Synommatodes patruelis* (Voss, 1953) | BalJ | P2b | 80 |
| 245 | *Electrocossonus kirejtshuki* Legalov, 2020 | BalJ | P2b | 71 |
| | Rhyncolini Gistel, 1856 Rhyncolina Gistel, 1856 | | | |
| 246 | *Rhyncolus sitonifrons* Zherikhin, 1992 | Rott | P3h | 98 |
| | Cossonini Schoenherr, 1825 | | | |
| 247 | *"Cossonus" devoratus* Cockerell, 1925 | Sunc | P1d | 134 |
| 248 | *"C." rutus* Scudder, 1893 | RoaM | P2i-l | 77 |
| 249 | *"C." gabbii* Scudder, 1893 | Flor | P2p | 77 |
| 250 | *C. robustus* Meunier, 1916 | Aix | P3h | 135 |

**Table 2.** *Cont.*

| No. | Taxon | Locality | Age | Sources |
|---|---|---|---|---|
| | Conoderinae Schoenherr, 1833 | | | |
| | Bariditae Schoenherr, 1836 | | | |
| | Apostasimerini Schoenherr, 1844 | | | |
| | Coelonertina Casey, 1922 | | | |
| 251 | *Geraeus diruptus* (Scudder, 1893) | GreR | P2i-l | 77 |
| 252 | *G. anvilis* Legalov, 2018 | GreR | P2i-l | 70 |
| 253 | *G. fossilis* Legalov, 2018 | GreR | P2i-l | 70 |
| 254 | *G. antediluviana* (Wickham, 1916) | Flor | P2p | 100 |
| 255 | *G. hoveyi* (Wickham, 1912) | Flor | P2p | 101 |
| 256 | *G. hypogaeus* (Wickham, 1916) | Flor | P2p | 100 |
| 257 | *G. matura* (Scudder, 1893) | Flor | P2p | 77 |
| 258 | *G. obnuptus* (Scudder, 1893) | Flor | P2p | 77 |
| 259 | *G. schucherti* (Wickham, 1912) | Flor | P2p | 101 |
| 260 | *G. vulcanicus* (Wickham, 1913) | Flor | P2p | 97 |
| 261 | *Lithogeraeus greenriverensis* Legalov, 2018 | GreR | P2i-l | 70 |
| 262 | *L. circumscripta* (Scudder, 1893) | RoaM | P2i-l | 77 |
| 263 | *L. ancilla* (Scudder, 1893) | RoaM | P2i-l | 77 |
| 264 | *L. comminute* (Scudder, 1893) | WhiR | P3r | 77 |
| 265 | *L. damnata* (Scudder, 1893) | Flor | P2p | 77 |
| 266 | *L. cremastorhynchoides* (Wickham, 1913) | Flor | P2p | 97 |
| 267 | *L. florissantensis* (Wickham, 1913) | Flor | P2p | 97 |
| 268 | *L. nearctica* (Wickham, 1916) | Flor | P2p | 100 |
| 269 | *L. primalis* (Wickham, 1917) | Flor | P2p | 136 |
| 270 | *L. renovata* (Wickham, 1916) | Flor | P2p | 100 |
| 271 | *Nicentrus curvirostris* Legalov, 2018 | GreR | P2i-l | 70 |
| 272 | *Steganus barrandei* Scudder, 1893 | RoaM | P2i-l | 77 |
| 273 | *Miogeraeus recurrens* Wickham, 1916 | Flor | P2p | 100 |
| 274 | *"Pachybaris" rudis* Wickham, 1912 | Flor | P2p | 96 |
| | Baridini Schoenherr, 1836 | | | |
| | Baridina Schoenherr, 1836 | | | |
| 275 | *"Baris" palaeophilus* Cockerell, 1920 | Bour | P2l | 137 |
| 276 | *"B." divisa* Scudder, 1893 | Flor | P2p | 77 |
| 277 | *"B." harlani* Scudder, 1893 | Flor | P2p | 77 |
| 278 | *"B." imperfecta* Scudder, 1893 | Flor | P2p | 77 |
| 279 | *"B." naviculare* (Foerster, 1891) | Brun | P3r | 87 |
| 280 | *Catobaris coenosa* Scudder, 1893 | Flor | P2p | 77 |
| | Eurhinina Lacordaire, 1865 | | | |
| 281 | *Eurhinus occultus* Scudder, 1876 | Flor | P2p | 77 |
| | Tribe incertae sedis | | | |
| 282 | *"Ceutorhynchus" clausus* Scudder, 1893 | Flor | P2p | 77 |
| | Conoderintae Schoenherr, 1833 | | | |
| | Conoderini Schoenherr, 1833 | | | |
| 283 | *Jantarhinus compressus* Legalov, Kirejtshuk et Nel, 2019 | OisJ | P2i | 81 |
| | Palaeomallerini Legalov, 2018 | | | |
| 284 | *Palaeomallerus longirostris* Legalov, 2018 | GreR | P2i-l | 70 |
| | Ceutorhynchitae Gistel, 1848 | | | |
| | Ceutorhynchini Gistel, 1848 | | | |

**Table 2.** *Cont.*

| No. | Taxon | Locality | Age | Sources |
|---|---|---|---|---|
| 285 | *"Ceutorhynchus" degravatus* Scudder, 1893 | RoaM | P2i-l | 77 |
| 286 | *"C." eocenicus* Cockerell, 1920 | Peck | P2i | 137 |
| 287 | *C. alekseevi* Legalov, 2016 | BalJ | P2b | 115 |
| 288 | *C. electrinus* Legalov, 2016 | BalJ | P2b | 115 |
| 289 | *C. succinus* Legalov, 2013 | BalJ | P2b | 47 |
| 290 | *"C." blaisdelli* Wickham, 1916 | Flor | P2p | 100 |
| 291 | *"C." compactus* Scudder, 1893 | Flor | P2p | 77 |
| 292 | *"C." duratus* Scudder, 1893 | Flor | P2p | 77 |
| 293 | *"C." irvingi* (Scudder, 1893) | Flor | P2p | 77 |
| 294 | *"C." fischeri* Foerster, 1891 | Brun | P3r | 87 |
| 295 | *"C." crassirostris* Foerster, 1891 | Brun | P3r | 87 |
| 296 | *"C." obliquus* Foerster, 1891 | Brun | P3r | 87 |
| 297 | *"C." funeratus* Heyden et Heyden, 1866 | Rott | P3h | 85 |
| 298 | *Baltocoeliodes sontagae* Legalov et Bukejs, 2018 | BalJ | P2b | 138 |

Cnemogonini Colonnelli, 1979

| No. | Taxon | Locality | Age | Sources |
|---|---|---|---|---|
| 299 | *"Coeliodes" primotinus* Scudder, 1893 | Flor | P2p | 77 |

Phytobiini Gistel, 1848

| No. | Taxon | Locality | Age | Sources |
|---|---|---|---|---|
| 300 | *"Ceuthorrhynchus" miegi* Theobald, 1937 | Klei | P3 | 112 |

Curculioninae Latreille, 1802
Acalyptini C.G. Thomson, 1859

| No. | Taxon | Locality | Age | Sources |
|---|---|---|---|---|
| 301 | *Jantaronosik nebulosus* Legalov, Kirejtshuk et Nel, 2019 | OisJ | P2i | 81 |

Ellescini C.G. Thomson, 1859

| No. | Taxon | Locality | Age | Sources |
|---|---|---|---|---|
| 302 | *Succinostyphlus mroczkowskii* Kuska, 1996 | BalJ | P2b | 139 |
|  | =*Electrotribus erectosquamata* Rheinheimer, 2007 |  |  | 122 |
| 303 | *Pachytychius eocenicus* Legalov, 2016 | BalJ | P2b | 115 |

Palaeoanoplini Legalov, in print.

| No. | Taxon | Locality | Age | Sources |
|---|---|---|---|---|
| 304 | *Palaeoanoplus horridus* Legalov, in print. | BalJ | P2b | 140 |

Smicronychini Seidlitz, 1891

| No. | Taxon | Locality | Age | Sources |
|---|---|---|---|---|
| 305 | *"Smicronyx" antiquus* Foerster, 1891 | Brun | P3r | 87 |

Curculionini Latreille, 1802
Erganiina Pelsue et O'Brien, 2011

| No. | Taxon | Locality | Age | Sources |
|---|---|---|---|---|
| 306 | *Pseudoergania perkovskyi* Legalov, 2019 | BalJ | P2b | 141 |

Timolina Pelsue et O'Brien, 2011

| No. | Taxon | Locality | Age | Sources |
|---|---|---|---|---|
| 307 | *Baltocurculio Manukyani* Legalov, 2020 | BalJ | P2b | 71 |

Curculionina Latreille, 1802

| No. | Taxon | Locality | Age | Sources |
|---|---|---|---|---|
| 308 | *Menatorhis elegans* (Piton, 1940) | Mena | P1sl- | 99 |
| 309 | *Curculio havighorstensis* Zherikhin, 1995 | Havi | tP2i | 142 |
| 310 | *C. anicularis* (Scudder, 1893) | Flor | P2p | 77 |
| 311 | *C. curvirostris* (Scudder, 1893) | Flor | P2p | 77 |
| 312 | *C. duttoni* (Scudder, 1893) | Flor | P2p | 77 |
| 313 | *C. extinctus* (Wickham, 1912) | Flor | P2p | 102 |
| 314 | *C. femoratus* (Scudder, 1893) | Flor | P2p | 77 |
| 315 | *C. flexirostris* (Scudder, 1893) | Flor | P2p | 77 |
| 316 | *C. florissantensis* (Wickham, 1913) | Flor | P2p | 97 |
| 317 | *C. minusculoides* (Wickham, 1911) | Flor | P2p | 76 |
| 318 | *C. minusculus* (Scudder, 1893) | Flor | P2p | 77 |
| 319 | *C. restrictus* (Scudder, 1893) | Flor | P2p | 77 |
| 320 | *"C." beeklyi* (Cockerell, 1918) | Flor | P2p | 143 |
| 321 | *Numitor claviger* Scudder, 1893 | Flor | P2p | 77 |

**Table 2.** *Cont.*

| No. | Taxon | Locality | Age | Sources |
|---|---|---|---|---|
| | Anthonomini C.G. Thomson, 1859 | | | |
| 322 | "*Anthonomus*" *sunchalensis* Cockerell, 1925 | Sunc | P1d | 134 |
| 323 | "*A.*" *soporus* Scudder, 1890 | GreR, RoaM, WhiR | P2i-l | 78 |
| | | GreR | P3r | |
| | | Flor | | |
| 324 | "*A.*" *revictus* Scudder, 1893 | Flor | P2i-l | 77 |
| 325 | "*A.*" *concussus* Scudder, 1893 | Flor | P2p | 77 |
| 326 | "*A.*" *debilatus* Scudder, 1893 | Flor | P2p | 77 |
| 327 | "*A.*" *defossus* Scudder, 1893 | Flor | P2p | 77 |
| 328 | "*A.*" *evigilatus* Scudder, 1893 | Flor | P2p | 77 |
| 329 | "*A.*" *primordius* Scudder, 1893 | Flor | P2p | 77 |
| 330 | "*A.*" *rohweri* Wickham, 1912 | Flor | P2p | 96 |
| 331 | *Coccotorus principalis* Scudder, 1893 | Flor | P2p | 77 |
| 332 | "*C.*" *requiescens* Scudder, 1893 | Flor | P2p | 77 |
| 333 | *Cremastorhynchus stabilis* Scudder, 1893 | | P2p | 77 |
| 334 | *Smicrorhynchus Macgeei* Scudder, 1893 | | P2p | 77 |
| | Eugnomini Lacordaire, 1863 | | | |
| 335 | *Archaeoeugnomus balticus* Legalov, 2016 | BalJ | P2b | 115 |
| 336 | *Anthonoeugnomus barsevskisi* Legalov, 2020 | BalJ | P2b | 71 |
| 337 | *Mazurieugnomus pilosus* Legalov, 2020 | BalJ | P2b | 71 |
| 338 | *Groehnius electrum* Bukejs et Legalov, 2019 | BalJ | P2b | 144 |
| 339 | *G. parvum* Legalov, 2020 | BalJ | P2b | 71 |
| | Rhamphini Rafinesque, 1815 Palaeorhamphina Legalov, 2016 | | | |
| 340 | *Palaeorhamphus damzeni* Legalov, 2020 | BalJ | P2b | 71 |
| 341 | *P. eichmanni* Legalov, 2020 | BalJ | P2b | 71 |
| 342 | *P. primitivus* Legalov, 2016 | BalJ | P2b | 115 |
| | Rhamphina Rafinesque, 1815 | | | |
| 343 | *Orchestes tatjanae* Legalov, 2016 | BalJ | P2b | 115 |
| 344 | "*O.*" *languidulus* Scudder, 1893 | Flor | P2p | 77 |
| 345 | *Tachyerges hyperoche* Legalov et Poinar, 2020 | PolJ | P2b | 91 |
| | Tychiini C.G. Thomson, 1859 Tychiina C.G. Thomson, 1859 | | | |
| 346 | *Eocenesibinia prussica* Legalov, 2016 | BalJ | P2b | 115 |
| 347 | "*Sibinia*" *whitneyi* (Scudder, 1893) | Flor | P2p | 77 |
| 348 | "*S.*" *melancholicus* (Oustalet, 1874) | Aix | P3h | 110 |
| 349 | "*Tychius*" *evolatus* Scudder, 1893 | Flor | P2p | 77 |
| 350 | "*T.*" *ferox* Wickham, 1917 | Flor | P2p | 136 |
| 351 | "*T.*" *secretus* Scudder, 1893 | Flor | P2p | 77 |
| 352 | "*T.*" *latus* Foerster, 1891 | Brun | P3r | 87 |
| 353 | "*T.*" *manderstjernai* Heyden et Heyden, 1866 | Rott | P3h | 85 |
| 354 | *Macrorhoptus intutus* Scudder, 1893 | Flor | P2p | 77 |
| | Camarotini Schoenherr, 1833 Prionomerina Lacordaire, 1863 | | | |
| 355 | *Paleodontopus smirnovae* Legalov, 2020 | BalJ | P2b | 71 |
| 356 | *Masteutes rupis* Scudder, 1893 | Flor | P2p | 77 |
| | Curculioninae incertae sedis | | | |
| 357 | "*Centrinus*" *longipes* Piton, 1940 | Mena | P1sl-t | 99 |
| 358 | "*Gymnetron*" *lecontei* Scudder, 1878 | GreR | P2i-l | 74 |
| 359 | "*G.*" *antecurrens* Scudder, 1893 | Flor RovJ | P2p | 77 |
| 360 | "*Protoceletes*" *hirtus* Nazarenko et Perkovsky, 2016 | | P2b | 145 |

**Table 2.** *Cont.*

| No. | Taxon | Locality | Age | Sources |
|---|---|---|---|---|
| | Cyclominae Schoenherr, 1826 | | | |
| | Listroderini LeConte, 1876 | | | |
| | Palaechthina Brinck, 1948 | | | |
| 361 | *"Listronotus" muratus* Scudder, 1890 | GreR | P2i-l | 78 |
| | Listroderina LeConte, 1876 | | | |
| 362 | *"Listroderes" differens* (Wickham, 1912) | Flor | P2p | 102 |
| 363 | *"L." eviscerates* (Scudder, 1893) | Flor | P2p | 77 |
| | Entiminae Schoenherr, 1823 | | | |
| | Entimintae Schoenherr, 1823 | | | |
| | Tropiphorini Marseul, 1863 | | | |
| 364 | *Primocentron wickhami* Legalov, 2018 | GreR | P2i-l | 69 |
| 365 | *Limalophus poinari* Legalov, 2020 | PolJ | P2b | 71 |
| 366 | *Scuccinalophus attenboroughi* Legalov, 2016 | BalJ | P2b | 90 |
| 367 | *Limalophus compositus* Scudder, 1893 | GreR, WhiR | P2i-l, | 77 |
| | | GreR | P3r | |
| 368 | *L. contractus* Scudder, 1893 | WhiR | P2i-l | 77 |
| 369 | *"L." receptus* Scudder, 1893 | WhiR | P3r | 77 |
| 370 | *"Coniatus" refractus* Scudder, 1893 | RoaM | P3r | 77 |
| 371 | *"Cryptorhynchus" durus* Scudder, 1893 | GreR | P2i-l | 77 |
| 372 | *"Lepyrus" evictus* Scudder, 1893 | GreR, RoaM | P2 i-l | 77 |
| 373 | *"Sitona" paginarum* Scudder, 1893 | RoaM | P2 i-l | 77 |
| 374 | | RoaM GreR | | |
| 375 | *"Otiorhynchus" subteractus* Scudder, 1893 | RoaM | P2 i-l | 77 |
| 376 | *"Ophryastes" grandis* Scudder, 1893 | Flor | P2 i-l | 77 |
| 377 | *"O." compactus* Scudder, 1878 | Flor | P2 i-l | 77 |
| 378 | *"O." petrarum* Scudder, 1893 | Flor | P2 i-l | 77 |
| 379 | *"O." championi* Wickham, 1912 | Flor | P2p | 102 |
| 380 | *Geralophus antiquarius* Scudder, 1893 | Flor | P2p | 77 |
| 381 | *G. fossicius* Scudder, 1893 | Flor | P2p | 77 |
| 382 | *G. lassatus* Scudder, 1893 | Flor | P2p | 77 |
| 383 | *G. occultus* Scudder, 1893 | Flor | P2p | 77 |
| 384 | *G. pumiceus* Scudder, 1893 | Flor | P2p | 77 |
| | *G. repositus* Scudder, 1893 | Flor | P2p | 77 |
| 385 | *G. retritus* Scudder, 1893 | Flor | P2p | 77 |
| 386 | *G. saxuosus* Scudder, 1893 | Flor | P2p | 77 |
| 387 | *G. scudderi* Wickham, 1911 | Aix | P2p | 76 |
| 388 | *Centron moricollis* Scudder, 1893 | | P2p | 77 |
| 389 | *Tenillus firmus* Scudder, 1893 | | P2p | 77 |
| 390 | *Rhytideres sexsulcatus* (Heer, 1856) | Aix | P3h | 146 |
| | =*Phytonomus annosus* Oustalet, 1874 | | | 110 |
| | =*Hipporhinus reynesii* Oustalet, 1874 | | | 110 |
| 391 | *"Rh." heeri* (Germar, 1849) | | P3h | 147 |
| | =*Hipporhinus Matheroni* Nicolas, 1891 | | | 148 |
| | =*Hipporhinus similis* Nicolas, 1891 | | | 148 |
| | =*Hipporhinus intermedius* Nicolas, 1891 | Aix | | 148 |
| | =*Hipporhinus Marioni* Nicolas, 1891 | Aix | | 148 |
| | =*Hipporhinus pertonii* Nicolas, 1891 | | | 148 |
| 392 | *"Rh." schaumi* (Heer, 1856) | | P3h | 146 |
| 393 | *"Hipporhinus" brevis* Giebel, 1856 | | P3h | 149 |
| | Entimini Schoenherr, 1823 | | | |
| | Entimina Schoenherr, 1823 | | | |
| 394 | *Entimus primordialis* Scudder, 1876 | GreR | P2i-l | 73 |

**Table 2.** *Cont.*

| No. | Taxon | Locality | Age | Sources |
|---|---|---|---|---|
| | Eudiagogini LeConte, 1874 | | | |
| 395 | *Eudiagogus vossi* Legalov, 2018 | GreR | P2i-l | 69 |
| 396 | *Tolstonosik oisensis* Legalov, Kirejtshuk et Nel, 2019 | OisJ | P2i | 81 |
| 397 | *Oligocryptus sectus* (Scudder, 1893) | Flor | P2p | 77 |
| 398 | *Eudomus pinguis* Scudder, 1893 | Flor | P2p | 77 |
| 399 | *E. robustus* Scudder, 1893 | Flor | P2p | 77 |
| | Hyperitae Lacordaire, 1863 Hyperini Lacordaire, 1863 Cepurina Capiomont, 1867 | | | |
| 400 | *Palaeophelypera kuscheli* Legalov, 2013 | BalJ | P2b | 47 |
| 401 | *"Geralophus" discessus* Scudder, 1893 | Flor | P2p | 77 |
| 402 | *Hyperites nadezhkini* Zherikhin, 1989 | Biam | P2p | 150 |
| | Otiorhynchitae Schoenherr, 1826 Hormorini Horn, 1876 | | | |
| 403 | *"Hormorus" saxorum* Scudder, 1893 | Flor | P2p | 77 |
| | Cyphiceritae Lacordaire, 1863 Sciaphilini Sharp, 1891 | | | |
| 404 | *"Mitostylus" obdurefactus* (Scudder, 1893) | RoaM | P2i-l | 77 |
| 405 | *"M." seculorum* (Scudder, 1890) | GreR | P2i-l | 78 |
| 406 | *"M." abacus* (Scudder, 1893) | WhiR | P3r | 77 |
| | Trachyphloeini Lacordaire, 1863 Pseudocneorrhinina Kono, 1930 | | | |
| 407 | *Archaeocallirhopalus alekseevi* Legalov et Bukejs, 2015 | BalJ | P2b | 151 |
| 408 | *A. larssoni* Legalov, 2013 | BalJ | P2b | 47 |
| | Polydrusitae Schoenherr, 1823 Sitonini Gistel, 1856 | | | |
| 409 | *Sitonitellus egregius* (Haupt, 1956) | Geis | P2l | 152 |
| 410 | *Sitona lata* Theobald, 1937 | Klei | P3 | 112 |
| 411 | *"S." venustulus* Heyden et Heyden, 1866 | Rott | P3h | 85 |
| | Anypotactini Champion, 1911 | | | |
| 412 | *Paonaupactus gracilis* Legalov, Nazarenko et Perkovsky, 2019 | RovJ | P2b | 117 |
| 413 | *P. katyae* Legalov, Nazarenko and Perkovsky, 2019 | RovJ | P2b | 117 |
| 414 | *P. microphthalmus* (Zherikhin, 1971) | BalJ | P2b | 79 |
| 415 | *P. sitonitoides* Voss, 1953 | BalJ, | P2b | 80 |
| | =*Polydrosus scheelei* Voss, 1953 | ScanJ, | | 80 |
| | =*Pyllobius cephalotes* Voss, 1972 | PolJ | | 105 |
| | =*Otiorhynchus pellucidipes* Voss, 1972 | | | 105 |
| 416 | *P. sobrinus* (Voss, 1972) | ScanJ, Balj | P2b | 105 |
| 417 | *P. viridis* (Wanat et Borowiec, 1986) | BalJ, PolJ | P2b | 153 |
| | Naupactini Gistel, 1856 | | | |
| 418 | *"Hipporhinus" ventricosus* (Piton, 1940) | Mena | P1sl-t | 99 |
| 419 | *Arostropsis groehni* Yunakov et Kirejtshuk, 2011 | BalJ | P2b | 154 |
| 420 | *A. gusakovi* Legalov, 2020 | BalJ | P2b | 71 |
| 421 | *A. perkovskyi* Bukejs et Legalov, 2019 | RovJ | P2b | 155 |
| 422 | *"Cyrtomon" subterraneus* Wickham, 1911 | Flor | P2p | 76 |
| 423 | *"C." florissantensis* Wickham, 1914 | Flor | P2p | 156 |

**Table 2.** *Cont.*

| No. | Taxon | Locality | Age | Sources |
|---|---|---|---|---|
| | Geonemini Gistel, 1856 | | | |
| 424 | "*Epicaerus*" *dilapsus* (Scudder, 1893) | GreR | P2i-l | 77 |
| 425 | "*E.*" *effossus* (Scudder, 1876) | GreR | P2i-l | 73 |
| 426 | "*E.*" *eradicatus* (Scudder, 1893) | WhiR | P2i-l | 77 |
| 427 | "*E.*" *exanimis* (Scudder, 1876) | GreR | P2i-l | 73 |
| 428 | "*E.*" *excissus* (Scudder, 1893) | RoaM | P2i-l | 77 |
| 429 | "*E.*" *fodinarum* (Scudder, 1893) | GreR | P2i-l | 77 |
| 430 | "*E.*" *saxatilis* (Scudder, 1876) | GreR | P2i-l | 73 |
| 431 | "*E.*" *subterraneus* (Scudder, 1893) | GreR | P2i-l | 77 |
| 432 | "*E.*" *terrosus* (Scudder, 1878) | RoaM, WhiR | P2i-l, P3r | 74 |
| 433 | "*E.*" *evigoratus* (Scudder, 1893) | WhiR | P3r | 77 |
| 434 | *Evopes veneratus* Scudder, 1893 | Flor | P2p | 77 |
| 435 | *E. occubatus* Scudder, 1893 | Flor | P2p | 77 |
| 436 | "*Lachnopus*" *recuperatus* Scudder, 1893 | Flor | P2p | 77 |
| 437 | "*Omileus*" *evanidus* Scudder, 1893 | Flor | P2p | 77 |
| | Psallidiini Lacordaire, 1863 | | | |
| 438 | *Trigonoscuta inventa* Scudder, 1893 | Flor | P2p | 77 |
| | Eustylini Lacordaire, 1863 | | | |
| 439 | *Pseudophaops perditus* (Scudder, 1876) | GreR | P2i-l | 73 |
| | Polydrusini Schoenherr, 1823 | | | |
| 440 | *Polydrusus archetypus* Zherikhin, 1971 | BalJ | P2b | 79 |
| 441 | *P. zherikhini* Legalov, 2020 | BalJ | P2b | 71 |
| 442 | *Archaeosciaphilus Marshalli* Legalov, 2012 | BalJ | P2b | 82 |
| | Brachyderini Schoenherr, 1826 | | | |
| 443 | *Palaeocrassirhinus messelensis* Rheinheimer, 2007 | Mess | P2l | 102 |
| 444 | *P. rugosithorax* Rheinheimer, 2007 | Mess | P2l | 102 |
| 445 | *Palaeocneorhinus messelensis* Rheinheimer, 2007 | Mess | P2l | 102 |
| 446 | "*Brachyderes*" *rugosus* (Deichmueller, 1881) | Kuts | P2b | 126 |
| 447 | *Brachymycterus curculionoides* Heyden et Heyden, 1866 | Rott | P3h | 85 |
| | =*Eurychirus induratus* Heyden et Heyden, 1866 | | | 85 |
| | =*Rhinocyllus improbus* Heyden et Heyden, 1866 | | | 85 |
| | =*Varus ignotus* Schlechtendal, 1894 | | | 89 |
| | Tanymecini Lacordaire, 1863 Tainophthalmina Desbrochers des Loges, 1873 | | | |
| 448 | *Protainophthalmus asperulus* (Heer, 1856) | Aix | P3h | 146 |
| | =*Brachyderes aquisextanus* Oustalet, 1874 | | | 110 |
| | =*Brachyderes longipes* Oustalet, 1874 | | | 110 |
| | = *Cleonus Marcelli* Oustalet, 1874 | | | 110 |
| 449 | *P. margarum* (Germar, 1849) | Aix | P3h | 147 |
| | =*Sitona antiqua* Giebel, 1856 | | | 149 |
| 450 | *P. punctulatus* (Nicolas, 1891) | Aix | P3h | 148 |
| 451 | *P. regularis* (Nicolas, 1891) | Aix | P3h | 148 |
| 452 | *P. thaisi* (Nicolas, 1891) | Aix | P3h | 148 |
| 453 | *P. tuberculatus* (Nicolas, 1891) | Aix | P3h | 148 |
| | Pandeleteina Pierce, 1913 | | | |
| 454 | *Pandeleteinus nudus* Wickham, 1917 | Flor | P2p | 138 |
| | Tribe incertae sedis | | | |

**Table 2.** *Cont.*

| No. | Taxon | Locality | Age | Sources |
|---|---|---|---|---|
| 455 | *"Sitona" exitiorum* Scudder, 1893 | Flor | P3r | 77 |
| 456 | *"Strophosomus" marcelini* Theobald, 1937 | Cela | P2p | 112 |
| 457 | *"Sciaphilus" nigrescens* Theobald, 1937 | Cela | P2p | 112 |
| 458 | *"?Argoptochus" incertus* Theobard, 1937 | Cela | P2p | 112 |
| 459 | *"Bagous" atavus* Oustalet, 1870 | Core | P3r | 119 |
| 460 | *"Brachycerus" exilis* Germar, 1837 | Aix | P3h | 157 |
| 461 | *"Phytonomus" firmus* Heer, 1856 | Aix | P3h | 146 |
| | Curculionidae incertae sedis | | | |
| 462 | *Otiorhynchites williamsi* Cockerell, 1943 | Mirs | P1 | 158 |
| 463 | *O. aterrimus* Cockerell, 1925 | Sunc | P1d | 134 |
| 464 | *O. crassus* Cockerell et Wagner, 1936 | Sunc | P1d | 159 |
| 465 | *O. densepunctatus* Haupt, 1956 | Geis | P2l | 152 |
| 466 | *O. fossilis* Scudder, 1893 | GreR | P2i-l | 77 |
| 467 | *O. commutatus* Scudder, 1893 | RoaM | P2i-l | 77 |
| 468 | *O. tysoni* Scudder, 1893 | RoaM | P2i-l | 77 |
| 469 | *O. absentivus* Scudder, 1893 | Flor | P2p | 77 |
| 470 | *O. florissantensis* Wickham, 1911 | Flor | P2p | 76 |
| 471 | *O. wilcoxianus* Wickham, 1929 | Flor | P2p | 160 |
| 472 | *Curculionites taxodii* Heer, 1870 | Star | P1d | 161 |
| 473 | *C. angustior* Cockerell et Wagner, 1936 | Sunc | P1d | 159 |
| 474 | *C. epistictus* Cockerell et Wagner, 1936 | Sunc | P1d | 159 |
| 475 | *C. eustictus* Cockerell et Wagner, 1936 | Sunc | P1d | 159 |
| 476 | *C. harringtoni* Cockerell, 1925 | Sunc | P1d | 134 |
| 477 | *C. jujuensis* Cockerell, 1925 | Sunc | P1d | 134 |
| 478 | *C. latiusculus* Cockerell et Wagner, 1936 | Sunc | P1d | 159 |
| 479 | *C. magdalinus* Cockerell et Wagner, 1936 | Sunc | P1d | 159 |
| 480 | *C. megastictus* Cockerell et Wagner, 1936 | Sunc | P1d | 159 |
| 481 | *C. microstictus* Cockerell et Wagner, 1936 | Sunc | P1d | 159 |
| 482 | *C. parastictus* Cockerell et Wagner, 1936 | Sunc | P1d | 159 |
| 483 | *C. stebingeri* Cockerell, 1926 | Sunc | P1d | 131 |
| 484 | *C. sunchalicus* Cockerell et Wagner, 1936 | Sunc | P1d | 159 |
| 485 | *C. wielandi* Cockerell, 1925 | Sunc | P1d | 134 |
| 486 | *C. marginatus* Giebel, 1856 | Corf | P2l | 149 |
| 487 | *C. punctillatus* Haupt, 1950 | Geis | P2l | 124 |
| 488 | *C. bartonicus* Cockerell, 1920 | Bour | P2l | 162 |
| 489 | *C. brenthiformis* Cockerell, 1920 | Bour | P2l | 162 |
| 490 | *C. optimus* Cockerell, 1920 | Bour | P2l | 162 |
| 491 | *C. ovatus* Oustalet, 1870 | Core | P3r | 119 |
| 492 | *C. morosus* Heer, 1856 | Aix | P3h | 146 |
| 493 | *Calandrites defessus* Scudder, 1893 | RoaM | P2i-l | 77 |
| 494 | *C. cineratius* Scudder, 1893 | GreR,RoaM | P2i-l | 77 |
| 495 | *C. hindsi* Cockerell, 1917 | Flor | P2p | |
| 496 | *C. ursorum* Cockerell, 1918 | Flor | P2p | |
| 497 | *Ophryastites gardneri* Cockerell, 1920 | Bour | P2l | |
| 498 | *O. absconsus* Scudder, 1893 | Flor | P2p | 77 |
| 499 | *O. cinereus* Scudder, 1893 | Flor | P2p | 77 |
| 500 | *O. digressus* Scudder, 1893 | Flor | P2p | 77 |
| 501 | *O. hendersoni* Cockerell, 1917 | Flor | P2p | |
| 502 | *O. miocenus* Wickham, 1912 | Flor | P2p | |
| 503 | *O. dispertitus* Scudder, 1893 | WhiR | P3r | 77 |
| 504 | *O. heribaudi* Piton, 1936 | PuStJ | P3 | |
| 505 | *Mononychites punctipennis* Haupt, 1956 | Geis | P2l | |
| 506 | *M. rotundatus* Haupt, 1956 | Geis | P2l | |
| 507 | *Syntomostylus rudis* Scudder, 1893 | RoaM | P2i-l | |
| 508 | *"S." fortis* Cockerell, 1909 | GreR | P2i-l | |
| 509 | *"Otiorhynchus" flaccus* Scudder, 1893 | RoaM | P2i-l | 77 |

**Table 2.** *Cont.*

| No. | Taxon | Locality | Age | Sources |
|---|---|---|---|---|
| 510 | "*Otiorhynchus*" *tumbae* Scudder, 1890 | GreR | P2i-l | |
| 511 | "*Pachylobius*" *depraedatus* Scudder, 1893 | RoaM | P2i-l | 77 |
| 512 | "*Pachylobius*" *compressus* Scudder, 1893 | GreR,RoaM | P2i-l | 77 |
| 513 | "*Phyllobius*" *antecessor* Scudder, 1893 | RoaM | P2i-l | 77 |
| 514 | "*Rhyssomatus*" *tabescens* Scudder, 1893 | WhiR | P3r | 77 |
| 515 | "*Scythropus*" *somniculosus* Scudder, 1893 | WhiR | P3r | 77 |
| 516 | "*Phyllobius*" *avus* Scudder, 1893 | RoaM | P2i-l | 77 |
| 517 | "*Phyllobius*" *carcerarius* Scudder, 1893 | RoaM | P2i-l | 77 |
| 518 | "*Pachylobius*" *yungi* Piton, 1936 | Gaub | P3 | 164 |
| 519 | "*Tanymecus*" *gautieri* Piton, 1936 | Gaub | P3 | 164 |
| 520 | "*Pachylobius*" *martyi* Piton et Theobald, 1937 | PuStJ | P3 | 166 |
| 521 | "*Lachnopus*" *dilatatus* Theobald, 1937 | Klei | P3 | 166 |
| 522 | "*Cleonus*" *foersteri* Theobald, 1937 | Klei | P3 | 112 |
| 523 | "*Lachnopus*" *humatus* Scudder, 1893 | Flor | P2p | 77 |
| 524 | "*Sphenophorus*" *elegans* Theobald, 1935 | Aix | P3h | 130 |

<table>
<tr><td colspan="5" align="center">Platypodidae Shuckard, 1840<br>Tesserocerinae Strohmeyer, 1914<br>Tesserocerini Strohmeyer, 1914</td></tr>
</table>

| No. | Taxon | Locality | Age | Sources |
|---|---|---|---|---|
| 525 | *Eoplatypus jordali* Peris, Solórzano Kraemer et Cognato, 2017 | BalJ | P2b | 167 |
| 526 | *Cenocephalus aniskini* Legalov, 2020 | BalJ | P2b | 71 |

<table>
<tr><td colspan="5" align="center">Scolytidae Latreille, 1804<br>Hylesininae Erichson, 1836<br>HylastiniLeConte, 1876</td></tr>
</table>

| No. | Taxon | Locality | Age | Sources |
|---|---|---|---|---|
| 527 | *Hylastes aterites* Schedl, 1947 | BalJ | P2b | 168 |
| 528 | "*H.*" *americanus* Wickham, 1913 | Flor | P2b | 97 |
| 529 | *Hylurgops corpulentus* Schedl, 1947 | BalJ | P2b | 168 |
| 530 | *H. dubius* (Hagedorn, 1906) | BalJ | P2b | 169 |
| 531 | *H. electrinus* (Germar, 1813) | BalJ | P2b | 170 |
| 532 | *H. granulatus* (Schedl, 1947) | BalJ | P2b | 168 |
| 533 | *H. pilosellus* Schedl, 1947 | BalJ | P2b | 168 |
| 534 | *H. schellwieni* (Hagedorn, 1906) | BalJ | P2b | 168 |
| 535 | *H. tuberculatus* Schedl, 1947 | BalJ | P2b | 168 |
| 536 | *H. piger* Wickham, 1913 | Flor | P2p | 97 |

<table>
<tr><td colspan="5" align="center">HylesininiErichson, 1836</td></tr>
</table>

| No. | Taxon | Locality | Age | Sources |
|---|---|---|---|---|
| 537 | *Hylesinus extractus* Scudder, 1893 | Flor | P2p | 77 |
| 538 | *H. hydropicus* (Wickham, 1916) | Flor | P2p | 101 |
| 539 | *H. neli* Petrov et Zherikhin, 2004 | Aix | P3h | 171 |
| 540 | "*H.*" *facilis* Heer, 1856 | Aix | P3h | 146 |

<table>
<tr><td colspan="5" align="center">Hylurgini Gistel, 1848</td></tr>
</table>

| No. | Taxon | Locality | Age | Sources |
|---|---|---|---|---|
| 541 | *Xylechinus mozolevskae* Petrov et Perkovsky, 2008 | RovJ | P2b | 172 |
| 542 | *Klesovia pubescens* Petrov et Perkovsky, 2018 | RovJ | P2b | 173 |
| 543 | *Xylechinites anceps* Hagedorn, 1906 | BalJ | P2b | 169 |

<table>
<tr><td colspan="5" align="center">Phloeosinini Nuesslin, 1912</td></tr>
</table>

| No. | Taxon | Locality | Age | Sources |
|---|---|---|---|---|
| 544 | *Phloeosinus assimilis* (Schedl, 1947) | BalJ | P2b | 168 |
| 545 | *Ph. brunni* (Hagedorn, 1906) | BalJ | P2b | 169 |
| 546 | *Ph. regimontanus* (Hagedorn, 1906) | BalJ | P2b | 169 |
| 547 | *Ph. rehi* (Hagedorn, 1906) | BalJ | P2b | 169 |
| 548 | *Ph. robustus* (Schedl, 1947) | BalJ | P2b | 168 |
| 549 | *Ph. sexspinosus* (Schedl, 1947) | BalJ | P2b | 168 |
| 550 | *Ph. tuberculifer* (Schedl, 1947) | BalJ | P2b | 168 |
| 551 | *Ph. wolffi* (Schedl, 1947) | BalJ | P2b | 168 |

**Table 2.** *Cont.*

| No. | Taxon | Locality | Age | Sources |
|---|---|---|---|---|
| | Phloeotribini Chapuis, 1869 | | | |
| 552 | *Phloeotribus zimmermani* Wickham, 1916 | Flor | P2p | 101 |
| | Polygraphini Chapuis, 1869 | | | |
| 553 | *Carphoborus keilbachi* (Schedl, 1947) | BalJ | P2b | 168 |
| 554 | C. *posticus* (Schedl, 1947) | BalJ | P2b | 168 |
| | Scolytinae Latreille, 1804 DryocoetiniLindemann, 1877 | | | |
| 555 | *Dryocoetes diluvialis* (Wickham, 1916) | GreR BalJ | P2i-l P2b | 101 |
| 556 | *Taphramites gnathotrichus* Schedl, 1947 | RovJ | P2b | 168 |
| 557 | *T. rovnoensis* Petrov et Perkovsky, 2008 | BalJ | P2b | 172 |
| 558 | *Taphrorychus immaturatus* Schedl, 1947 | | | 168 |
| | Incertae sedis | | | |
| 559 | *"Dryocoetes" carbonarius* Scudder, 1878 | GreR | P2i-l | 74 |
| 560 | *"Trypodendron" impressus* Scudder, 1876 | GreR | P2i-l | 73 |
| 561 | *"Polygraphus" wortheni* Scudder, 1893 | RoaM | P2i-l | 77 |
| 562 | *Xyleborites longipennis* Wickham, 1913 | Flor | P2p | 97 |
| 563 | *Duartia pulchella* Martins-Neto, 2001 | Fons | P2p | 174 |
| | Curculionoidea incertae sedis | | | |
| 564 | *Thryogenosoma cariniger* (Motschulsky, 1857) | BalJ | P2b | 175 |

Abbreviations for localities and ages are given in Material and methods.

### 3.1. Paleocene Weevil Fauna

The weevil fauna of the Paleocene is the poorest among the Paleogene faunas. This is primarily due to the small number of Paleocene localities, as well as the lack of Paleocene amber with Curculionoidea. Curculionoid beetles were found in five localities, including France, Svalbard (Denmark), south of the Russian Far East, China (Hong Kong) and Argentina (Figure 2). Three of these localities are of early Paleocene age (Arkhara, Starostin and Sunchal) and one of them is late Paleocene (Menat). The age of Mirs Bay from Hong Kong, the Ping Chau Formation (China) is assumed to be Paleocene, however the stage is not specified [176].

Twenty seven species were described from the Paleocene (Figure 3). Additionally, 18 species were known from Starostin, Sunchal and Mirs based on isolated elytra, which are assigned to recent genera or placed in the formal genera *Otiorhynchites* and *Curculionites* that were established for elytra [131,134,158,159,161]. *"Cossonus" devoratus* and *"Anthonomus" sunchalensis* from the early Paleocene of Argentina are not only the earliest findings of the tribes Cossonini and Anthinomini but also the first appearance of the subfamilies Cossoninae and Curculioninae in the fossil record. The only early Paleocene beetle represented by an almost complete impression is *Archaralites zherichini* of the subfamily Molytinae (earliest record of this subfamily) from the Danian of Outer Manchuria.

Eight species represented as complete impressions are known from Selandian-Thanetian of France. The families Ithyceridae, Rhynchitidae, Attelabidae, Brentidae and Curculionidae are recorded. Ithyceridae is represented by one species of the subfamily Chilecarinae. This is the only representative of the family in the Paleocene and the last find in the Eastern Hemisphere. Records of Ithyceridae are common in Cretaceous of the Northern Hemisphere [29,30,48,49,51,177]. One species, the genus of which requires clarification, is from the subfamily Rhynchitinae of the family Rhynchitidae. The families Attelabidae and Brentidae are each represented by one species of the recent tribe Attelabini and the extinct species of the recent genus *Perapion* from the tribe Aplemonini (earliest record, 61.0–59.0 Ma).

Four species belong to the family Curculionidae. One species of the recent genus *Lixus* is from the subfamily Lixinae (earliest record, 61.0–59.0 Ma). Two species of the subfamily Curculioninae, one of which belongs to the extinct genus of the tribe Curculionini (earliest record, 61.0–59.0 Ma), and the systematic position of the second ("*Centrinus*" *longipes*) in the subfamily require clarification. One species, "*Hipporhinus*" *ventricosus* with an unclear generic position, is in the tribe Naupactini from the diverse subfamily Entiminae (earliest record, 61.0–59.0 Ma). Representatives of other families were not found in the Paleocene.

Most of the forms represented in the Paleocene belong to widespread groups (Attelabini, Aplemonini (*Perapion*), Lixini (*Lixus*), Cossonini (*Cossonus*), Curculionini, Anthonomini (*Anthonomus*)), are now found in the recent fauna. Two species show characters typical of the Paleocene fauna of Europe. *Petropsis rostratus* somehow links the fauna of Menat with recent relict fauna of the Chilean-Patagonian and Australian regions, where modern representatives of the tribe Chilecarini live [178,179]. The discovery of species of the tribe Naupactini similar to Neotropical forms provides additional evidence for the faunogenetic relationships of the Paleocene Europe, with Central and South America. The ecological composition of these beetles is homogeneous. Mostly they are forest forms. *Petropsis rostratus*, like the recent Ithyceridae of the subfamily Chilecarinae, could be associated with gymnosperms from the family Cupressaceae [178,179], which were found at the site [99]. "*Phytonomus*" *punctatus* (Attelabini) folded tubes from angiosperm leaves, for example, from oak known from the deposit [99], as recent representatives of this tribe [180]. *Archaralites zherichini* and "*Cossonus*" *devoratus* developed in wood. *Menatorhis elegans*, like modern Curculionini, developed in flower buds or fruits of angiosperms [181], for example, on oak, several species that are known from this locality [99]. "*Anthonomus*" *sunchalensis* could be connected with trees or shrubs. Two species were very probably associated with herbaceous vegetation, the recent species of the genus *Lixus* usually develop on herbaceous plants and, as an exception, with shrubby plants [182]. *Lixus ligniticus* could be associated with *Atriplex* (Amaranthaceae). *Perapion menatensis,* as well as recent representatives of the genus, probably developed on Polygonaceae [183]. Polygonaceae are not known from Menat but pollen of *Polygonum* was recorded in the late Paleocene of France [184]. Curculionoidea of Menat shows the presence of coniferous-deciduous forest and herbaceous near-water vegetation.

*3.2. Eocene Weevil Fauna*

As Many as 441 species of weevil-shaped beetles were described from Eocene deposits (Figure 4), which originated from 24 localities. Undescribed forms are known from Quilchena (Canada) [185], Romanian amber [186], Huitrera Formation (Argentina) [187] and Bitterfeld amber.

3.2.1. Review of Curculionoidea Families in the Eocene

Nemonychidae in the Eocene

Three species of the family Nemonychidae from the extinct subfamily Cretonemonychinae and recent Cimberidinae were found in the Eocene. The subfamily Cretonemonychinae is also known in the early and late Cretaceous [29,30,177].

Anthribidae in the Eocene

Nearly thirty species of Anthribidae were described from the early, middle and late Eocene of America and Europe. The subfamily Anthribinae is most represented. Twenty seven species from eight tribes are known in this subfamily from the Eocene. Twelve species from three genera of the tribes Mecocerini, Allandrini, Oiserhinini and Zygaenodini were described from European amber and 15 species from seven genera of the tribes Cratoparini, Anthribini, Ecelonerini and Tropiderini from North American localities. There are no common genera and tribes between the American and European deposits. The subfamily Choraginae is represented by two species from two tribes. "*Choragus*" *fictilis*

of the tribe Choragini was described from the Eocene of the United States and *Eduardoxenus unicus* of the tribe Valenfriesiini from Rovno amber.

Ithyceridae in the Eocene

Two American genera *Eoceneithycerus* and *Ithyceroides* close to the recent American genus *Ithycerus* were found in North America.

Belidae in the Eocene

Nine species from the subfamily Oxycoryninae belong to the family Belidae. Several species of the tribe Oxycraspedini were found in Baltic amber, one of which was described. The tribe Metrioxenini is found both in American deposits and the middle Eocene amber. The extinct tribe Palaeorhopalotriini, close to the recent Central American tribe Allocorynin, was found in the Eocene of France.

Rhynchitidae in the Eocene

The family Rhynchitidae is found in American and European deposits and is represented by 34 described species. Five species of the genus *Baltocar* of the tribe Sanyrevilleini from the subfamily Sayrevilleinae are known only from Baltic amber. The subfamily Rhynchitinae is represented by species of the tribes Auletini, Rhynchitini and Eugnamptini. Thirteen species from subtribes Auletina, Pseudauletina and Pseudomesauletina were described in the tribe Auletini. Most species (nine) are known from the terminal Eocene of Florissant. Finds of Auletini are rare in the Green River deposits as well as in Baltic and Rovno amber. The tribe Rhynchitini is represented by 12 species, with nine of them recorded in the late Eocene of the United States and their affiliation to the subtribes has not yet been established. Two species from Baltic amber belong to the archaic subtribes Temnocerina and Perrhynchitina. One genus with three species of the tribe Eugnamptini is described from the Florissant beds.

Attelabidae in the Eocene

One or two extinct genera, *Palaeoalatorostrum* of the subtribe Attelabini from the tribe Attelabini and *Paleoclinolabus* of the subtribe Clinolabina from the tribe Euscelini, belonging to the subfamily Attelabinae of the Attelabidae, are known from the middle Eocene of Germany and the Late Eocene of the United States.

Brentidae in the Eocene

The family Brentidae is represented in the Eocene by three subfamilies, the Apioninae, Nanophyinae and Brentinae. Thirty six species belong to this family. The subfamily Apioninae is the most species-rich. The primitive tribes Tanaini, Rhadinocybini, Notapionini and Palaeotanaini each have one monotypic genus in the early Eocene of Europe and Asia, as well as in the end of the middle Eocene of Europe. The supertribes Aspidapiitae and Apionitae include representatives of recent (*Pseudaspidapion*, *Melanapion*, *Conapium*, *Perapion*, *Apionion* and *Toxorhynchus*) and extinct genera (*Baltoapion*, *Succinapion*, *Baltoconapium*, *Electrapion*). A third of the species of the latter groups belong to the extinct genera and two-thirds to the recent ones. Both known tribes (Nanophyini and Corimaliini) of the subfamily Nanophyinae were found in the Eocene. Three extinct genera of the Nanophyini were described from the middle Eocene of the USA and Baltic amber. One species placed in the genus *Corimalia* is known from the middle Eocene of France. The subfamily Brentinae is recorded only in the middle Eocene of Germany, where the extinct genus *Eckfelderolispa* with three species and extinct representative of the recent genus *Cerobates*, belonging to the tribe Trachelizini, were described.

Curculionidae in the Eocene

The Curculionidae is the Main group of Eocene Curculionoidea numbering 290 described species. All known subfamilies (Erirhininae, Molytinae, Lixinae, Dryophthorinae, Cossoninae, Conoderinae, Curculioninae, Cyclominae and Entiminae) are represented in the Eocene (Figure 5).

The most primitive subfamily Erirhininae is represented by species from the tribes Erirhinini and Dorytomini. Several species from different genera of the former tribe were recorded in the early Eocene of England, the middle and the late Eocene of the USA. The genus *Dorytomus* from the Dorytomini includes nine extinct species from Baltic and Rovno amber, as well as species from the terminal Eocene of North America.

The subfamily Molytinae is one of the most diverse groups of Curculionidae [39,188]. The tribes Molytini, Plinthini, Acicnemidini, Magdalini, Cleogonini, Sciabregmini, Camptorhinini, Aedemonini and Cryptorhynchini were found in Eocene deposits. Two extinct genera, *Furhylobius*—with one species from the early Eocene of Denmark—and *Archaeoheilus*—with five species from the early-middle and terminal Eocene of the USA—belong to the tribe Molytini. One species of the recent genus *Leiosoma* of the tribe Plinthini was described from Baltic amber. One extinct genus *Electrotribus* of the tribe Acicnemidini is known from Baltic amber, where its species are one of the most common Curculionidae. In other Eocene localities, neither the tribe Acicnemidini, nor this genus were found. "*Magdalis*" *sedimentorum* of the tribe Magdalini is recorded from the Eocene of the United States. The American tribe Cleogonini is represented in the Eocene of the United States by two species of the extinct genus *Rhysosternum* and one species of the genus *Conotrachelus*. The extinct tribe Sciabregmini with one genus is known from three species, one from the early Eocene of France and two from the early-middle Eocene of North America. Two fossil genera from the early Eocene of England belong to the tribe Camptorhinini. The Afrotropical tribe Aedemonini was recently discovered in Baltic amber. Eleven species from six genera (five of which are extinct) of the tribe Cryptorhynchini are known from the Eocene.

The subfamily Lixinae is represented in the Eocene by only the tribe Cleonini. The extinct genus *Eocleonus* with one species and five species formally placed in the genus *Cleonus* were described from the late Eocene of the United States.

Eleven Eocene species belong to the subfamily Dryophthorinae. Most species (seven) are from the tribes Stromboscerini and Dryophthorini living in the forest litter. The former tribe is noted in Baltic and Rovno amber and the latter is in early Eocene Oise amber and late Eocene Florissant deposits. Four species of the tribe Sphenophorini were described from the late Eocene of the United States.

The subfamily Cossoninae is represented by the tribe Dryotribini in middle Eocene amber and Cossonini in early-middle and late Eocene deposits of the United States.

The diverse subfamily Conoderinae is divided into four supertribes [40], three of which are found in Eocene. Thirty species belong to the supertribe Bariditae of which twenty nine were described from early-middle and late Eocene of the USA and one from the middle Eocene of England. Most North American species belong to the subtribe Coelonertina of the tribe Apostasimerini. They were described both in recent (*Geraeus*, *Pachybaris*, *Nicentrus*) and extinct (*Miogeraeus*, *Lithogeraeus*, *Steganus*) genera. Six species belong to the tribe Baridini. These are four representatives of the recent genus *Baris* and one of the extinct genus *Catobaris* (Baridina), as well as one species of the recent genus *Eurhinus* (Eurhinina). Two monotypic extinct genera of the recent tribe Conoderini and the extinct tribe Palaeomallerini are from the early Eocene of France and the early-middle Eocene of the United States belong to the supertribe Conoderintae. The supertribe Ceutorhynchitae is represented by ten species of the tribe Ceutorhynchini and one species of the tribe Cnemogonini, which was noted in the late Eocene of the United States. Two genera, the recent genus *Ceutorhynchus* Marked from the early to late Eocene of Europe and the USA and extinct *Baltocoeliodes* from Baltic amber, belong to the tribe Ceutorhynchini.

The subfamily Curculioninae is the second largest group by species of Eocene Curculionidae. Nine tribes with 53 species are known from the Eocene. The tribe Acalyptini is represented by an extinct monotypic genus in early Eocene Oise amber. Two species from Baltic amber belong to the

tribe Ellescini. The monotypic tribe Palaeoanoplini is known only from the middle Eocene of Europe. Fifteen species were described in the tribe Curculionini. Representatives of the subtribes Erganiina and Timolina are found only in Baltic amber. The most common group in the modern fauna on all continents is the subtribe Curculionina. In the Eocene, all (without *Curculio havighorstensis*) species of this subtribe were described from the terminal Eocene of the USA and one species from the early Eocene of Germany. The situation is similar with the tribe Anthonomini. All Eocene species of this tribe are known only from North America. The tribe Eugnomini is represented by five species from four genera found in Baltic amber. The extinct subtribe Palaeorhamphina with three species from Baltic amber and the recent subtribe Rhamphina with species of the recent genera *Orchestes* and *Tachyerges* from middle Eocene amber and the terminal Eocene of the USA belong to the tribe Rhamphini. The tribe Tychiini is noted in Baltic amber and the late Eocene of the USA, where it is represented by five species from the recent genera *Sibinia*, *Tychius*, *Macrorhoptus* and the extinct genus *Eocenesibinia*. The tribe Camarotini is noted in the middle Eocene of Europe and the late Eocene of North America.

The subfamily Cyclominae is known only from the Eocene of the USA, where species of the recent genera *Listronotus* and *Listroderes* were described. Ьщыеplace of the numberrous species among the Eocene Curculionidae are in the subfamily Entiminae. Ninety species were described from the tribes Tropiphorini, Entimini, Eudiagogini, Hyperini, Hormorini, Sciaphilini, Trachyphloeini, Sitonini, Anypotactini, Naupactini, Geonemini, Psallidiini, Eustylini, Polydrusini, Brachyderini and Tanymecini. Twenty four species belong to the tribe Tropiphorini. All but one representative of this tribe were found in the Eocene of North America and only one species of the genus *Limalophus* has recently been described from middle Eocene Baltic (Polish) amber. The Neotropical tribe Entimini is known from the early-middle Eocene Green River deposits by one extinct species of the recent genus *Entimus*. The tribe Eudiagogini is now distributed only in the Western Hemisphere [26]. *Tolstonosik oisensis* was found in the early Eocene of France, *Eudiagogus vossi* from the early-middle Eocene, *Oligocryptus sectus* and the genus *Eudomus* with two species from the late Eocene of the United States. Three species of the tribe Hyperini from the subtribe Cepurina have been described since the end of the middle Eocene of Europe, the late Eocene of the USA and the Far East of Russia. One species, "*Hormorus*" *saxorum* Scudder, 1893 from the late Eocene of the USA was described in the American tribe Hormorini. The tribe Sciaphilini with a contemporary centre of diversity in the Western Palaearctic [26] was found only in the early-middle Eocene of North America. The tribe Trachyphloeini is represented by the extinct genus *Archaeocallirhopalus* with two species belonging to the subtribe Pseudocneorrhinina, which is now distributed in East Asia. One species of the extinct genus *Sitonitellus* from the middle Eocene of Germany was assigned to the tribe Sitonini. The Neotropical tribe Anypotactini was found in the middle Eocene of Europe. In Baltic amber, this is one of the most common groups. It is interesting to note that this tribe was not found in the Eocene deposits of North America. The tribe Naupactini is noted in Baltic and Rovno amber and in the late Eocene of the Florissant. Thirteen species of the tribe Geonemini are found in the Eocene of the United States. The tribe Psallidiini is represented by one species of the genus *Trigonoscuta* from the terminal Eocene of North America. The extinct genus *Pseudophaops* of the tribe Eustylini was described from the early-middle Eocene of Green River. Two species of the genus *Polydrusus* and one species of the extinct genus *Archaeosciaphilus* of the tribe Polydrusini were found in Baltic amber. Three species from extinct genera *Palaeocrassirhinus* and *Palaeocneorhinus* from the Lutetian of Germany and one species placed in the genus *Brachyderes* from the Bartonian of the Czech Republic belong to the tribe Brachyderini. Only one species of the tribe Tanymecini was described in the recent genus *Pandeleteinus* from the late Eocene of North America.

Platypodidae in the Eocene

The family Platypodidae, with two species of the tribe Tesserocerini, were described from Baltic and Romanian amber [71,167,186,189].

Scolytidae in the Eocene

Scolytidae is represented by 35 species, 24 of which were described from Baltic and Rovno amber. Twenty six species belong to the subfamily Hylesininae and four to Scolytinae. The systematic position of five species requires clarification. Three genera are extinct, known only from the late Eocene and nine genera are recent. The genera *Hylurgops* and *Phloeosinus* are the richest in species with 8 species in each. The largest number of species (21) of bark beetles was described from Baltic amber.

### 3.2.2. Early Eocene Weevil Fauna

Early Eocene (Ypresian) localities (Figure 6) are in the USA, England (Peckham, London Clay), France (Oise amber), Denmark (Mors), Germany (Havighorst) and the south of the Russian Far East (Tadushi). Twenty one species of Curculionoidea were described from these localities (Figure 7). The families Anthribidae, Ithyceridae, Brentidae, Curculionidae and Scolytidae were recorded for Oise amber [190] from the early Eocene. Unfortunately, the fauna of Oise amber and the London clays include only 7–8 described species, with 1–2 species known (Figure 8) from other localities. The USA contains representatives of the subfamily Ithycerinae now living in North America and has a fundamentally different biota from the rest of the Eocene faunas. Common species and genera intermediate between the early Eocene faunas are absent. Representatives of the tribes Oiserhinini, Palaeotanaini, Sciabregmini, Cryptorhynchini, Dryophthorini, Conoderini, Acalyptini, Eudiagogini are found in Oise amber, Ceutorhynchini in Peckham, Curculionini in Havighorst, Erirhinini, Camptorhinini and Cryptorhynchini in London Clay, Molytini in Mors and Tanaini in Tadushi. Here are found some of the earliest records of representatives of the tribes Sciabregmini (53.0 Ma), Cryptorhynchini (54.0–50.0 Ma), Dryophthorini (53.0 Ma), Eudiagogini (53.0 Ma), Ceutorhynchini and Camptorhinini (54.0–50.0 Ma) and the latest find of the tribe Tanaini in the fossil record. The tribes Oiserhinini, Palaeotanaini, Conoderini and Acalyptini are noted only in the early Eocene. The Oise amber fauna is the most diverse in taxonomic composition and includes four families Anthribidae, Brentidae, Curculionidae and Scolytidae. Only Curculionidae were described from Peckham, London Clay, Havighorst and Mors.

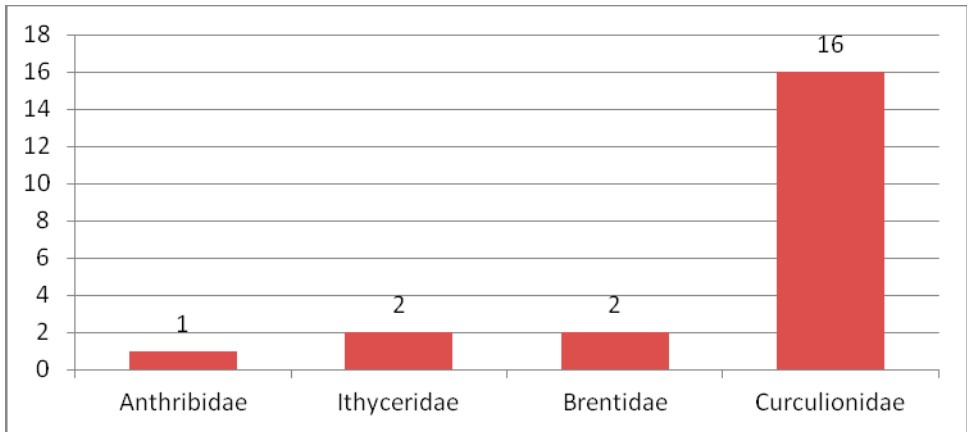

**Figure 7.** Composition of species of Curculionoidea in the Early Eocene fauna.

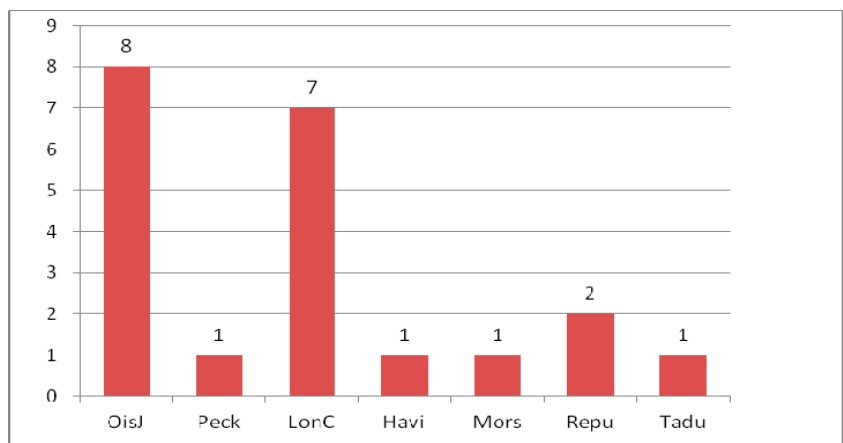

**Figure 8.** Composition of Curculionoidea species between Early Eocene localities.

The connections between the Paleocene and the early Eocene faunas are in a similar fauna structure with a dominance of Curculionidae and the presence of Ithyceridae. Common genera are not found. Almost all of the early Eocene Curculionidae were associated with trees. The only species which could develop on herbs was "*Ceutorhynchus*" *eocenicus*.

3.2.3. Bridgerian Weevil Fauna

Within the American localities Green River and Roan Mountain of the Green River Formation (Figure 9), dating from the end of the early to the beginning of middle Eocene, 75 weevil species were described, with six being common to both localities. Two species of the family Nemonychidae, six species of the family Anthribidae, one species of the family Belidae, one species of the family Rhynchitidae, three species of the family Brentidae, 58 species of the family Curculionidae and four species of the family Scolytidae were described (Figure 10). Entiminae dominate the Curculionidae (Figure 11).

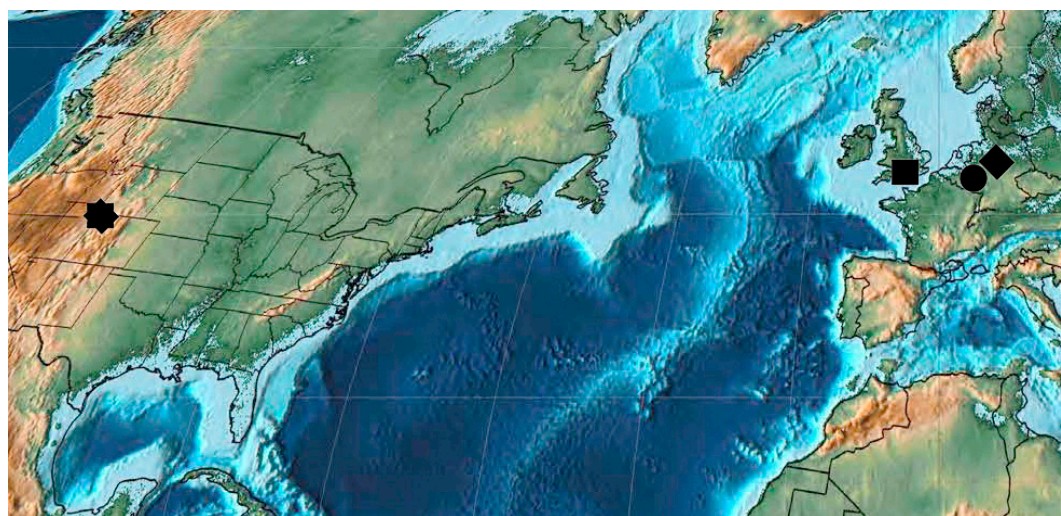

**Figure 9.** Lutetian Curculionoidea deposits: octagon—Roan Mountain and Green River; square—Corfe and Bournemouth; circle—Messel and Eckfelder Maar, rhombus—Geiseltal.

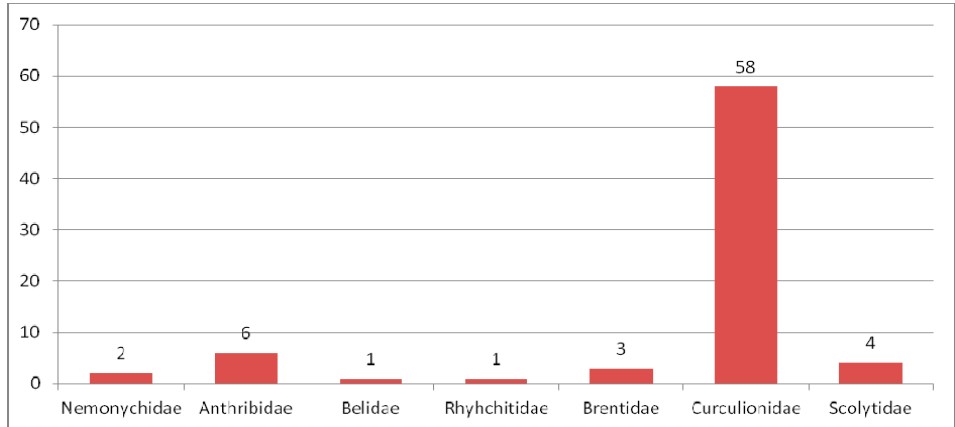

**Figure 10.** Composition of species of Curculionoidea in the Green River Formation.

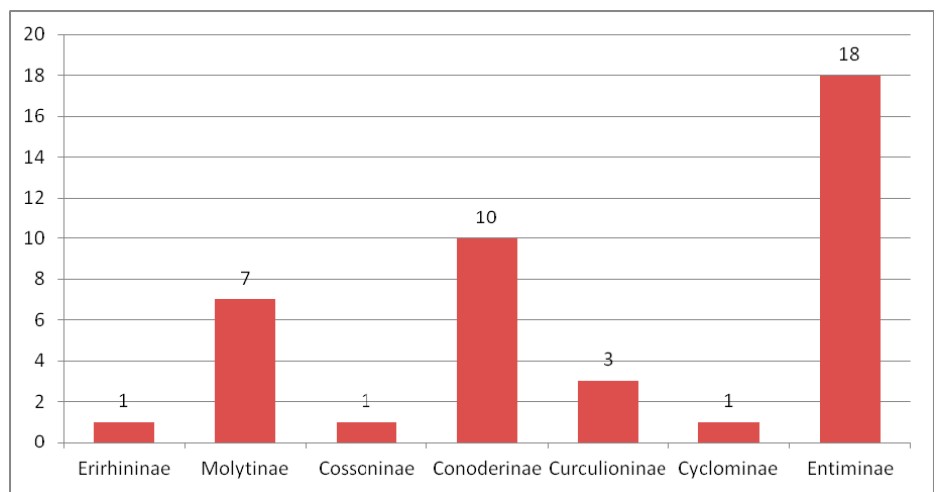

**Figure 11.** Composition of species of Curculionidae in the Green River Formation.

Here are the earliest records (53.5–48.5 Ma) of the subfamilies Nanophyinae, Conoderinae and Cyclominae, tribes Eocaenonemonychini, Cratoparini, Anthribini, Tropiderini, Choragini, Metrioxenini, Apionini, Apostasimerini, Listroderini, Tropiphorini, Entimini, Sciaphilini, Geonemini, Eustylini and Dryocoetini in the fossil record. The faunas of Green River (51 species) and Roan Mountain (30 species) are very similar. In addition to common species, the generic composition is similar also. Five genera (*Sciabregma*, *Lithogeraeus*, *Anthonomus*, *Mitostylus* and *Epicaerus*) are shared by both localities. More samples were collected from the Green River and accordingly more described species. Roan Mountain has a depleted version of the Green River but representatives of the tribes Metrioxenini, Apionini, Erirhinini, Cossonini, Ceutorhynchini and the extinct genus *Steganus* are known only from this locality. Representatives of Nemonychidae, Anthribidae, Rhynchitidae, Aplemonini, Nanophyinae, Molytini, Cryptorhynchini, Conoderintae, *Primocentron*, *Limalophus*, Entimini, Eudiagogini, Eustylini and Dryocoetini are found in the Green River.

The early-middle Eocene faunas of the Green River Formation are quite isolated and do not show obvious connections with other weevil faunas. The genus *Perapion* connects the Green River fauna with the Paleocene of Menat, the tribe Molytini with the Paleocene of Arkhara, the genera *Cossonus* and *Anthonomus* with the Paleocene of Argentina but these widespread groups do not show specificity of the faunogenetic relationships. Two taxa, the genus *Sciabregma* and the tribe Eudiagogini, show specific relationships between the early Eocene of France and the early-middle Eocene of North America. Most of the Curculionoidea of the Green River were associated with forests. It can be assumed that

Aplemonini, Nanophyini, Ceutorhynchini, Apostasimerini, Listroderini and possibly some Entiminae lived on riverine meadows.

3.2.4. Middle Eocene Weevil Fauna

The middle Eocene consists of two stages, Lutetian and Bartonian. Lutetian localities (Figure 9) are Corfe with one described Curculionoid beetle species, Bournemouth with five species (England), Messel with four species, Geiseltal with six species and Eckfelder Maar with four species (Germany). The systematic position of nine species requires clarification. Here are the earliest records of the subfamily Brentinae (48.27 ± 0.22–47.0 Ma) and the tribes Baridini (50.0–42.0 Ma), Brachyderini (48.27 ± 0.22–47.0 Ma) and Sitonini (47.5–42.5 Ma) in the fossil record. Comparison of the fauna of the Lutheian localities among themselves is impossible due to the small number of described forms. I note that the genera *Palaeoalatorostrum*, *Eckfelderolispa*, *Cryptorrhynchites*, *Palaeocrassirhinus*, *Palaeocneorhinus* and *Sitonitellus* are known only from Lutetian. The richest fauna of Messel and Eckfelder Maar have not yet been described.

Bartonian localities (Figure 12) are Kutschlin (Czech Republic) with one species of the subfamily Molytinae and middle Eocene amber (with 140 species). Specimens in Baltic and Rovno amber are usually separate but an attempt has been Made to compare Baltic amber from individual localities, namely, from Scandinavia, Poland and the Kaliningrad Region. There are very few samples of weevis in Polish and Scandinavian amber. Five species (*Baltocyba electrinus*, *Toxorhynchus michalskii*, *Baltonanophyes crassirostre*, *Tachyerges hyperoche* and *Limalophus poinari*) were described from Polish and four species (*Archinvolvulus liquidus*, *Baltoconapium anderseni*, *Electrotribus henningseni* and *Ampharthropelma decipiens*) from Scandinavian amber were not found in Kaliningrad amber. The tribes Rhadinocybini and Notapionini from the Rhadinocybitae were found only in these ambers, respectively. However, there is no reason to consider these amber separate, since the common species (*Electrotribus theryi* and *Paonaupactus sitonitoides*) in Kaliningrad amber were found in Scandinavian and Polish ambers. The absence of species in Kaliningrad amber can be explained by their rarity. Together, 124 species from the families Nemonychidae, Anthribidae, Belidae, Rhynchitidae, Brentidae, Curculionidae, Platypodidae and Scolytidae are recorded fromBaltic amber. The fauna of Rovno amber with 16 species of Curculionoidea has no common species with other amber localities and the two most common species, *Electrotribus theryi* and *Paonaupactus sitonitoides*, were not found in Rovno amber. At the generic level, the fauna of Rovno amber is also quite separate since from 13 genera, only 7 genera are common with Baltic amber. All tribes (except Valenfriesiini) recorded in Rovno amber were also found in Baltic amber. Representatives of eight families were found in Rovno and Baltic amber (Figure 13). A detailed analysis of the faunas of Eocene ambers was carried out in other articles [71,191]. In Eocene amber are the first records of the subfamily Tesserocerinae, the supertribes Rhadinocybitae and Aspidapiitae, the tribes Mecocerini, Allandrini, Zygaenodini, Oxycraspedini, Piezotrachelini, Dorytomini, Plinthini, Acicnemidini, Stromboscerini, Dryotribini, Ellescini, Eugnomini, Rhamphini, Camarotini, Hyperini, Trachyphloeini, Anypotactini, Polydrusini, Hylastini, Hylurgini, Phloeosinini and Polygraphini and the last find of the Sayrevilleinae subfamily in the fossil record. The faunas of Bartonian are very different from the fauna of Lutetian. Common genera are absent.

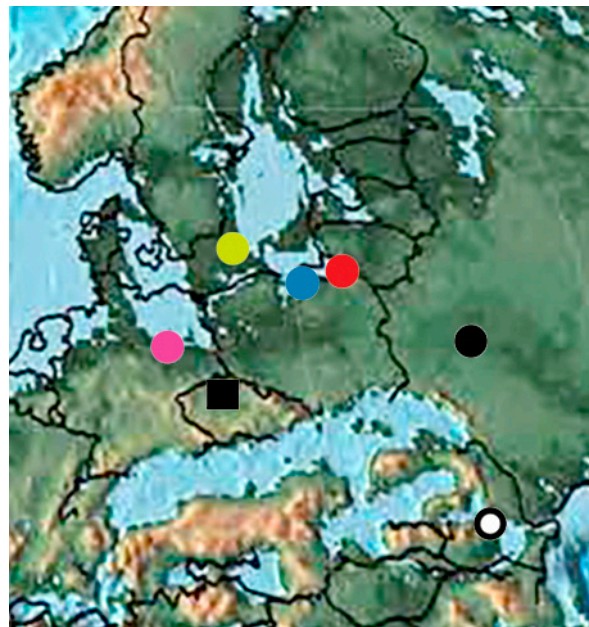

**Figure 12.** Bartonian Curculionoidea deposits: red circle—Baltic amber (Kaliningrad region); blue circle—Polish amber; yellow circle—Scandinavian amber; pink circle—Bitterfeld amber; ring—Romanian amber; circle—Rovno amber; square—Kutschlin.

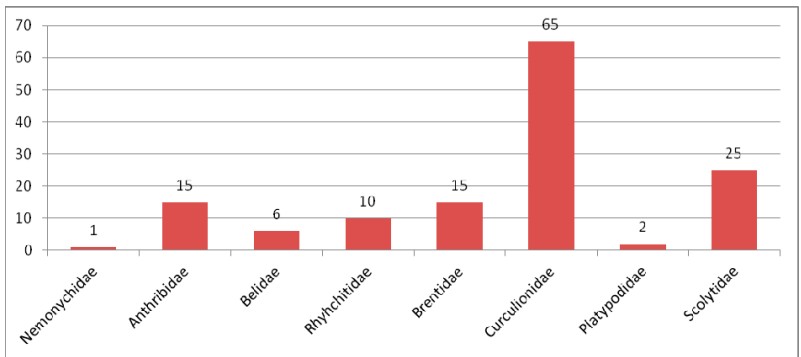

**Figure 13.** Composition of species of Curculionoidea in the Baltic amber.

In general, the weevil fauna of the middle Eocene of Europe is very distinct from earlier faunas. The connections between it and the North American Ypresian-Luthean faunas includethe subtribe Zherichinixenina of the tribe Metrioxenini and the genus *Limalophus*, in both of them. The similarity with the Ypresian and Paleocene faunas of Europe lies only in the presence of widespread groups at the level of tribes (Cryptorhynchini, Ceutorhynchini, Curculionini and Naupactini) and subfamilies (Anthribinae, Apioninae, Molytinae, Dryophthorinae, Cossoninae, Curculioninae).

3.2.5. Late Eocene Weevil Fauna

The Priabonian faunas are described from five localities (Figure 14), two in France (Celas and Ales-Monteils), one in the USA (Florissant), one in the south of the Russian Far East (Biamo) and one in Brazil (Fonseca). Some 184 species were described from these localities. One species of the tribe Hyperini was described from Biamo. The genus *Duartia*, probably related to Scolytidae, is known from Fonseca. *Palaeorhopalotria neli* was described from Ales-Monteils. This species belongs to the extinct tribe Palaeorhopalotriini belonging to the supertribe Allocorynitae now distributed in Central America [192] and also noted in the Miocene [193]. The three representatives of Entiminae and species from the tribe Corimaliini (earliest record) are known from Celas.

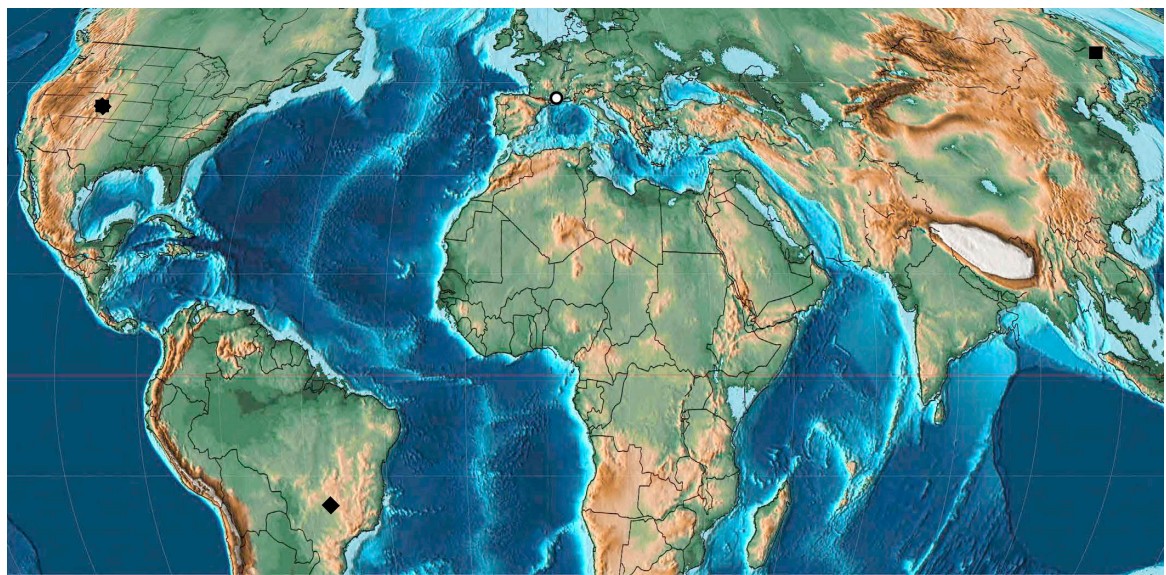

**Figure 14.** Priabonian Curculionoidea deposits: Circle—Celas and Ales-Monteils; octagon—Florissant; square—Biamo; rhombus—Fonseca.

The Florissant fauna, from which 177 species are known (Figure 15), is of great interest as being the richest of the Paleogene fauna. However, families Nemonychidae, Ithyceridae and Platypodidae are absent in this fauna. Representatives of the tribes Cratoparini, Anthribini, Ecelonerini, Tropiderini, Metrioxenini, Auletini, Rhynchitini, Eugnamptini (earliest record), Euscelini (earliest record), Apionini, Erirhinini, Dorytomini, Molytini, Magdalini (earliest record), Cryptorhynchini, Cleonini (earliest record), Dryophthorini, Sphenophorini (earliest record), Cossonini, Apostasimerini, Baridini, Ceutorhynchini, Cnemogonini (earliest record), Curculionini, Anthonomini, Rhamphini, Tychiini, Camarotini, Listroderini, Tropiphorini, Eudiagogini, Hyperini, Hormorini (earliest record), Naupactini, Geonemini, Psallidiini (earliest record), Tanymecini (earliest record), Hylastini and Phloeotribini (earliest record) are found in Florissant deposits (34.07 ± 0.10 Ma).

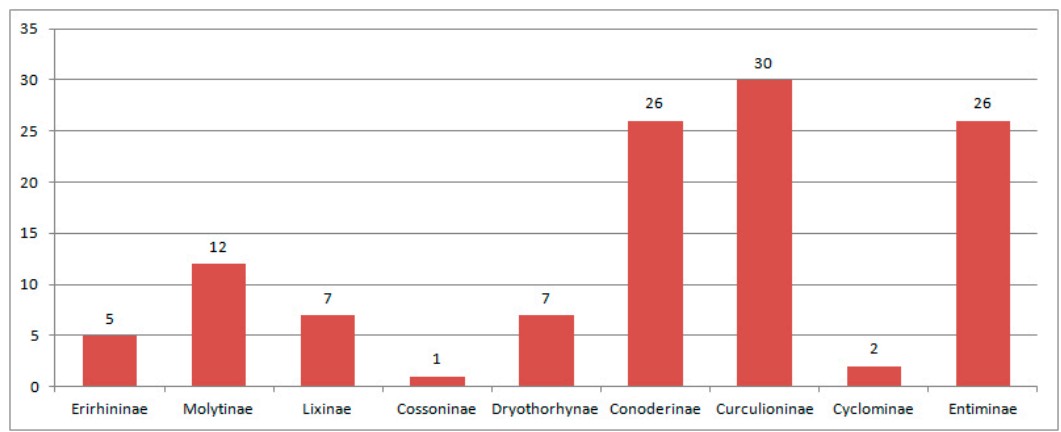

**Figure 15.** Composition of species of Curculionoidea in the Florissant.

About half (52%) of the genera are recent. This fauna of Florissant is very different from the fauna of Barton amber, where 70% of genera are extinct. The structure of the fauna is rather unusual (Figure 15). Curculionidae dominates but Rhynchitidae and Brentidae also play an important role. Three subfamilies (Curculioninae, Conoderinae and Entiminae) of the weevils form the basis of the fauna (Figure 16), while Curculioninae and Entiminae and Conoderinae do not play a significant role (Figure 17) in Baltic amber. It differs from the fauna of the Green River, where only Curculionidae

dominates and at the subfamily level of this family, the fauna is formed by Entiminae and Conoderinae. However, due to the presence of common groups (genera and tribes, usually found only in these localities), the Florissant fauna shows similarities primarily with the Green River (15 common genera and tribes) and Baltic and Rovno amber (11 common genera and tribes) faunas.

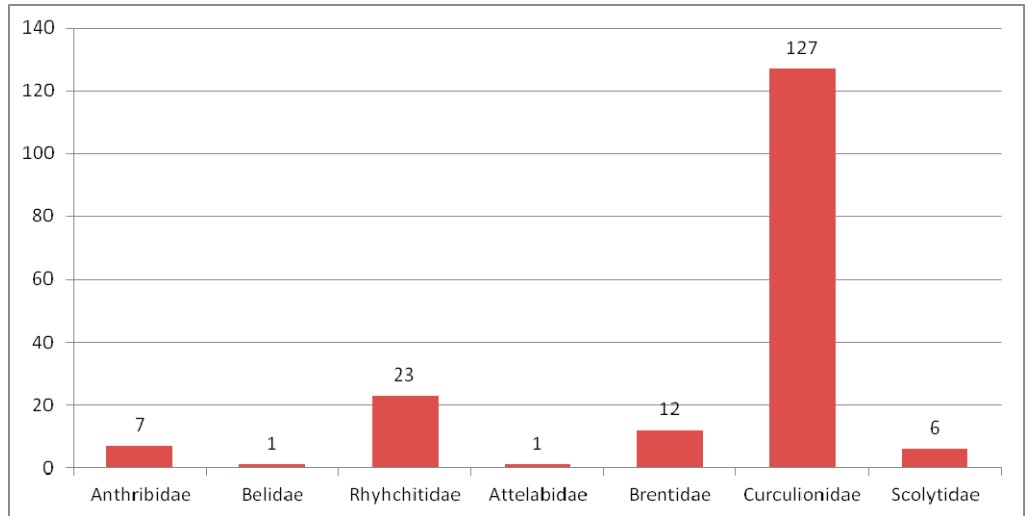

**Figure 16.** Composition of species of Curculionidae in the Florissant.

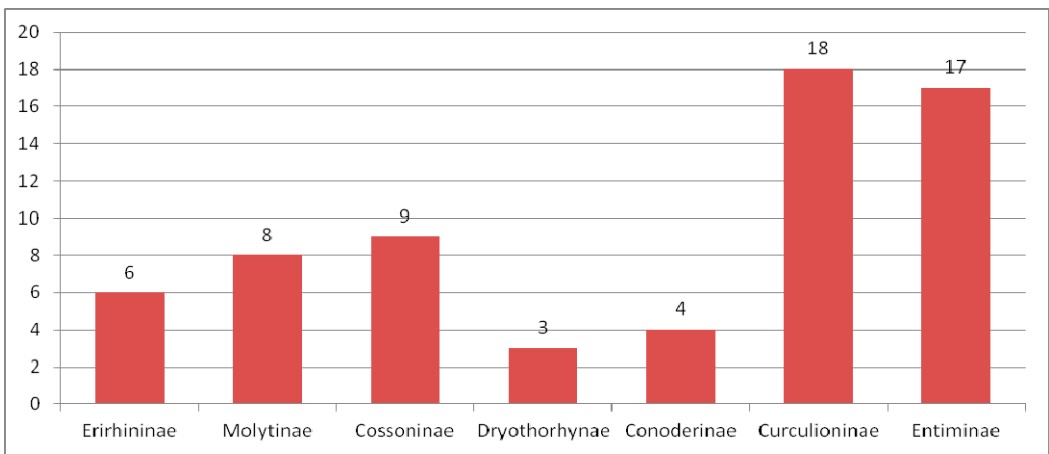

**Figure 17.** Composition of species of Curculionidae in the Baltic amber.

### 3.2.6. Comparison of the Eocene Weevil Faunas

In the Eocene, weevil diversity occurs from early to late (Figures 18 and 19). A small number of species in the Lutetian is due to poorly studied fauna. The number of representatives of modern genera increases towards the late Eocene.

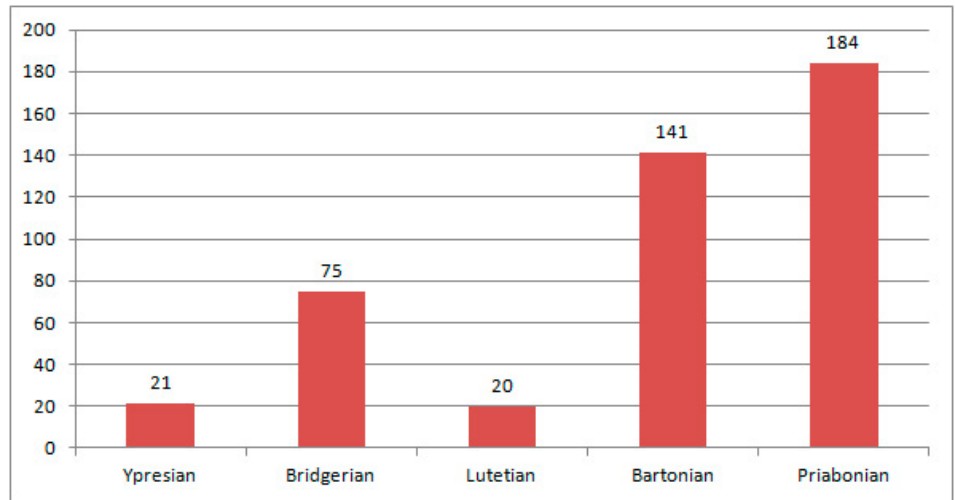

**Figure 18.** Change in the number of Curculionoidea species in the Eocene.

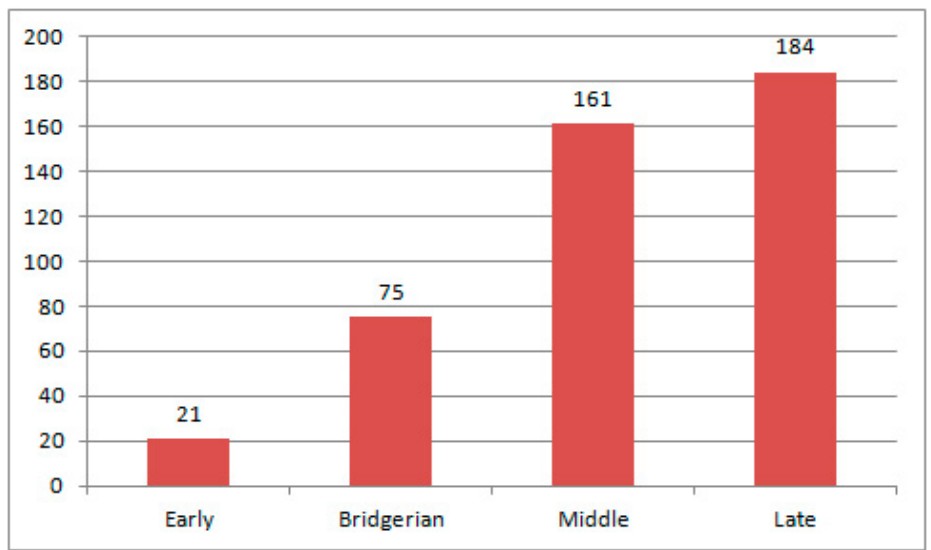

**Figure 19.** Change in the number of Curculionoidea species in the Eocene.

### 3.3. Oligocene Weevil Fauna

The Oligocene fauna of Curculionoid beetles is represented by 94 described species from six families. Ten localities related to the early (White River Badlands from USA, Brunnstatt and Corent from France), middle (border between early and late) (Sieblos and Kleinkembs from Germany, Gaube and Puy-Saint-Jean from France) and late Oligocene (Luzice from Czech Republic, Aix-en-Provance from France, Rott from Germany) (Figures 20 and 21), contain the remains of Curculionoidea that were described. The Greenland locality of Kap Dalton, where representative of the Nemonychidae family was described, dates from the Oligocene without specifying the stage [194]. Curculionoidea are also represented in the Rupelian Quilchena (Canada), Cereste (France), Seifhennersdorf (Germany), Ahuehuetes (Mexico), Kundratice (Czech Republic), Hutt Enspel (Germany), Ashutas (Kazakhstan), Perekishkyul' (Azerbaijan) localities [30,195–199].

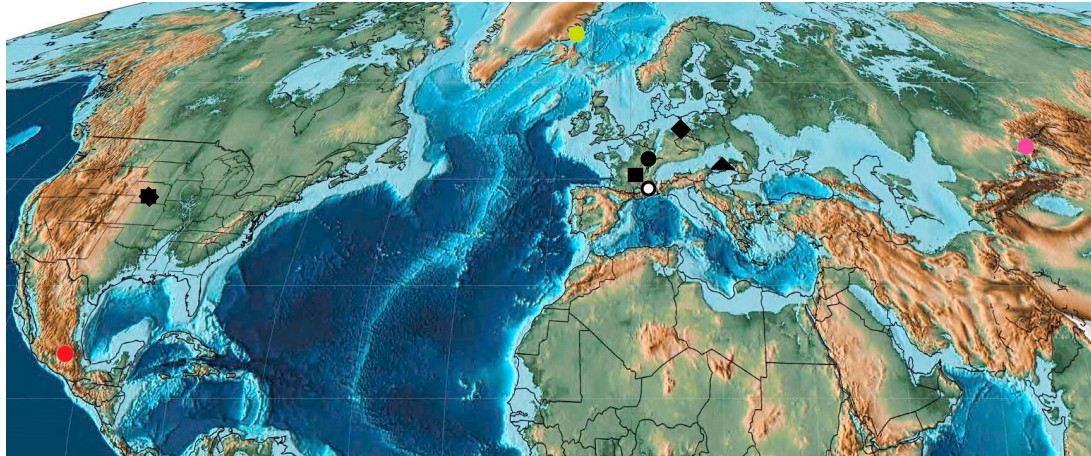

**Figure 20.** Rupelian Curculionoidea deposits: octagon—White River Badlands; circle—Brunnstatt; square—Corent; ring—Cereste; rhombus—Seifhennersdorf; red circle—Ahuehuetes; triangle—Kundratice; yellow circle—Kap Dalton; pink circle—Kenderlyk II.

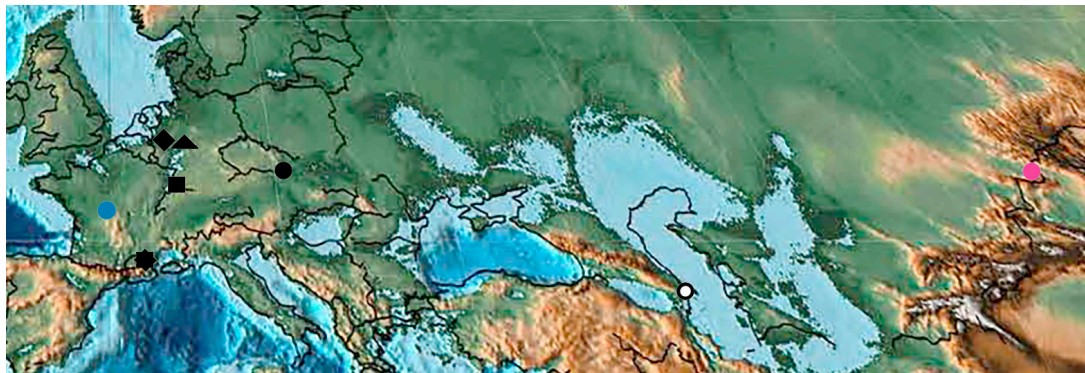

**Figure 21.** Middle and late Oligocene Curculionoidea deposits: circle—Luzice; rhombus—Rott and Enspel; octagon—Aix-en-Provance; pink circle—Ashutas; ring—Perekishkyul'; triangle—Sieblos; square—Kleinkembs; blue circle—Gaube and Puy-Saint-Jean.

Representatives of the family Curculionidae and a few species of the subfamily Apioninae from the family Brentidae were found in these localities.

### 3.3.1. Review of Curculionoidea Families in the Oligocene

#### Nemonychidae in the Oligocene

The family Nemonychidae in the Oligocene is represented by one species assigned to the tribe Eocaenonemonychini from the subfamily Cretonemonychinae. This is the latest find of the family in the fossil record.

#### Anthribidae in the Oligocene

The family Anthribidae is very poorly represented in Oligocene deposits (Figure 22). No representatives of the subfamily Anthribinae were found. One species of the tribe Choragini was described from the late Oligocene. Choraginae were very rare in the Eocene, so this find shows a greater abundance of this subfamily in the Oligocene. Three species of the genus *Bruchela* of the subfamily Urodontinae were described from the Oligocene of France and Germany. This is the first reliable indication of Urodontinae in the fossil record.

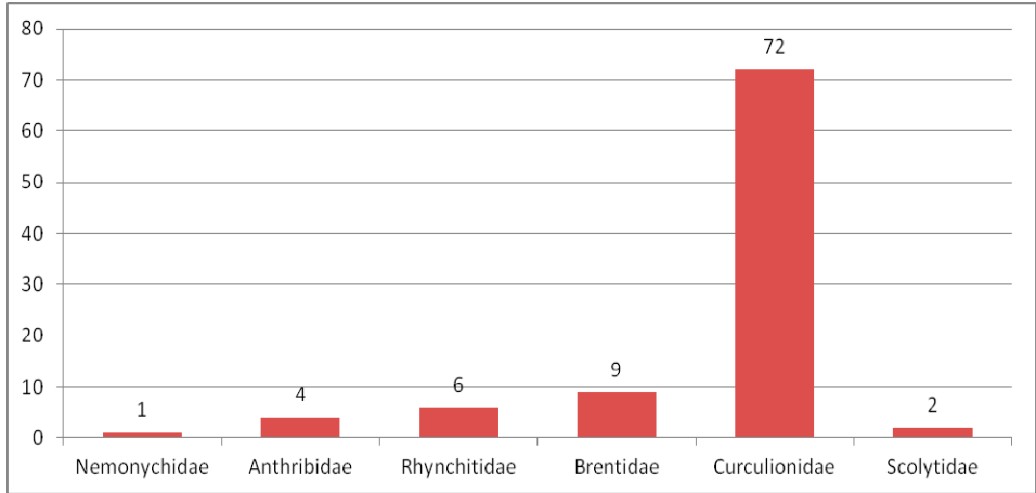

**Figure 22.** Composition of species of Curculionoidea in the Oligocene.

Rhynchitidae in the Oligocene

Six species of the family Rhynchitidae occur in the late Oligocene. The extinct genus of the tribe Vossicartini, now distributed only in tropical Africa and Madagascar [60], is the only find of this tribe in the fossil record. The tribe Rhynchitini is represented by extinct species of recent genera belonging to both the primitive subtribe Perrhynchitina and the advanced subtribe Rhynchitina.

Attelabidae in the Oligocene

The Attelabidae family was not found in the Oligocene.

Brentidae in the Oligocene

The family Brentidae is found in the Oligocene of Europe. Eight species belong to the subfamily Apioninae and one to the subfamily Nanophyinae (tribe Nanophyini). The systematic position of most species in the tribes and genera requires clarification. One species can be assigned to the genus *Perapion* of the tribe Aplemonini. Previously, this genus was discovered in the Paleocene of Europe and the Eocene of North America.

Curculionidae in the Oligocene

The Curculionidae family is represented by 72 species from the subfamilies Erirhininae, Molytinae, Lixinae, Dryophthorinae, Cossoninae, Conoderinae, Curculioninae and Entiminae (Figure 23). Only the subfamily Cyclominae was not encounteredin Oligocene deposits. Two species from the tribes Erirhinini and Bagoini of the subfamily Erirhininae are noted in the Oligocene. Representatives of the tribes Molytini, Pissodini, Magdalini and Cryptorhynchini belonging to the subfamily Molytinae were described from different Oligocene localities. *Lithopissodes luschitzensis* is the only Pissodini in the fossil record. Two species of the genus *Larinus* of the tribe Lixini and two species formally assigned to the genus *Cleonis* belong to the subfamily Lixinae. The subfamily Dryophthorinae with one species of the tribe Dryophthorini and one species of the tribe Sphenophorini is known from the late Oligocene. Here is the only record of the tribe Sphenophorini in the Paleogene of Europe. Only two species from the tribes Rhyncolini and Cossonini of the subfamily Cossoninae were found in the late Oligocene. It is important to note that the tribe Dryotribini common in the middle Eocene amber are not found in the Oligocene. The diverse subfamily Conoderinae is poorly represented in the Oligocene. Two species of Bariditae are noted. *Lithogeraeus comminute* from the tribe Apostasimerini was described from the early Oligocene of the United States. One species placed in the genus *Baris* of the tribe Baridini is known from the early Oligocene of Europe. The supertribe Conoderintae is not found in the

Oligocene. The supertribe Ceutorhynchitae is represented by the tribes Ceutorhynchini and Phytobiini. The tribes Smicronychini, Anthonomini and Tychiini from the subfamily Curculioninae are found in the Oligocene. "*Smicronyx*" *antiquus* is the only species of the tribe Smicronychini in the fossil record. "*Anthonomus*" *soporus* from the tribe Anthonomini is Marked for the early Oligocene of the United States. The Oligocene species of the genera *Sibinia* and *Tychius* belong to the tribe Tychiini. Twenty three species from six tribes belong to the subfamily Entiminae. The most diverse tribes are Tropiphorini with 7 species and Tanymecini with 6 species. One or two species belong to the tribes Sciaphilini, Sitonini, Geonemini and Brachyderini. The position of four species in these tribes requires clarification. The family Platypodidae is not found in the Oligocene. Two species of the genus *Hylesinus* of the tribe Hylesinini from the family Scolytidae are known from the late Oligocene of France.

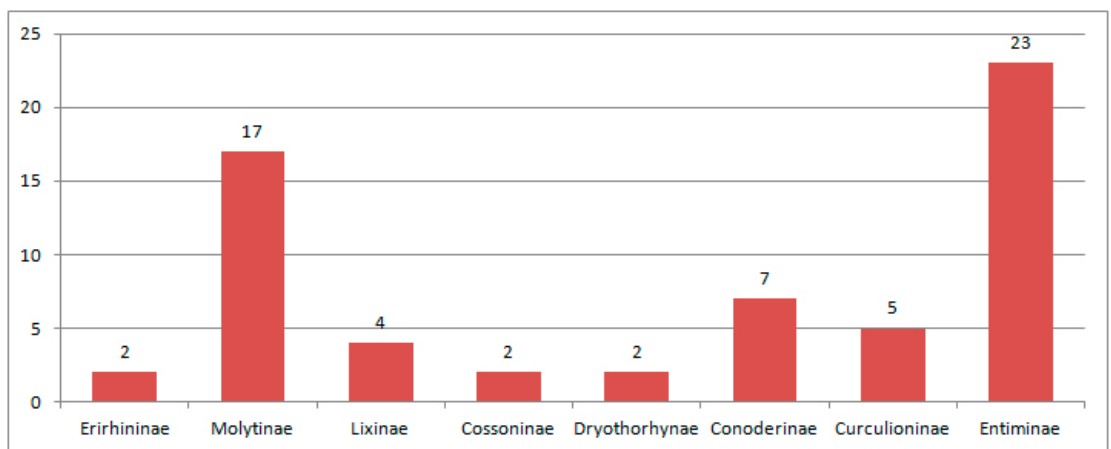

**Figure 23.** Composition of species of Curculionidae in the Oligocene.

Localities with the described fauna are divided into those located in North America and Europe. Curculionidae from the only Asian locality in Kazakhstan have not yet been described. Species were described from the White River Badlands and Kap Dalton in Greenland. Their fauna is radically different from the fauna of the Old World. At least 13 species of the family Curculionidae are known from the White River Badlands. These are representatives of the genera *Procas*, *Archaeoheilus*, *Lithogeraeus*, *Anthonomus*, *Limalophus*, *Mitostylus* and *Epicaerus*. None of these genera were found in the Oligocene of Europe. The fauna of this locality shows a close similarity with the fauna of the middle Eocene, Green River and Road Mountain. Moreover, three species from the White River are found inthese faunas. At the level of genera and tribes, the similarity between them is complete. One can consider White River as a reduced version of the Eocene American faunas. The only Curculionoidea from the Oligocene of Greenland belongs to the North American tribe Eocaenonemonychini and possibly to a genus described from the Green River.

### 3.3.2. Early Oligocene Weevil Fauna

Twenty one species are found in the early Oligocene of Europe (France), 13 species in Brunnstatt and 8 species in Corent. There is one species of the subfamily Urodintinae (Anthribidae), three species of the subfamily Apioninae (Brentidae), 17 species of Curculionidae (one species of the subfamily Erirhininae, five species of the subfamily Molytinae, three species of the subfamily Lixinae, four species of the subfamily Conoderinae, two species of the subfamily Curculioninae, one species of the subfamily Entiminae and one species of the subfamily insertae sedis). The faunas of Brunnstatt and Corent are very different. No species and genera occur in both localities. Three species in Corent can be attributed to the meadow complex (*Anisorhynchus* and Cleonini) and three to the forest complex (representatives of Molytinae). The fauna of Brunnstatt is much more diverse. Most species noted there belong to the meadow-steppe complex (11 species from the genera *Bruchela*, *Larinus*, *Baris*, *Ceutrhynchus*, *Smicronyx*,

*Tychius* and the subfamily Apioninae) while Molytinae probably belonged to the forest complex and *Bagous* to the near-water habitat.

### 3.3.3. Middle Oligocene Weevil Fauna

Only ten weevil species were described from the middle Oligocene (boundaries are of the early and late Oligocene) and six of them belong to Curculionidae insertae sedis. The rest belong to the subfamilies Molytinae, Conoderinae and Curculioninae of the family Curculionidae.

### 3.3.4. Late Oligocene Weevil Fauna

Thirty four species were described from the late Oligocene. One species is known from Luzice, 22 from Aix-en-Provance and 27 from Rott. The fauna of Aix-en-Provance is represented by the three families, Brentidae, Curculionidae and Scolytidae (Figure 24). They involveforest species (*Dryophthorus incertus*, *Cossonus robustus*, *Rhytidoderes* spp., *Hylesinus* spp.) and open space species (Apioninae, "*Sibinia*" *whitneyi*), *Protainophthalmus* spp. Extinct species belong to widespread groups. Noteworthy are six species of the extinct genus *Protainophthalmus* known only from this locality. This genus belongs to the subtribe Tainophthalmina that is distributed in Central and East Asia and is also found in the Mediterranean. Three extinct species of the recent genus *Rhytideres* from the Mediterranean are also only found in Aix-en-Provance. The families Anthribidae, Rhynchitidae, Brentidae and Curculionidae form the Rott fauna (Figure 25). Species of the family Curculionidae dominate. Most species belong to widespread genera (*Choragus*, *Perapion*, *Nanophyes*, *Hylobius*, *Magdalis*, *Acalles*, *Larinus*, *Sphenophorus*, *Rhyncolus*, *Ceutorhynchus* and *Tychius*). Three species belong to the West Palaearctic genera *Bruchela* and *Tatianaerhynchites*. Representatives of the oriental genera *Cartorhynchites* and *Opacoinvolvulus* are also present. The two genera *Germanocartus* and *Brachymycterus* are endemic to this locality. The Rott fauna consisted of approximately half forest and meadow species. It is obviousthat the Rott fauna is much more diverse than that of Aix-en-Provance. Common genera are absent.

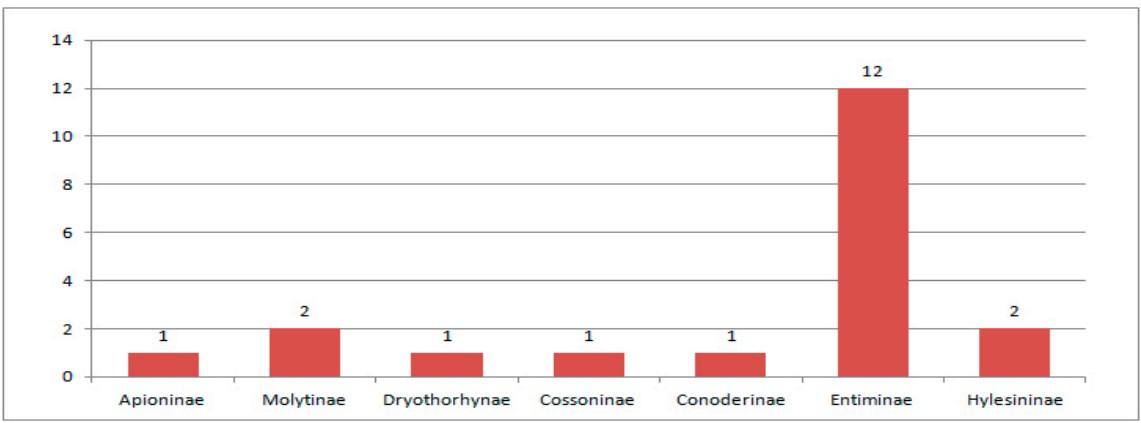

**Figure 24.** Composition of species of Curculionoidea in Aix-en-Provance.

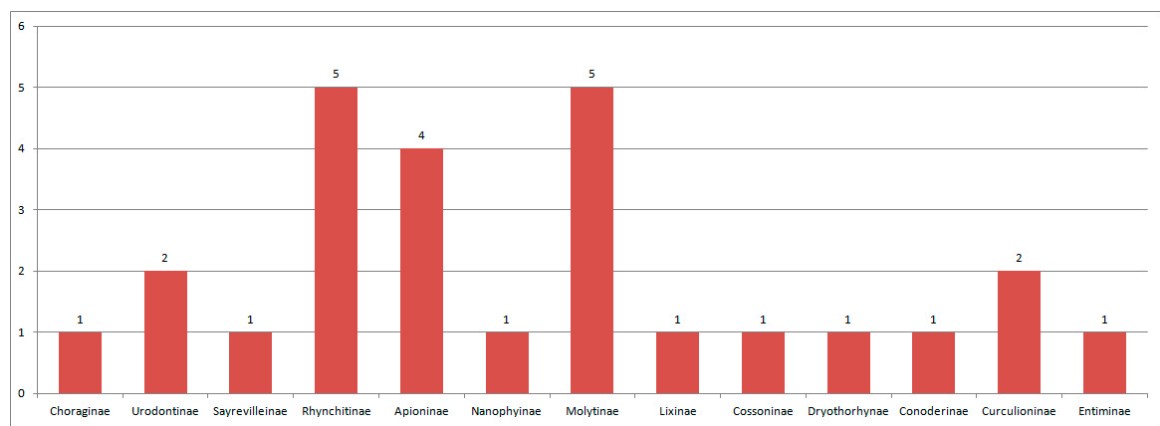

**Figure 25.** Composition of species of Curculionoidea in Rott.

### 3.3.5. Comparing of the Oligocene Weevil Faunas

Comparing the faunas of the early and late Oligocene, it is noted that the Chattian faunas are more diverse. They contain not only representatives of widespread or West Palaearctic genera but also oriental forms. Relationships with the African fauna are indicated the presence of a representative of the tribe Vossicartini. In general, the Oligocene fauna was formed by forest and meadow-steppe species, which indicates the presence of open spaces.

## 4. Discussion

In total, 564 species of Curculionoidea from nine families were described from the Paleogene. They occur in seven localities in North America (four from Eocene and three from Oligocene), three in South America (one from the Paleocene and two from the Eocene), 34 in Europe (two from the Paleocene, 18 from the Eocene and 14 from the Oligocene) and six in North and East Asia (two from each of the Paleocene, the Eocene and the Oligocene). They have not yet been found in African, South Asian and Australian Paleogene localities. An increase in the number of species is observed from the Paleocene to the Eocene (end of the Eocene) and then decreases in the Oligocene (Figure 26). The greatest diversity of Curculionoidea is described from the Eocene of Europe and North America. The richest faunas are known from the terminal Eocene of Florissant (177 species), the middle Eocene Baltic amber (124 species) and the early-middle Eocene Green River formation (75 species). Discovery of relict groups with a local distribution in the contemporary fauna, such as Ithyceridae—Chilecarinae and Ithycerinae, Belidae—Metrioxenini, Rhynchitinae—Sayrevilleinae, were Made in the Eocene of America and the Eocene and Oligocene of Europe. The most numerous group of all the Paleogene Coleoptera faunas is the superfamily Curculionoidea [200]. The family Curculionidae dominates in all localities of the Paleogene. Anthribidae, Rhynchitidae, Brentidae and Scolytidae are sometimes subdominant in Eocene localities. For example, Anthribidae, Brentidae and Scolytidae account for 40% of the Baltic amber fauna (weevils 47%) and Rhynchitidae comprise 13% of the Florissant fauna (weevils 72%). In all localities, species associated with woody vegetation dominate. Species associated with herbaceous vegetation are present in most localities since the middle Paleocene. Their proportion is increasing in the Oligocene. Further study of the Curculionoidea from Paleogene localities May clarify the picture somewhat.

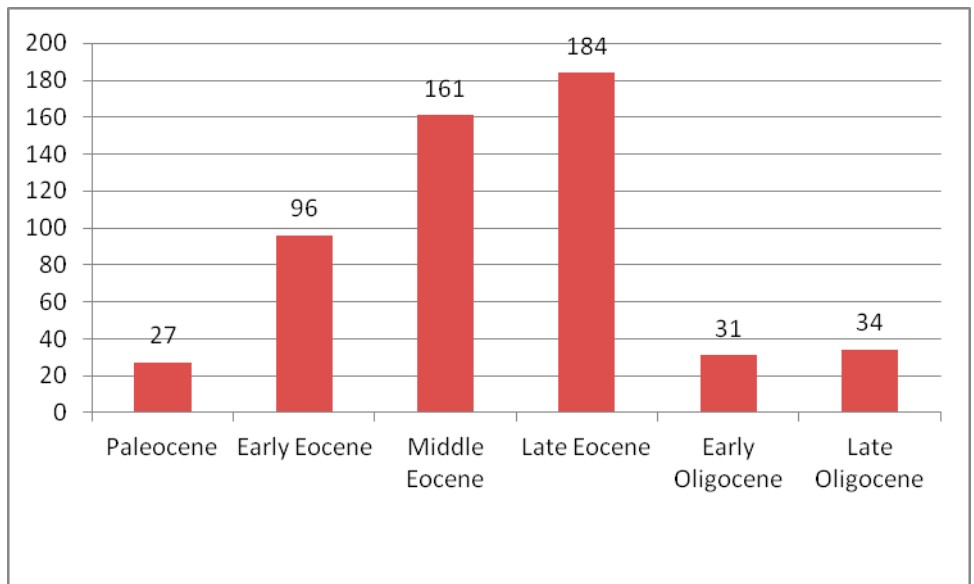

**Figure 26.** Change in number of Curculionoidea species in the Paleogene subepochs. The fauna of the Green River Formation is included in the Early Eocene.

**Funding:** The study was partially supported by the Russian Foundation for Basic Research (project nos. 18-04-00243-a and 19-04-00465-a) and the Federal Fundamental Scientific Research Program for 2013–2020 (project no. AAAA-A16-116121410121-7).

**Acknowledgments:** The author thanks Many colleagues for their assistance throughout his studies: V.I. Alekseev, A.R. Manukyan and A.V. Smirnova (Kaliningrad Regional Amber Museum, Russia: Kaliningrad), A. Allen (USA: Boise), A. Bukejs (Institute of Life Sciences and Technologies, Daugavpils University, Latvia: Daugavpils), D. Berthet (Centre de Conservation du musée des confluences, France, Lyon), J. Damzen (Lithuania: Vilnius), F. Eichmann (Germany: Hannover), A. Górski (Poland: Bielsko-Biaùa), C. Gröhn (Germany: Glinde), V.A. Gusakov (Russia: Moscow), A.G. Kirejtshuk (Zoological Institute RAS, Russia: St. Petersburg), U. Kotthoff (Center of Natural History, Germany: Hamburg), N.V. Martynovich (Museum of the World Ocean, Russia: Kaliningrad), A. Nel (Muséum national d'histoire naturelle, France: Paris), E.E. Perkovsky and V.Y. Nazarenko (Schmalhausen Institute of Zoology, NASU, Ukraine: Kiev), A.G. Ponomarenko, A.P. Rasnitsyn, E.D. Lukashevich, D.E. Shcherbakov, I.D. Sukatsheva, D.V. Vassilenko (Borissiak Paleontological Institute RAS, Russia: Moscow), E. Yu. Shevnin (Russia: Novosibirsk), L.B. Vilhelmsen (Zoological Museum, University of Copenhagen, Denmark: Copenhagen), K. Szczepaniak (Earth Institute, Poland: Warsaw), T. Wappler (Hessisches Landesmuseum Darmstadt, Germany: Darmstadt), Ch. R. Scotese (Northwestern University, USA: Evanston) allowing me to use his paleomaps and G.O. Poinar, Jr (Oregon State University, USA: Corvallis) for improving the Manuscript. The two reviewers are also acknowledged for the valuable comments that improved the manuscript.

**Conflicts of Interest:** The author declares no conflict of interest.

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
