# Peer review of "Fossil History of Curculionoidea (Coleoptera) from the Paleogene"

_geosciences, doi:10.3390/geosciences10090358_

Round 1

Reviewer 1 Report

The paper is good for publication and well prepared. some editory corrections were made.

1.The author have to add the author(s) and year of publication for suprageneric taxa when they are first mentioned.

2. Besides, it id poddiblr to publish new names without description and therefore they should indicated as sp. nov in litt (in litt. = in litteris and in Italics).

3.The references for argumentation of the age of Baltic amber should be chosen among publications of more experiencd experts.

4. The proportions of different coleopterous groups in Cenozoic outcrops of different age are discussed in the below monograph the reference to which would be better to add in the Discussion

Kirejtshuk, A.G.; Ponomarenko, A.G.; Kurochkin, A.S.; Alexeev, A.V.; Gratshev, V.G.; Solodovnikov, A.V.; Krell, F.‐T.; Soriano, C. The beetle (Coleoptera) fauna of the Insect Limestone (late Eocene), Isle of
Wight, southern England. Earth Environm. Sci. Trans. R. Soc. Edinburgh 2019, 110, 405–492.
doi:10.1017/S1755691018000865

Author Response

I thank the reviewer for his comments that helped improve my manuscript.

1.The author have to add the author(s) and year of publication for suprageneric taxa when they are first mentioned.
- This was done.

2. Besides, it id poddiblr to publish new names without description and therefore they should indicated as sp. nov in litt (in litt. = in litteris and in Italics).
- I agree with the reviewer that it is not very correct if the articles describing these new taxa are being prepared or are under review. But if articles are accepted for publication in the journal, then I think that these names can be used as "in print". The designation "sp. nov." should be used when describing of new taxa.

3.The references for argumentation of the age of Baltic amber should be chosen among publications of more experiencd experts.
-The age of Baltic amber is still debatable. I used 34–48 million years old (Lutetian-Bartonian), middle Eocene (Seyfullah et al. 2018) in my review (Legalov, 2020). The fauna of the Baltic amber differs from the faunas of the Middle Eocene (Lutetian) in Germany and is similar to the faunas of Priabonian. Therefore, the Bartonian suggested by Bukejs at al. (2019) is consistent with my views on the age of this amber.
Seyfullah, L.J.; Beimforde, C.; Dal Corso, J.; Perrichot, V.; Rikkinen, J.; Schmidt, A.R. Production and preservation of resins – past and present. Biol. Rev. 2018, 93, 1684–1714, doi:10.1111/brv.12414.

4. The proportions of different coleopterous groups in Cenozoic outcrops of different age are discussed in the below monograph the reference to which would be better to add in the Discussion
Kirejtshuk, A.G.; Ponomarenko, A.G.; Kurochkin, A.S.; Alexeev, A.V.; Gratshev, V.G.; Solodovnikov, A.V.; Krell, F.‐T.; Soriano, C. The beetle (Coleoptera) fauna of the Insect Limestone (late Eocene), Isle of Wight, southern England. Earth Environm. Sci. Trans. R. Soc. Edinburgh 2019, 110, 405–492.
doi:10.1017/S1755691018000865
-This was done.

Reviewer 2 Report

This paper by Legalov summarises the state of knowledge of fossil Curculionoidea from the Palaeogene. This is a useful work, but in its current form it suffers from a lack of clear focus. It has also been submitted too soon, with many formatting errors and hanging references to figures, which should have been tidied up prior to submission. I recommend that the author consider the key aspects of the story he wants to tell, and deliver it more concisely.

Subheadings should be added to the paper to make the organisation and the logical flow of the paper more readily accessible to the reader.

The author uses an unusally broad concept of the Ithyceridae, which combines the Caridae and the Ithycerinae. In using this classification, he is at odds with the currently accepted classification of a distinct Caridae, and with Ithycerinae included in a broad concept of Brentidae [1], a classification which has received support from molecular evidence [2]. If the author persists in using a broad Ithyceridae, he needs to make this clear in an introductory section, summarise why he prefers this classification. More generally, I find the author over-confident about the inferred relationships of these fossils, and about their inferred life histories.

Statements like "Here is the first reliable indication of Urodontinae in the fossil record" (Line 565) should be followed with some explanation as to why this is important. Does this correspond with our understanding of weevil phylogeny? Is it earlier/later than might be expected? What confidence do we have that these Urodontinae are in that tribe?

A summary of the timing, setting and importance of the Palaeogene would be a good addition for those readers who are not as familiar with the geological time. A figure showing the duration and context of the Palaeogene would be very useful in this respect.

All plots in the paper have a 3D effect added to them. This is inappropriate and unhelpful for presenting data in a scientific paper. All charts should be presented in two dimensions only. I would also recommend that all pie charts be changed to bar charts.

Other suggestions follow:

Table 1:
- In several entries (e.g rows 15, 16, 76, 77, ... 544), only the genus name has been italicised.
- Lines 136--139: These names have been validly published in 2019, in his reference 69.

Line 93: "(Figs. ??):" needs to be completed.

Line 165: "(reference)" needs to be inserted

Lines 134 & 135: Strange spiral-shaped glyphs are present in the PDF.

Line 679: "They have not yet been found in African, South Asian and Australian Paleogene locations" What locations are these? Does the context of these locations mean that Curculionoidea could be expected from them?

Figure 20 on Pg 37 should be Figure 21.

Figure 25 is unnecessary if figure 26 is included.

Figure 24 on pg 40 should be Figure 26.

Author Response

I thank the reviewer for his comments that helped improve my manuscript.

This paper by Legalov summarises the state of knowledge of fossil Curculionoidea from the Palaeogene. This is a useful work, but in its current form it suffers from a lack of clear focus. It has also been submitted too soon, with many formatting errors and hanging references to figures, which should have been tidied up prior to submission. I recommend that the author consider the key aspects of the story he wants to tell, and deliver it more concisely.
-The aim of this work was to characterize the faunas of weevils in the Paleogene in time and space (in different locations).
-”with many formatting errors and hanging references to figures” is corrected in the layout.

Subheadings should be added to the paper to make the organisation and the logical flow of the paper more readily accessible to the reader.
-This was done.

The author uses an unusally broad concept of the Ithyceridae, which combines the Caridae and the Ithycerinae. In using this classification, he is at odds with the currently accepted classification of a distinct Caridae, and with Ithycerinae included in a broad concept of Brentidae [1], a classification which has received support from molecular evidence [2]. If the author persists in using a broad Ithyceridae, he needs to make this clear in an introductory section, summarise why he prefers this classification. More generally, I find the author over-confident about the inferred relationships of these fossils, and about their inferred life histories.
- I do not understand the term "accepted classification". In science, everyone can voice an opinion if one is reasonable. In my numerous works (see the list) I have expressed my views and argued for them. The broad interpretation of this family Ithyceridae is my opinion based on extensive paleontological material. I see no convincing evidence that this position is wrong. Molecular data cannot yet be reliable, since the development of this direction is just beginning. Perhaps in 20-30 years the situation with molecular research will change. This manuscript does not discuss the Curculionoidea system and I give links to the used system and of course I use the system which I developed.
Legalov A.A. 2009. A review of fossil and recent species of the family Ithyceridae (Coleoptera) from the world fauna // Амурский зоологический журнал. Т. 1. № 2. С. 117–131 + col. pl. I–IV.
Gratshev V.G., Legalov A.A. 2011. New Mesozoic Ithyceridae beetles (Coleoptera) // Paleontological Journal. Vol. 45. No 1. 76–81. https://doi.org/10.1134/S0031030111010060
Legalov A.A. 2013. New and little known weevils (Coleoptera: Curculionoidea) from the Paleogene and Neogene // Historical Biology. Vol. 25. No. 1. P. 59–80. https://doi.org/10.1080/08912963.2012.692681
Gratshev V.G., Legalov A.A. 2014. The Mesozoic stage of evolution of the family Nemonychidae (Coleoptera, Curculionoidea) // Paleontological Journal. Vol. 48. No 8. P. 851–944. https://doi.org/10.1134/S0031030114080012
Legalov A.A., Poinar G.Jr. 2015. New tribes of the superfamily Curculionoidea (Coleoptera) in Burmese amber // Historical Biology. Vol. 27. No. 5. P. 558–564. https://doi.org/10.1080/08912963.2014.896908
Legalov A.A. 2015. Fossil Mesozoic and Cenozoic weevils (Coleoptera, Obrienioidea, Curculionoidea) // Paleontological Journal. Vol. 49. No 13. P. 1442–1513. https://doi.org/10.1134/S0031030115130067
Poinar G.,Jr., Brown A.E., Legalov A.A. 2016. A new weevil tribe, Mekorhamphini trib. nov. (Coleoptera, Ithyceridae) with two new genera in Burmese amber // Biological Bulletin of Bogdan Chmelnitskiy Melitopol State Pedagogical University. Vol. 6. No. 3. P. 157–163. https://doi.org/10.15421/201683
Legalov A.A. 2018. Annotated key to weevils of the world. Part 1. Families Nemonychidae, Anthribidae, Belidae, Ithyceridae, Rhynchitidae, Brachyceridae and Brentidae // Ukrainian Journal of Ecology. Vol. 8. No. 1. P. 780–831. https://doi.org/10.15421/2018_280
Poinar Jr., G.O., Brown A.E., Legalov A.A. 2019. A new weevil, Periosomerus tanyorhynchus gen. et sp. nov. (Coleoptera; Ithyceridae) in mid-Cretaceous Burmese amber // Cretaceous Research. Vol. 104. No. 104195. P. 1-4. https://doi.org/10.1016/j.cretres.2019.104195

Statements like "Here is the first reliable indication of Urodontinae in the fossil record" (Line 565) should be followed with some explanation as to why this is important. Does this correspond with our understanding of weevil phylogeny? Is it earlier/later than might be expected? What confidence do we have that these Urodontinae are in that tribe?
-In this work, these statements are needed only for consistent display of events in the fossil record. I did not have the task of confirming the belonging of certain species to certain taxa. This work was done in my review of fossil weevils (Legalov, 2015). If we talk about the time when taxa of the tribe level appeared in the fossil record, then I think that this happened in the Late Cretaceous. Unfortunately, data on weevils of the Late (not Middle - Burmese amber) Cretaceous are practically absent.

A summary of the timing, setting and importance of the Palaeogene would be a good addition for those readers who are not as familiar with the geological time. A figure showing the duration and context of the Palaeogene would be very useful in this respect.
-This was done.
All plots in the paper have a 3D effect added to them. This is inappropriate and unhelpful for presenting data in a scientific paper. All charts should be presented in two dimensions only. I would also recommend that all pie charts be changed to bar charts.
-This was done.

Other suggestions follow:
Table 1:
In several entries (e.g rows 15, 16, 76, 77, ... 544), only the genus name has been italicised.
-This was done.

Lines 136--139: These names have been validly published in 2019, in his reference 69.
-This was done.

Line 93: "(Figs. ??):" needs to be completed.
-This was done.

Line 165: "(reference)" needs to be inserted
-This was done.

Lines 134 & 135: Strange spiral-shaped glyphs are present in the PDF.
-This was done.

Line 679: "They have not yet been found in African, South Asian and Australian Paleogene locations" What locations are these? Does the context of these locations mean that Curculionoidea could be expected from them?
- Yes.

Figure 20 on Pg 37 should be Figure 21.
-This was done.

-Figure 25 is unnecessary if figure 26 is included.
-I think that Figure 26 does not duplicate Figure 25 and is needed to understand the change in fauna.

Figure 24 on pg 40 should be Figure 26.
-This was done.

Round 2

Reviewer 1 Report

the manuscipt considered is ready to be published

Author Response

I thank the reviewer for his work to improve my manuscript.

Reviewer 2 Report

The updated manuscript addresses some of my concerns; but I consider that the most serious flaw with the manuscript (the lack of a clear focus) still remains.

First, some comments on the Author's response to my first review.

>>"Unfortunately, data on weevils of the Late (not Middle - Burmese amber) Cretaceous are practically absent."
This is an interesting and important point, which the author should include somewhere in the paper, probably the Introduction.

>>"I did not have the task of confirming the belonging of certain species to certain taxa. This work was done in my review of fossil weevils (Legalov, 2015)"
Again, this rationale was not clear to me in the text of the current paper. I recommend that this also be included in the Introduction, and that this paper is cited when species included in that work are discussed.

>>"I do not understand the term "accepted classification". In science, everyone can voice an opinion if one is reasonable."
The author raises a point here which goes right to the depths of philosophy of science! Unfortunately, it's too big a question for me to discuss adequately here. However, whatever the arguments for the pros and cons of scientific consensus, the reality is that dissenting conclusions need to be clearly argued and have the supporting evidence clearly presented to a degree much more so than if the general classification held by the majority is followed.

By "accepted classification" I mean that presented in the recently published Handbook of Zoology (Leschen RAB, Beutel RG (Eds). 2014. Handbook of Zoology. Coleoptera, Beetles. Volume 3 Morphology and Systematics Phytophaga. De Gruyter, Berlin. https://doi.org/10.1515/9783110274462)

>>"In my numerous works (see the list) I have expressed my views and argued for them. The broad interpretation of this family Ithyceridae is my opinion based on extensive paleontological material. I see no convincing evidence that this position is wrong. "
I regret that I haven't yet taken the time to go through the author's reasoning for a broad concept of Ithyceridae, so I cannot comment on the evidence for or against his conclusions. However, it remains the case that his "Ithyceridae" has a very different composition than many other people's "Ithyceridae". I recommend that, at the very least, the author summarise his rationale for grouping the carines and ithycerines together, cite his prior papers on the subject, and make clear that his Ithyceridae is composed of the Ithycerinae + Caridae of the Handbook of Zoology.

>>"Molecular data cannot yet be reliable, since the development of this direction is just beginning. Perhaps in 20-30 years the situation with molecular research will change."
I am not as quick to dismiss the findings of molecular systematics research as the author; though I agree with him that their unquestioned acceptance is unwise. I am seeing consistent results from the various publications being produced by the groups working on the problem. Again, the author can hold a different conclusion, but he must be able to provide a rationale for it.

It is also worth pointing out that the desire for providing date estimates for phylogenetic trees makes works like the present paper of value to molecular systematics researchers. I would recommend that the author consider how the present work might be made more accessible to that audience.

The addition of subheadings, and the addition of Table 1 helps greatly.

Multiple instances: Change "Rovne" to "Rovno"
Multiple instances: Change "localitis" to "localities"

Date ranges: There are inconsistencies in the formatting of the date ranges. Some are older--younger (e.g. lines 43, ; while others are younger--older (Table 1 Bridgerian,

Keywords: Spelling of "taxonomical stuctute" should be corrected to "taxonomical structure"; but even then, I'm not certain that this is an informative keyword. Maybe something like "classification"?

Table 1: The addition of this table greatly helps when navigating the time periods discussed in the paper. However it is unclear what exactly the dates in each cell refer to (I believe they are the earliest date of each time period). I like the range given for the Bridgerian time (I assume start--end?), and would prefer to see that for each of the periods. Clarification of what the dates are should be given in the table caption.

Table 1: It would also be really cool if the height of each cell was proportional to the length of time of each age. Maybe this would be a challenge to prepare; but I'd encourage the author to consider it :)

Table 1: If this was turned into a figure, it would also be *really* neat to have a column where the age of each fossil deposit discussed in the paper was shown. Again, I think the value this would add to the paper would amply reward the effort involved in preparing it.

Fossil weevil table: Change caption from "Table 1" to "Table 2" and check all references to it in the text.

Fossil weevil table, row 15: Change '"Trpideres"' to '"Tropideres"'

"> -Figure 25 is unnecessary if figure 26 is included.
>> -I think that Figure 26 does not duplicate Figure 25 and is needed to understand the change in fauna."
I agree that Figure 26 more informative and should be retained. Figure 25 should be deleted.

I recognise the author's knowledge and experience with these fossil weevils, and I want him to clearly tell me (and his other readers) about them so that I/we can benefit from his efforts. Unfortunately, I feel that the current manuscript is still too vague and unfocused, making it hard to read and enjoy. The many minor errors also detract from the value of the paper. I make these comments to encourage the author to improve his writing and his paper so that his knowledge does not get dismissed.

Author Response

I thank the reviewer for his work to improve my manuscript.

"Unfortunately, data on weevils of the Late (not Middle - Burmese amber) Cretaceous are practically absent." This is an interesting and important point, which the author should include somewhere in the paper, probably the Introduction.

- I will not add this to the introduction, since this is published and known information (Legalov, 2012, 2015, etc.) not related to the Paleogene.

Legalov A.A. 2012. Fossil history of Mesozoic weevils (Coleoptera: Curculionoidea) // Insect Science. Vol. 19. No. 6. P. 683–698. https://doi.org/10.1111/j.1744-7917.2012.01508.x

Legalov A.A. 2015. Fossil Mesozoic and Cenozoic weevils (Coleoptera, Obrienioidea, Curculionoidea) // Paleontological Journal. Vol. 49. No 13. P. 1442–1513. https://doi.org/10.1134/S0031030115130067

>>"I did not have the task of confirming the belonging of certain species to certain taxa. This work was done in my review of fossil weevils (Legalov, 2015)"

Again, this rationale was not clear to me in the text of the current paper. I recommend that this also be included in the Introduction, and that this paper is cited when species included in that work are discussed.

- I added it to the Material and methods

>>"I do not understand the term "accepted classification". In science, everyone can voice an opinion if one is reasonable."

The author raises a point here which goes right to the depths of philosophy of science! Unfortunately, it's too big a question for me to discuss adequately here. However, whatever the arguments for the pros and cons of scientific consensus, the reality is that dissenting conclusions need to be clearly argued and have the supporting evidence clearly presented to a degree much more so than if the general classification held by the majority is followed.

- I agree with this, but the reality is that the presence of evidence does not affect the point of view, “ideas die with their carriers”.

By "accepted classification" I mean that presented in the recently published Handbook of Zoology (Leschen RAB, Beutel RG (Eds). 2014. Handbook of Zoology. Coleoptera, Beetles. Volume 3 Morphology and Systematics Phytophaga. De Gruyter, Berlin. https://doi.org/10.1515/9783110274462)

-This is just one of the reference books, little known and inaccessible for most researchers (I am sure that in Russia and other poor countries there is no printed version at all). Curculionoidea in it is written rather superficially and most of the provisions on the system are debatable.

>>"In my numerous works (see the list) I have expressed my views and argued for them. The broad interpretation of this family Ithyceridae is my opinion based on extensive paleontological material. I see no convincing evidence that this position is wrong. " I regret that I haven't yet taken the time to go through the author's reasoning for a broad concept of Ithyceridae, so I cannot comment on the evidence for or against his conclusions. However, it remains the case that his "Ithyceridae" has a very different composition than many other people's "Ithyceridae". I recommend that, at the very least, the author summarise his rationale for grouping the carines and ithycerines together, cite his prior papers on the subject, and make clear that his Ithyceridae is composed of the Ithycerinae + Caridae of the Handbook of Zoology.

- I added it to the material and methods.

>>"Molecular data cannot yet be reliable, since the development of this direction is just beginning. Perhaps in 20-30 years the situation with molecular research will change." I am not as quick to dismiss the findings of molecular systematics research as the author; though I agree with him that their unquestioned acceptance is unwise. I am seeing consistent results from the various publications being produced by the groups working on the problem. Again, the author can hold a different conclusion, but he must be able to provide a rationale for it.

- I agree that this needs to be justified, but my ms is not devoted to the Curculionoid system, but to the analysis of their distribution in the fossil record.

It is also worth pointing out that the desire for providing date estimates for phylogenetic trees makes works like the present paper of value to molecular systematics researchers. I would recommend that the author consider how the present work might be made more accessible to that audience.

- Yes, I added Ma in the ms, where it is possible.

The addition of subheadings, and the addition of Table 1 helps greatly.

- OK

Multiple instances: Change "Rovne" to "Rovno"

- It was done

Multiple instances: Change "localitis" to "localities"

- It was done

Date ranges: There are inconsistencies in the formatting of the date ranges. Some are older--younger (e.g. lines 43, ; while others are younger--older (Table 1 Bridgerian,

- It was done

Keywords: Spelling of "taxonomical stuctute" should be corrected to "taxonomical structure"; but even then, I'm not certain that this is an informative keyword. Maybe something like "classification"?

-It was done. No, Classification it is worse because the ms is devoted to the analysis of the fauna structure, and not to the system and classification of weevils.

Table 1: The addition of this table greatly helps when navigating the time periods discussed in the paper. However it is unclear what exactly the dates in each cell refer to (I believe they are the earliest date of each time period). I like the range given for the Bridgerian time (I assume start--end?), and would prefer to see that for each of the periods. Clarification of what the dates are should be given in the table caption.

- I was done.

Table 1: It would also be really cool if the height of each cell was proportional to the length of time of each age. Maybe this would be a challenge to prepare; but I'd encourage the author to consider it :)

- I was done.

Table 1: If this was turned into a figure, it would also be *really* neat to have a column where the age of each fossil deposit discussed in the paper was shown. Again, I think the value this would add to the paper would amply reward the effort involved in preparing it.

-I prepared a table of locations where I have indicated the age of them where possible.

Fossil weevil table: Change caption from "Table 1" to "Table 2" and check all references to it in the text.

- It was done

Fossil weevil table, row 15: Change '"Trpideres"' to '"Tropideres"'

- It was done

"> -Figure 25 is unnecessary if figure 26 is included. >> -I think that Figure 26 does not duplicate Figure 25 and is needed to understand the change in fauna."I agree that Figure 26 more informative and should be retained. Figure 25 should be deleted.

- It was done

I recognise the author's knowledge and experience with these fossil weevils, and I want him to clearly tell me (and his other readers) about them so that I/we can benefit from his efforts. Unfortunately, I feel that the current manuscript is still too vague and unfocused, making it hard to read and enjoy. The many minor errors also detract from the value of the paper. I make these comments to encourage the author to improve his writing and his paper so that his knowledge does not get dismissed.

- I thank you for your efforts to improve my manuscript and I try to correct it according to your recommendations (as much as possible).

Round 3

Reviewer 2 Report

The addition of the paragraph in Lines 59--72 is much appreciated. This summary of the author's views on the higher classification of the Curculionoidea, is very helpful for reconciling the system used in the paper with alternatives.

I suggest a number of other corrections below, which should be fairly quick to implement.

Line 23: Change "Studied the biodiversity of the past are very important" to "Studies of the biodiversity of the past are very important" or "Research into the biodiversity of the past is very important"

Line 46: Change "Curculionoidea are described (XREFS) and recorded (XREFS) in 53 localities, which are given in the chronological order in Table 2" "Curculionoidea have been found from 53 localities in NUMBER countries (XREFS), spanning all ages of the Palaeogene (Table 2)"

Table 2: I recommend that the order of rows be reversed to go from the most recent deposits to the most ancient ones, to make it consistent with Table 1. I much prefer this way of conveying this information. Good job.

Line 53: Delete "The systematics of the superfamily Curculionoidea is currently fairly stable". This line is inconsistent with later statements (e.g. "The number of families .. are not universally accepted" lines 57--58), and by the author's not following the "accepted classification" ;)

Lines 53--58: Merge this paragraph with the following. I recommend that the author begins the paragraph with "The number of families ... not universally accepted (REFS). In this work the higher classification proposed by the author (REFS) is adopted."

>>>Keywords: Spelling of "taxonomical stuctute" should be corrected to "taxonomical structure"; but even then, I'm not certain that this is an informative keyword. Maybe something like "classification"?
>It was done. No, Classification it is worse because the ms is devoted to the analysis of the fauna structure, and not to the system and classification of weevils.
Ah, I see. In that case I think that "faunal structure" is a more informative keyword than "taxonomical structure"

Lines 53--57: These sentences sentences considering the Obrieniidae contains some redundancies. I recommend rewording to make it more succinct

Line 62: Change "volume" to "composition"

Lines 62--63: Change "Ithyceridae and Caridae are usually regarded as unrelated groups (REFS). The author understands the family Ithyceridae ..." to "Ithyceridae and Caridae are often regarded as unrelated groups (REFS), however, the author understands the family Ithyceridae ..."

Line 64: Change "as diverse" to "as a diverse"

Line 66: Change "Rhynchitidae and Attelabidae are accepted as independant families" to "Rhynchitidae and Attelabidae are considered as independant families"

Line 69: Delete "Ithyceridae ... separate family". The author's view of the Ithyceridae has already been summarised.

Table 3: I preferred the text in this table when it was left-aligned.

Table 3: I recommend the author go through the name carefully to check the spelling of names. In particular, the spelling of names in rows 15, 251, 398, 437, 462, 472, 507, 537, 544 look odd. I haven't checked against original descriptions, but encourage the author to do so.

Line 117: Change "27 species" to "Twenty seven species". It's usually considered poor style to begin a sentence with numerals. Either write the number in words, or change the order of the sentence. Also needed for lines 168, 251, 264, 299, 320, 344, 372, 430, 595, 614, 640, ...

Line 149: Change "Naupactini similar to the Neotropical forms confirms the faunogenetic" to "Naupactini similar to Neotropical forms provides additional evidence for faunogenetic"

Line 402: "lived on grassy vegetation" I suggest "riverine meadows" or a similar description to avoid the word "grass". These taxa are more likely to be associated with the diverse range of herbs in the habitat, rather than the grasses as such.

Line 421: Change "little" to "few".

Lines 506, 584: Change "localiti" to "locality"

Line 529: Change "I is noted that Choraginae were" to "Choraginae were"

Lines 589--591 "The most of the known American species in the Oligocene were probably associated with woody-shrubby vegetation and single with grassy." The intent of this sentence is unclear and should be rewritten.

Line 599: "four species of the subfamily Conoderinae, one species of the subfamily Conoderinae". Correct this duplication.

Line 601: "Common species and genera between them are absent". I assume the author was intending something like "The most common species and genera of each site are absent from the other", or something like that? The sentence needs clarification.

Author Response

Dear Sir, I thank you for your comments and I corrected ms according them.

Line 23: Change "Studied the biodiversity of the past are very important" to "Studies of the biodiversity of the past are very important" or "Research into the biodiversity of the past is very important"

- It was done.

Line 46: Change "Curculionoidea are described (XREFS) and recorded (XREFS) in 53 localities, which are given in the chronological order in Table 2" "Curculionoidea have been found from 53 localities in NUMBER countries (XREFS), spanning all ages of the Palaeogene (Table 2)"

- It was done.

Table 2: I recommend that the order of rows be reversed to go from the most recent deposits to the most ancient ones, to make it consistent with

- I cannot correct the table 2 because the order of taxa in the table 3 is done according to the table 2. I corrected table 1.

Line 53: Delete "The systematics of the superfamily Curculionoidea is currently fairly stable". This line is inconsistent with later statements (e.g. "The number of families .. are not universally accepted" lines 57--58), and by the author's not following the "accepted classification" ;)

- It was done.

Lines 53--58: Merge this paragraph with the following. I recommend that the author begins the paragraph with "The number of families ... not universally accepted (REFS). In this work the higher classification proposed by the author (REFS) is adopted."

- It was done.

Ah, I see. In that case I think that "faunal structure" is a more informative keyword than "taxonomical structure"

- It was done.

Lines 53--57: These sentences sentences considering the Obrieniidae contains some redundancies. I recommend rewording to make it more succinct

- It was done.

Line 62: Change "volume" to "composition"

- It was done.

Lines 62--63: Change "Ithyceridae and Caridae are usually regarded as unrelated groups (REFS). The author understands the family Ithyceridae ..." to "Ithyceridae and Caridae are often regarded as unrelated groups (REFS), however, the author understands the family Ithyceridae ..."

- It was done.

Line 64: Change "as diverse" to "as a diverse"

- It was done.

Line 66: Change "Rhynchitidae and Attelabidae are accepted as independant families" to "Rhynchitidae and Attelabidae are considered as independant families"

- It was done.

Line 69: Delete "Ithyceridae ... separate family". The author's view of the Ithyceridae has already been summarised.

- It was done.

Table 3: I preferred the text in this table when it was left-aligned.

- It was done.

Table 3: I recommend the author go through the name carefully to check the spelling of names. In particular, the spelling of names in rows 15, 251, 398, 437, 462, 472, 507, 537, 544 look odd. I haven't checked against original descriptions, but encourage the author to do so.

- It was done.

Line 117: Change "27 species" to "Twenty seven species". It's usually considered poor style to begin a sentence with numerals. Either write the number in words, or change the order of the sentence. Also needed for lines 168, 251, 264, 299, 320, 344, 372, 430, 595, 614, 640, ...

- It was done.

Line 149: Change "Naupactini similar to the Neotropical forms confirms the faunogenetic" to "Naupactini similar to Neotropical forms provides additional evidence for faunogenetic"

- It was done.

Line 402: "lived on grassy vegetation" I suggest "riverine meadows" or a similar description to avoid the word "grass". These taxa are more likely to be associated with the diverse range of herbs in the habitat, rather than the grasses as such.

- It was done.

Line 421: Change "little" to "few".

- It was done.

Lines 506, 584: Change "localiti" to "locality"

- It was done.

Line 529: Change "I is noted that Choraginae were" to "Choraginae were"

- It was done.

Lines 589--591 "The most of the known American species in the Oligocene were probably associated with woody-shrubby vegetation and single with grassy." The intent of this sentence is unclear and should be rewritten.

- It was done.

Line 599: "four species of the subfamily Conoderinae, one species of the subfamily Conoderinae". Correct this duplication.

- It was done.

Line 601: "Common species and genera between them are absent". I assume the author was intending something like "The most common species and genera of each site are absent from the other", or something like that? The sentence needs clarification.

- It was done.

The English of this manuscript has been improved again by Dr. George Poinar (Oregon State Univ.)